# Diffusion Schrödinger Bridge Matching

**Yuyang Shi**[*]            **Valentin De Bortoli**[*]            **Andrew Campbell**            **Arnaud Doucet**
University of Oxford            ENS ULM            University of Oxford            University of Oxford

## Abstract

Solving transport problems, i.e. finding a map transporting one given distribution to another, has numerous applications in machine learning. Novel mass transport methods motivated by generative modeling have recently been proposed, e.g. Denoising Diffusion Models (DDMs) and Flow Matching Models (FMMs) implement such a transport through a Stochastic Differential Equation (SDE) or an Ordinary Differential Equation (ODE). However, while it is desirable in many applications to approximate the deterministic dynamic Optimal Transport (OT) map which admits attractive properties, DDMs and FMMs are not guaranteed to provide transports close to the OT map. In contrast, Schrödinger bridges (SBs) compute stochastic dynamic mappings which recover entropy-regularized versions of OT. Unfortunately, existing numerical methods approximating SBs either scale poorly with dimension or accumulate errors across iterations. In this work, we introduce Iterative Markovian Fitting (IMF), a new methodology for solving SB problems, and Diffusion Schrödinger Bridge Matching (DSBM), a novel numerical algorithm for computing IMF iterates. DSBM significantly improves over previous SB numerics and recovers as special/limiting cases various recent transport methods. We demonstrate the performance of DSBM on a variety of problems.

## 1 Introduction

Mass transport problems are ubiquitous in machine learning (Peyré and Cuturi, 2019). For discrete measures, the Optimal Transport (OT) map can be computed exactly but is computationally intensive. In a landmark paper, Cuturi (2013) showed that an entropy-regularized version of OT can be computed more efficiently using the Sinkhorn algorithm (Sinkhorn, 1967). This has enabled the use of OT techniques in a variety of applications ranging from biology (Bunne et al., 2022) to shape correspondence (Feydy et al., 2017). However, applications involving high-dimensional continuous distributions and/or large datasets remain challenging for these techniques.

One of such data-rich applications is generative modeling, a central transport problem in machine learning which requires designing a deterministic or stochastic mapping transporting a reference "noise" distribution to the data distribution. For example, Generative Adversarial Networks (Goodfellow et al., 2014) define a static, deterministic transport map, while Denoising Diffusion Models (DDMs) (Song et al., 2021b; Ho et al., 2020) build a dynamic, stochastic transport map by simulating a Stochastic Differential Equation (SDE), whose drift is learned using score matching (Hyvärinen, 2005; Vincent, 2011). The excellent performances of DDMs have motivated recent developments of Bridge Matching and Flow Matching models, which are dynamic transport maps using SDEs (Song et al., 2021a; Peluchetti, 2021; Liu, 2022; Albergo et al., 2023) or ODEs (Albergo and Vanden-Eijnden, 2023; Heitz et al., 2023; Lipman et al., 2023; Liu et al., 2023b). Compared to DDMs, Bridge and Flow Matching methods do not rely on a forward "noising" diffusion converging to the reference distribution in infinite time, and are also more generally applicable as they can approximate transport maps between two general distributions based on their samples. Nonetheless, these transport maps

---

[*]Equal contribution.

37th Conference on Neural Information Processing Systems (NeurIPS 2023).

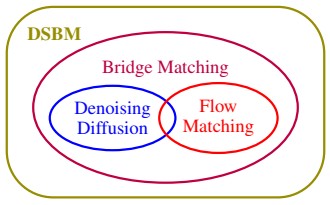

Figure 1: Relationship between DSBM and existing methods.

| | Sets for alternating projections | Preserved properties |
|---|---|---|
| IPF | $\mathbb{P}_0 = \pi_0; \mathbb{P}_T = \pi_T$ | $\mathcal{M}, \mathcal{R}(\mathbb{Q})$ |
| IMF | $\mathcal{M}; \mathcal{R}(\mathbb{Q})$ | $\mathbb{P}_0 = \pi_0, \mathbb{P}_T = \pi_T$ |

Table 1: Comparison between Iterative Markovian Fitting (IMF) and Iterative Proportional Fitting (IPF). The Schrödinger Bridge is the *unique* $\mathbb{P}$ s.t. $\mathbb{P}_0 = \pi_0$, $\mathbb{P}_T = \pi_T$, $\mathbb{P} \in \mathcal{M}$, $\mathbb{P} \in \mathcal{R}(\mathbb{Q})$ simultaneously by Proposition 5. $\mathcal{M}$ is the space of (regular) Markov measures and $\mathcal{R}(\mathbb{Q})$ the space of reciprocal measures of $\mathbb{Q}$.

are not necessarily close to the OT map minimizing the Wasserstein-2 metric, which is appealing for its many attractive properties (Peyré and Cuturi, 2019; Villani, 2009).

In contrast, the Schrödinger Bridge (SB) problem is a dynamic version of entropy-regularized OT (EOT) (Föllmer, 1988; Léonard, 2014b). The SB is the finite-time diffusion which admits as initial and terminal distributions the two distributions of interest and is the closest in Kullback–Leibler divergence to a reference diffusion. Numerous methods to approximate SBs numerically have been proposed, see e.g. (Bernton et al., 2019; Chen et al., 2016; Finlay et al., 2020; Caluya and Halder, 2021; Pavon et al., 2021), but these techniques tend to be restricted to low-dimensional settings. Recently, novel techniques using diffusion-based ideas have been proposed in (De Bortoli et al., 2021; Vargas et al., 2021; Chen et al., 2022) based on Iterative Proportional Fitting (IPF) (Fortet, 1940; Kullback, 1968; Rüschendorf and Thomsen, 1993), a continuous state-space extension of the Sinkhorn algorithm (Essid and Pavon, 2019). These approaches have been shown to scale better empirically, but numerical errors tend to accumulate over iterations (Fernandes et al., 2021).

In this paper, our contributions are three-fold. First, we introduce Iterative Markovian Fitting (IMF), a new procedure to compute SBs which alternates between projecting on the space of *Markov processes* and on the *reciprocal class*, i.e. the measures which have the same bridge as the reference measure of SB (Léonard et al., 2014). We establish various theoretical results for IMF. Contrary to IPF, the IMF iterates always preserve the initial and terminal distributions. The differences between IPF and IMF are presented in Table 1. Second, we propose Diffusion Schrödinger Bridge Matching (DSBM), a novel algorithm approximating numerically the SB solution derived from IMF. DSBM requires at each iteration solving a simple regression problem in the spirit of Bridge and Flow Matching, and does not suffer from the time-discretization and "forgetting" issues of previous DSB techniques (De Bortoli et al., 2021; Vargas et al., 2021; Chen et al., 2022). Finally, we demonstrate the performance of DSBM on a variety of transport tasks.[2]

**Notations.** We denote by $\mathcal{P}(\mathcal{C})$, the space of *path measures*, i.e. $\mathcal{P}(\mathcal{C}) = \mathcal{P}(\mathrm{C}([0,T], \mathbb{R}^d))$ where $T > 0$. The subset of *Markov* path measures associated with an SDE of the form $\mathrm{d}\mathbf{X}_t = v_t(\mathbf{X}_t)\mathrm{d}t + \sigma_t \mathrm{d}\mathbf{B}_t$, with $\sigma, v$ locally Lipschitz, is denoted $\mathcal{M}$. For any $\mathbb{Q} \in \mathcal{M}$, the *reciprocal class* of $\mathbb{Q}$ is denoted $\mathcal{R}(\mathbb{Q})$, see Definition 3. We also denote $\mathbb{Q}_t$ its marginal distribution at time $t$, $\mathbb{Q}_{s,t}$ the joint distribution at times $s$ and $t$, $\mathbb{Q}_{s|t}$ the conditional distribution at time $s$ given state at time $t$, and $\mathbb{Q}_{|0,T} \in \mathcal{P}(\mathcal{C})$ its *diffusion bridge*. Unless specified otherwise, all gradient operators $\nabla$ are w.r.t. the variable $x_t$ with time index $t$. Let $(\mathsf{X}, \mathcal{X})$ and $(\mathsf{Y}, \mathcal{Y})$ be probability spaces. Given a Markov kernel $\mathrm{K} : \mathsf{X} \times \mathcal{Y} \rightarrow [0,1]$ and a probability measure $\mu$ defined on $\mathcal{X}$, we write $\mu\mathrm{K}$ the probability measure on $\mathcal{Y}$ such that for any $\mathsf{A} \in \mathcal{Y}$ we have $\mu\mathrm{K}(\mathsf{A}) = \int_{\mathsf{X}} \mathrm{K}(x, \mathsf{A})\mathrm{d}\mu(x)$. In particular, for any joint distribution $\Pi_{0,T}$ over $\mathbb{R}^d \times \mathbb{R}^d$, we denote the *mixture of bridges* measure as $\Pi = \Pi_{0,T}\mathbb{Q}_{|0,T} \in \mathcal{P}(\mathcal{C})$, which is short for $\Pi(\cdot) = \int_{\mathbb{R}^d \times \mathbb{R}^d} \mathbb{Q}_{|0,T}(\cdot|x_0, x_T)\Pi_{0,T}(\mathrm{d}x_0, \mathrm{d}x_T)$.

## 2 Dynamic Mass Transport Techniques

### 2.1 Denoising Diffusion and Bridge Matching Models

Denoising Diffusion Models (Song et al., 2021b; Ho et al., 2020) are a popular class of generative models. They define a forward noising process $\mathbb{Q} \in \mathcal{M}$ using the SDE $\mathrm{d}\mathbf{X}_t = -\frac{1}{2}\mathbf{X}_t\mathrm{d}t + \mathrm{d}\mathbf{B}_t$ on the time-interval $[0, T]$, where $\mathbf{X}_0 \in \mathbb{R}^d$ is drawn from the data distribution $\pi_0$ and $(\mathbf{B}_t)_{t \in [0,T]}$ is a $d$-

---

[2]Code can be found at `https://github.com/yuyang-shi/dsbm-pytorch`.

dimensional Brownian motion. This diffusion[3] converges towards the standard Gaussian distribution $\mathrm{N}(0, \mathrm{Id})$ as $T \to \infty$. A generative model is given by its *time-reversal* $(\mathbf{Y}_t)_{t \in [0,T]} = (\mathbf{X}_{T-t})_{t \in [0,T]}$, where $\mathbf{Y}_0 \sim \mathbb{Q}_T$ and $\mathrm{d}\mathbf{Y}_t = \{\frac{1}{2}\mathbf{Y}_t + \nabla \log \mathbb{Q}_{T-t}(\mathbf{Y}_t)\}\mathrm{d}t + \mathrm{d}\mathbf{B}_t$ (Anderson, 1982; Haussmann and Pardoux, 1986). In practice, $(\mathbf{Y}_t)_{t \in [0,T]}$ is initialized with $\mathbf{Y}_0 \sim \pi_T = \mathrm{N}(0, \mathrm{Id})$, and the *Stein score* $\nabla \log \mathbb{Q}_t(x_t) = \mathbb{E}_{\mathbb{Q}_{0|t}}[\nabla \log \mathbb{Q}_{t|0}(\mathbf{X}_t|\mathbf{X}_0) \mid \mathbf{X}_t = x_t]$ is approximated using a neural network $s_\theta(t, x_t)$ minimizing the *denoising score matching* loss $\mathbb{E}_{\mathbb{Q}_{0,t}}[\|\nabla \log \mathbb{Q}_{t|0}(\mathbf{X}_t|\mathbf{X}_0) - s_\theta(t, \mathbf{X}_t)\|^2]$.

An alternative to considering the time-reversal of a forward noising process is to "build bridges" between the two distributions and learn a *mimicking* diffusion process. This approach generalizes DDMs and allows for more flexible choices of sampling processes. We call this framework *Bridge Matching* and adopt a presentation similar to Peluchetti (2021); Liu et al. (2022b), where $\pi_T$ is the data distribution.[4] We denote $\mathbb{Q} \in \mathcal{M}$ the path measure associated with the following process

$$\mathrm{d}\mathbf{X}_t = f_t(\mathbf{X}_t)\mathrm{d}t + \sigma_t \mathrm{d}\mathbf{B}_t, \qquad \mathbf{X}_0 \sim \mathbb{Q}_0. \tag{1}$$

Consider now the distribution of this process pinned down at an initial and terminal point $x_0, x_T$, denoted $\mathbb{Q}_{|0,T}(\cdot|x_0, x_T)$. Under mild assumptions, the *pinned* process $\mathbb{Q}_{|0,T}(\cdot|x_0, x_T)$ is a *diffusion bridge* and is given by

$$\mathrm{d}\mathbf{X}_t^{0,T} = \{f_t(\mathbf{X}_t^{0,T}) + \sigma_t^2 \nabla \log \mathbb{Q}_{T|t}(x_T|\mathbf{X}_t^{0,T})\}\mathrm{d}t + \sigma_t \mathrm{d}\mathbf{B}_t, \qquad \mathbf{X}_0^{0,T} = x_0, \tag{2}$$

which satisfies $\mathbf{X}_T^{0,T} = x_T$ using Doob $h$-transform theory (Rogers and Williams, 2000). Next, we define an independent coupling $\Pi_{0,T} = \pi_0 \otimes \pi_T$, and let $\Pi = \Pi_{0,T}\mathbb{Q}_{|0,T}$. This path measure $\Pi$ is a *mixture of bridges*. We aim to find a Markov diffusion $\mathrm{d}\mathbf{Y}_t = \{f_t(\mathbf{Y}_t) + v_t(\mathbf{Y}_t)\}\mathrm{d}t + \sigma_t \mathrm{d}\mathbf{B}_t$ on $[0,T]$ which admits the same marginals as $\Pi$; i.e. for any $t \in [0,T]$, $\mathbf{Y}_t \sim \Pi_t$, so $\mathbf{Y}_T \sim \pi_T$. For such $v_t$, a generative model for sampling data distribution $\pi_T$ is obtained by simulating $(\mathbf{Y}_t)_{t \in [0,T]}$. It can be verified that indeed $\mathbf{Y}_t \sim \Pi_t$ for $v_t^\star(x_t) = \sigma_t^2 \mathbb{E}_{\Pi_{T|t}}[\nabla \log \mathbb{Q}_{T|t}(\mathbf{X}_T|\mathbf{X}_t) \mid \mathbf{X}_t = x_t]$. We present the theory behind this idea more formally using Markovian projections in Section 3.1. In practice, we do not have access to $v_t^\star$ and it is learned using neural networks with regression loss

$$\mathbb{E}_{\Pi_{t,T}}[\|\sigma_t^2 \nabla \log \mathbb{Q}_{T|t}(\mathbf{X}_T|\mathbf{X}_t) - v_\theta(t, \mathbf{X}_t)\|^2]. \tag{3}$$

For $f_t = 0$ and $\sigma_t = \sigma$, $\mathbb{Q}_{|0,T}$ is a *Brownian Bridge* and we have

$$\mathbf{X}_t^{0,T} = \tfrac{t}{T}x_T + (1 - \tfrac{t}{T})x_0 + \sigma_t(\mathbf{B}_t - \tfrac{t}{T}\mathbf{B}_T), \quad \mathrm{d}\mathbf{X}_t^{0,T} = \{(x_T - \mathbf{X}_t^{0,T})/(T - t)\}\mathrm{d}t + \sigma_t \mathrm{d}\mathbf{B}_t, \tag{4}$$

with $(\mathbf{B}_t - \tfrac{t}{T}\mathbf{B}_T) \sim \mathrm{N}(0, t(1 - \tfrac{t}{T})\,\mathrm{Id})$. The regression loss (3) associated with (4) is given by

$$\mathbb{E}_{\Pi_{t,T}}[\|(\mathbf{X}_T - \mathbf{X}_t)/(T - t) - v_\theta(t, \mathbf{X}_t)\|^2]. \tag{5}$$

Letting $\sigma \to 0$, we recover Flow Matching models (see Appendix A.1 for further details).

## 2.2 Schrödinger Bridges and Optimal Transport

The Schrödinger Bridge (SB) problem (Schrödinger, 1932) consists in finding a path measure $\mathbb{P}^{\mathrm{SB}} \in \mathcal{P}(\mathcal{C})$ such that

$$\mathbb{P}^{\mathrm{SB}} = \mathrm{argmin}_{\mathbb{P}}\{\mathrm{KL}(\mathbb{P}|\mathbb{Q}) \ : \ \mathbb{P}_0 = \pi_0, \ \mathbb{P}_T = \pi_T\}, \tag{6}$$

where $\mathbb{Q} \in \mathcal{P}(\mathcal{C})$ is a reference path measure. In what follows, we consider $\mathbb{Q}$ defined by the diffusion process (1) which is Markov, and without loss of generality, we assume $\mathbb{Q}_0 = \pi_0$. Hence $\mathbb{P}^{\mathrm{SB}}$ is the path measure closest to $\mathbb{Q}$ in terms of Kullback–Leibler divergence which satisfies the initial and terminal constraints $\mathbb{P}_0^{\mathrm{SB}} = \pi_0$ and $\mathbb{P}_T^{\mathrm{SB}} = \pi_T$.

Another crucial property of $\mathbb{P}^{\mathrm{SB}}$ is that it can also be defined as a mixture of bridges $\mathbb{P}^{\mathrm{SB}} = \Pi_{0,T}^{\mathrm{SB}}\mathbb{Q}_{|0,T}$, where $\Pi_{0,T}^{\mathrm{SB}} = \mathrm{argmin}_{\Pi_{0,T}}\{\mathrm{KL}(\Pi_{0,T}|\mathbb{Q}_{0,T}) \ : \ \Pi_0 = \pi_0, \ \Pi_T = \pi_T\}$ is the solution of the *static* SB problem (Léonard, 2014b). In particular, for $\mathbb{Q}$ associated with $(\sigma\mathbf{B}_t)_{t \in [0,T]}$ we have

$$\Pi_{0,T}^{\mathrm{SB}} = \mathrm{argmin}_{\Pi_{0,T}}\{\mathbb{E}_{\Pi_{0,T}}[\|\mathbf{X}_0 - \mathbf{X}_T\|^2] - 2\sigma^2 T\,\mathrm{H}(\Pi_{0,T}) \ : \ \Pi_0 = \pi_0, \ \Pi_T = \pi_T\},$$

---

[3]This is known as the Ornstein–Uhlenbeck (OU) process or VPSDE (Song et al., 2021b).

[4]To keep notations consistent with existing works, $\pi_0$ is the data distribution in the context of DDM and SB, whereas $\pi_T$ is the data distribution in Bridge Matching. However, both SB and Bridge Matching methods allow transfer between arbitrary distributions $\pi_0, \pi_T$, so this distinction is not important.

where $H(\mu)$ denotes the entropy, i.e. $\Pi_{0,T}^{\mathrm{SB}}$ is the solution of the entropy-regularized OT problem. In this case, the SB can also be obtained theoretically by solving the following problem (Dai Pra, 1991)

$$v_{\mathrm{SB}} = \mathrm{argmin}_v \{ \int_0^T \mathbb{E}_{\mathbb{P}_t}[||v(t, \mathbf{X}_t)||^2] \mathrm{d}t \; : \; \mathrm{d}\mathbf{X}_t = v(t, \mathbf{X}_t) \mathrm{d}t + \sigma \mathrm{d}\mathbf{B}_t, \; \mathbb{P}_0 = \pi_0, \; \mathbb{P}_T = \pi_T \}.$$

Then $\mathbb{P}^{\mathrm{SB}}$ is given by the SDE with drift $v_{\mathrm{SB}}$ initialized with $\mathbf{X}_0 \sim \pi_0$. For $\sigma = 0$, we recover the classical OT problem and the Benamou–Brenier formula (Benamou and Brenier, 2000).

A common approach to solve (6) is the Iterative Proportional Fitting (IPF) method (Fortet, 1940; Kullback, 1968; Rüschendorf, 1995) defining a sequence of path measures $(\tilde{\mathbb{P}}^n)_{n \in \mathbb{N}}$ where

$$\tilde{\mathbb{P}}^{2n+1} = \mathrm{argmin}_{\tilde{\mathbb{P}}}\{\mathrm{KL}(\tilde{\mathbb{P}}|\tilde{\mathbb{P}}^{2n}) \; : \; \tilde{\mathbb{P}}_T = \pi_T\}, \; \tilde{\mathbb{P}}^{2n+2} = \mathrm{argmin}_{\tilde{\mathbb{P}}}\{\mathrm{KL}(\tilde{\mathbb{P}}|\tilde{\mathbb{P}}^{2n+1}) \; : \; \tilde{\mathbb{P}}_0 = \pi_0\}, \; (7)$$

with initialization $\tilde{\mathbb{P}}^0 = \mathbb{Q}$. This procedure alternates between projections on the set of path measures with given initial distribution $\pi_0$ and terminal distribution $\pi_T$. It can be shown (De Bortoli et al., 2021) that $(\tilde{\mathbb{P}}^n)_{n \in \mathbb{N}}$ are associated with diffusions and that for any $n \in \mathbb{N}$, $\tilde{\mathbb{P}}^{2n+1}$ is the time-reversal of $\tilde{\mathbb{P}}^{2n}$ with initialization $\pi_T$, and $\tilde{\mathbb{P}}^{2n+2}$ is the time-reversal of $\tilde{\mathbb{P}}^{2n+1}$ with initialization $\pi_0$. Leveraging this property, De Bortoli et al. (2021) proposed Diffusion Schrödinger Bridge (DSB), an algorithm which learns the time-reversals iteratively. In particular, DDMs can be seen as the first iteration of DSB.

# 3 Iterative Markovian Fitting

## 3.1 Markovian Projection and Reciprocal Projection

**Markovian Projection.** Projecting on Markov measures is a key ingredient in our methodology and in the Bridge Matching framework. This concept was introduced multiple times in the literature (Gyöngy, 1986; Peluchetti, 2021; Liu et al., 2022b). In particular, we focus on Markovian projection of path measures given by a mixture of bridges $\Pi = \Pi_{0,T} \mathbb{Q}_{|0,T} \in \mathcal{P}(\mathcal{C})$.

**Definition 1.** *Assume that $\mathbb{Q}$ is given by (1) and that for any $(x_0, x_T) \in \mathbb{R}^d$, $\mathbb{Q}_{|0,T}(\cdot|x_0, x_T)$ is associated with $(\mathbf{X}_t^{0,T})_{t \in [0,T]}$ given by $\mathrm{d}\mathbf{X}_t^{0,T} = \{f_t(\mathbf{X}_t^{0,T}) + \sigma_t^2 \nabla \log \mathbb{Q}_{T|t}(x_T | \mathbf{X}_t^{0,T})\} \mathrm{d}t + \sigma_t \mathrm{d}\mathbf{B}_t$, with $\sigma : [0,T] \to (0, +\infty)$. Then, when it is well-defined, we introduce the Markovian projection of $\Pi$, $\mathbb{M}^\star = \mathrm{proj}_{\mathcal{M}}(\Pi) \in \mathcal{M}$, which is associated with the SDE*

$$\mathrm{d}\mathbf{X}_t^\star = \{f_t(\mathbf{X}_t^\star) + v_t^\star(\mathbf{X}_t^\star)\}\mathrm{d}t + \sigma_t \mathrm{d}\mathbf{B}_t, \qquad v_t^\star(x_t) = \sigma_t^2 \mathbb{E}_{\Pi_{T|t}}[\nabla \log \mathbb{Q}_{T|t}(\mathbf{X}_T | \mathbf{X}_t) \mid \mathbf{X}_t = x_t].$$

Note that in our definition $\sigma_t > 0$ so $\nabla \log \mathbb{Q}_{T|t}(x_T | x_t)$ is well-defined, but Flow Matching can be recovered as the *deterministic* case in the limit $\sigma_t = \sigma \to 0$. In the following proposition, we show that the Markovian projection is indeed a projection for the *reverse* Kullback–Leibler divergence, and that it preserves marginals of $\Pi_t$.

**Proposition 2.** *Assume that $\sigma_t > 0$. Let $\mathbb{M}^\star = \mathrm{proj}_{\mathcal{M}}(\Pi)$. Then, under mild assumptions, we have*

$$\mathbb{M}^\star = \mathrm{argmin}_{\mathbb{M}}\{\mathrm{KL}(\Pi|\mathbb{M}) \; : \; \mathbb{M} \in \mathcal{M}\},$$

$$\mathrm{KL}(\Pi|\mathbb{M}^\star) = \tfrac{1}{2} \int_0^T \mathbb{E}_{\Pi_{0,t}}[||\sigma_t^2 \mathbb{E}_{\Pi_{T|0,t}}[\nabla \log \mathbb{Q}_{T|t}(\mathbf{X}_T|\mathbf{X}_t) \mid \mathbf{X}_0, \mathbf{X}_t] - v_t^\star(\mathbf{X}_t)||^2]/\sigma_t^2 \mathrm{d}t.$$

*In addition, we have that for any $t \in [0,T]$, $\mathbb{M}_t^\star = \Pi_t$. In particular, $\mathbb{M}_T^\star = \Pi_T$.*

**Reciprocal Projection.** While the Markovian projection ensures that the obtained measure is Markov, the associated *bridge* measure is not preserved in general, i.e. $\mathrm{proj}_{\mathcal{M}}(\Pi)_{|0,T} \neq \Pi_{|0,T} = \mathbb{Q}_{|0,T}$. Measures with same bridge as $\mathbb{Q}$ are said to be in its *reciprocal class* (Léonard et al., 2014).

**Definition 3.** *$\Pi \in \mathcal{P}(\mathcal{C})$ is in the reciprocal class $\mathcal{R}(\mathbb{Q})$ of $\mathbb{Q} \in \mathcal{M}$ if $\Pi = \Pi_{0,T} \mathbb{Q}_{|0,T}$. We define the reciprocal projection of $\mathbb{P} \in \mathcal{P}(\mathcal{C})$ as $\Pi^\star = \mathrm{proj}_{\mathcal{R}(\mathbb{Q})}(\mathbb{P}) = \mathbb{P}_{0,T} \mathbb{Q}_{|0,T}$.*

Similarly to Proposition 2, we have the following result, which justifies the term reciprocal projection.

**Proposition 4.** *Let $\mathbb{P} \in \mathcal{P}(\mathcal{C})$, $\Pi^\star = \mathrm{proj}_{\mathcal{R}(\mathbb{Q})}(\mathbb{P})$. Then, $\Pi^\star = \mathrm{argmin}_{\Pi}\{\mathrm{KL}(\mathbb{P}|\Pi) \; : \; \Pi \in \mathcal{R}(\mathbb{Q})\}$.*

The reciprocal projection $\Pi^\star$ of a Markov path measure $\mathbb{M}$ does not preserve the Markov property in general. In fact, the Schrödinger Bridge is the *unique* path measure which satisfies the initial and terminal conditions, is Markov and is in the reciprocal class of $\mathbb{Q}$, see (Léonard, 2014b).

**Proposition 5.** *Let $\mathbb{P}$ be a Markov measure in the reciprocal class of $\mathbb{Q}$ such that $\mathbb{P}_0 = \pi_0$, $\mathbb{P}_T = \pi_T$. Then, under assumptions on $\mathbb{Q}$, $\pi_0$ and $\pi_T$, $\mathbb{P}$ is unique and is equal to the Schrödinger Bridge $\mathbb{P}^{\mathrm{SB}}$.*

## 3.2 Iterative Markovian Fitting

Based on Proposition 5, we propose a novel methodology called *Iterative Markovian Fitting* (IMF) to solve Schrödinger Bridges. We consider a sequence $(\mathbb{P}^n)_{n \in \mathbb{N}}$ such that

$$\mathbb{P}^{2n+1} = \mathrm{proj}_{\mathcal{M}}(\mathbb{P}^{2n}), \qquad \mathbb{P}^{2n+2} = \mathrm{proj}_{\mathcal{R}(\mathbb{Q})}(\mathbb{P}^{2n+1}), \qquad (8)$$

with $\mathbb{P}^0$ such that $\mathbb{P}_0^0 = \pi_0$, $\mathbb{P}_T^0 = \pi_T$ and $\mathbb{P}^0 \in \mathcal{R}(\mathbb{Q})$. These updates correspond to alternatively performing Markovian projections and reciprocal projections.

Combining Proposition 2 and Definition 3, we get that for any $n \in \mathbb{N}$, $\mathbb{P}_0^n = \pi_0$ and $\mathbb{P}_T^n = \pi_T$. This property is in contrast to the IPF algorithm (7) for which the marginals at the initial and final times are *not* preserved. We highlight this duality between IPF (7) and IMF (8) in Table 1.

We conclude this section with a theoretical analysis of IMF. First, we start by showing a Pythagorean theorem for both the Markovian projection and the reciprocal projection.

**Lemma 6.** *Under mild assumptions, if $\mathbb{M} \in \mathcal{M}$, $\Pi \in \mathcal{R}(\mathbb{Q})$ and $\mathrm{KL}(\Pi|\mathbb{M}) < +\infty$, we have*

$$\mathrm{KL}(\Pi|\mathbb{M}) = \mathrm{KL}(\Pi|\mathrm{proj}_{\mathcal{M}}(\Pi)) + \mathrm{KL}(\mathrm{proj}_{\mathcal{M}}(\Pi)|\mathbb{M}).$$

*If $\mathrm{KL}(\mathbb{M}|\Pi) < +\infty$, we have*

$$\mathrm{KL}(\mathbb{M}|\Pi) = \mathrm{KL}(\mathbb{M}|\mathrm{proj}_{\mathcal{R}(\mathbb{Q})}(\mathbb{M})) + \mathrm{KL}(\mathrm{proj}_{\mathcal{R}(\mathbb{Q})}(\mathbb{M})|\Pi).$$

Using Lemma 6, we have the following proposition.

**Proposition 7.** *Under mild assumptions, we have $\mathrm{KL}(\mathbb{P}^{n+1}|\mathbb{P}^{SB}) \leq \mathrm{KL}(\mathbb{P}^n|\mathbb{P}^{SB}) < \infty$, and $\lim_{n \to +\infty} \mathrm{KL}(\mathbb{P}^n|\mathbb{P}^{n+1}) = 0$.*

Hence, for the IMF sequence $(\mathbb{P}^n)_{n \in \mathbb{N}}$, the Markov path measures $(\mathbb{P}^{2n+1})_{n \in \mathbb{N}}$ are getting closer to the reciprocal class, while the reciprocal path measures $(\mathbb{P}^{2n+2})_{n \in \mathbb{N}}$ are getting closer to the set of Markov measures. Proposition 7 should be compared with (Rüschendorf, 1995, Proposition 2.1, Equation (2.16)) which shows that, for the IPF sequence $(\tilde{\mathbb{P}}^n)_{n \in \mathbb{N}}$, we have $\lim_{n \to +\infty} \mathrm{KL}(\tilde{\mathbb{P}}^{n+1}|\tilde{\mathbb{P}}^n) = 0$. This result is similar to Proposition 7 but for the *forward* Kullback–Leibler divergence.

Using Proposition 7, we finally prove the convergence of the IMF sequence $(\mathbb{P}^n)_{n \in \mathbb{N}}$ to the Schrödinger Bridge. This result was first shown in the concurrent work (Peluchetti, 2023, Theorem 2). We present a simpler proof in Appendix C.6.

**Theorem 8.** *Under mild assumptions, the IMF sequence $(\mathbb{P}^n)_{n \in \mathbb{N}}$ admits a unique fixed point $\mathbb{P}^\star = \mathbb{P}^{SB}$, and $\lim_{n \to +\infty} \mathrm{KL}(\mathbb{P}^n|\mathbb{P}^\star) = 0$.*

## 4 Diffusion Schrödinger Bridge Matching

In this section, we present Diffusion Schrödinger Bridge Matching (DSBM), a practical algorithm for solving the SB problem obtained by combining the IMF procedure with Bridge Matching.

**Iterative Markovian Fitting in practice.** IMF alternatively projects on the Markov class $\mathcal{M}$ and the reciprocal class $\mathcal{R}(\mathbb{Q})$. We denote $\mathbb{M}^{n+1} = \mathbb{P}^{2n+1} \in \mathcal{M}$ and $\Pi^n = \mathbb{P}^{2n} \in \mathcal{R}(\mathbb{Q})$. Assuming we know how to sample from the bridge $\mathbb{Q}_{|0,T}$ given the initial and terminal conditions, sampling from the reciprocal projection $\mathrm{proj}_{\mathcal{R}(\mathbb{Q})}(\mathbb{M})$ is simple: First, sample $(\mathbf{X}_0, \mathbf{X}_T)$ from the joint distribution $\mathbb{M}_{0,T}$.[5] Then, sample from the bridge $\mathbb{Q}_{|0,T}(\cdot|\mathbf{X}_0, \mathbf{X}_T)$. The bottleneck of IMF is in the computation of Markovian projections. By Definition 1, $\mathbb{M}^\star = \mathrm{proj}_{\mathcal{M}}(\Pi)$ is associated with the process

$$\mathrm{d}\mathbf{X}_t = \{f_t(\mathbf{X}_t) + \sigma_t^2 \mathbb{E}_{\Pi_{T|t}}[\nabla \log \mathbb{Q}_{T|t}(\mathbf{X}_T|\mathbf{X}_t) \mid \mathbf{X}_t]\}\mathrm{d}t + \sigma_t \mathrm{d}\mathbf{B}_t, \qquad \mathbf{X}_0 \sim \pi_0.$$

By Proposition 2, we can learn $\mathbb{M}^\star$ using $\mathbb{M}^{\theta^\star}$ given by

$$\mathrm{d}\mathbf{X}_t = \{f_t(\mathbf{X}_t) + v_{\theta^\star}(t, \mathbf{X}_t)\}\mathrm{d}t + \sigma_t \mathrm{d}\mathbf{B}_t, \qquad \mathbf{X}_0 \sim \pi_0, \qquad (9)$$

$$\theta^\star = \mathrm{argmin}_\theta\{\textstyle\int_0^T \mathbb{E}_{\Pi_{t,T}}[\|\sigma_t^2 \nabla \log \mathbb{Q}_{T|t}(\mathbf{X}_T|\mathbf{X}_t) - v_\theta(t, \mathbf{X}_t)\|^2]/\sigma_t^2 \mathrm{d}t \; : \; \theta \in \Theta\}, \qquad (10)$$

---

[5]In practice, we sample the SDE associated with $\mathbb{M}$ and save a batch of joint samples $(\mathbf{X}_0, \mathbf{X}_T)$. This is similar to the *trajectory caching* procedure in De Bortoli et al. (2021), but we only retain initial and final samples.

where $\{v_\theta \; : \; \theta \in \Theta\}$ is a parametric family of functions, usually given by a neural network. The optimal $v_{\theta^\star}(t, x_t) = \sigma_t^2 \mathbb{E}_{\Pi_{T|t}}[\nabla \log \mathbb{Q}_{T|t}(\mathbf{X}_T | \mathbf{X}_t) \mid \mathbf{X}_t = x_t]$ for any $t \in [0, T]$ and $x_t \in \mathbb{R}^d$.

With the above two procedures for computing $\mathrm{proj}_{\mathcal{R}(\mathbb{Q})}(\mathbb{M})$ and $\mathrm{proj}_{\mathcal{M}}(\Pi)$, we can now describe a numerical method implementing IMF (8). Let $\Pi^0 = \Pi^0_{0,T} \mathbb{Q}_{|0,T}$ where $\Pi^0_0 = \pi_0, \Pi^0_T = \pi_T$. Learn $\mathbb{M}^1 \approx \mathrm{proj}_{\mathcal{M}}(\Pi^0)$ given by (9) with $v_{\theta^\star}$ given by (10). Next, sample from $\Pi^1 = \mathrm{proj}_{\mathcal{R}(\mathbb{Q})}(\mathbb{M}^1) = \mathbb{M}^1_{0,T} \mathbb{Q}_{|0,T}$ by sampling from $\mathbb{M}^1_{0,T}$ and reconstructing the bridge $\mathbb{Q}_{|0,T}$. We iterate the process to obtain a sequence $(\Pi^n, \mathbb{M}^{n+1})_{n \in \mathbb{N}}$. In practice, this algorithm performs poorly (see Figure 3), since the approximate minimization (10) for computing $\mathbb{M}^{n+1}$ may not admit $\mathbb{M}^{n+1}_T = \pi_T$ exactly as in Proposition 2. Instead, we incur a bias between $\mathbb{M}^{n+1}_T$ and $\pi_T$ which accumulates for each $n \in \mathbb{N}$.

To mitigate this problem, we alternate between a *forward* Markovian projection and a *backward* Markovian projection. This procedure is justified by the following proposition.

**Proposition 9.** *Assume that* $\Pi = \Pi_{0,T} \mathbb{Q}_{|0,T}$ *with* $\mathbb{Q}$ *associated with* $\mathrm{d}\mathbf{X}_t = f_t(\mathbf{X}_t)\mathrm{d}t + \sigma_t \mathrm{d}\mathbf{B}_t$. *Under mild conditions, the Markovian projection* $\mathbb{M}^\star = \mathrm{proj}_{\mathcal{M}}(\Pi)$ *is associated with both*

$$\mathrm{d}\mathbf{X}_t = \{f_t(\mathbf{X}_t) + \sigma_t^2 \mathbb{E}_{\Pi_{T|t}}[\nabla \log \mathbb{Q}_{T|t}(\mathbf{X}_T | \mathbf{X}_t) \mid \mathbf{X}_t]\}\mathrm{d}t + \sigma_t \mathrm{d}\mathbf{B}_t, \quad \mathbf{X}_0 \sim \Pi_0, \tag{11}$$

$$\mathrm{d}\mathbf{Y}_t = \{-f_{T-t}(\mathbf{Y}_t) + \sigma_{T-t}^2 \mathbb{E}_{\Pi_{0|T-t}}[\nabla \log \mathbb{Q}_{T-t|0}(\mathbf{Y}_t | \mathbf{Y}_T) \mid \mathbf{Y}_t]\}\mathrm{d}t + \sigma_{T-t} \mathrm{d}\mathbf{B}_t, \mathbf{Y}_0 \sim \Pi_T. \tag{12}$$

In Proposition 9, (11) is the definition of the Markovian projection, see Definition 1. However, (12) is an equivalent representation as a *time-reversal*. In practice, $(\mathbf{Y}_t)_{t \in [0,T]}$ is approximated with

$$\mathrm{d}\mathbf{Y}_t = \{-f_{T-t}(\mathbf{Y}_t) + v_{\phi^\star}(T - t, \mathbf{Y}_t)\}\mathrm{d}t + \sigma_{T-t} \mathrm{d}\mathbf{B}_t, \qquad \mathbf{Y}_0 \sim \pi_T, \tag{13}$$

$$\phi^\star = \mathrm{argmin}_\phi \{\textstyle\int_0^T \mathbb{E}_{\Pi_{0,t}}[\|\sigma_t^2 \nabla \log \mathbb{Q}_{t|0}(\mathbf{X}_t | \mathbf{X}_0) - v_\phi(t, \mathbf{X}_t)\|^2]/\sigma_t^2 \mathrm{d}t \; : \; \phi \in \Phi\}. \tag{14}$$

The optimal $v_{\phi^\star}(t, x_t) = \sigma_t^2 \mathbb{E}_{\Pi_{0|t}}[\nabla \log \mathbb{Q}_{t|0}(\mathbf{X}_t | \mathbf{X}_0) \mid \mathbf{X}_t = x_t]$ for any $t \in [0, T]$ and $x_t \in \mathbb{R}^d$.

---

**Algorithm 1** Diffusion Schrödinger Bridge Matching

1: **Input:** Joint distribution $\Pi^0_{0,T}$, tractable bridge $\mathbb{Q}_{|0,T}$, number of outer iterations $N \in \mathbb{N}$.
2: Let $\Pi^0 = \Pi^0_{0,T} \mathbb{Q}_{|0,T}$.
3: **for** $n \in \{0, \ldots, N-1\}$ **do**
4:    Learn $v_{\phi^\star}$ using (14) with $\Pi = \Pi^{2n}$.
5:    Let $\mathbb{M}^{2n+1}$ be given by (13).
6:    Let $\Pi^{2n+1} = \mathbb{M}^{2n+1}_{0,T} \mathbb{Q}_{|0,T}$.
7:    Learn $v_{\theta^\star}$ using (10) with $\Pi = \Pi^{2n+1}$.
8:    Let $\mathbb{M}^{2n+2}$ be given by (9).
9:    Let $\Pi^{2n+2} = \mathbb{M}^{2n+2}_{0,T} \mathbb{Q}_{|0,T}$.
10: **end for**
11: **Output:** $v_{\theta^\star}, v_{\phi^\star}$

---

Note that $\mathbf{X}_0 \sim \pi_0$ in the forward projection, while $\mathbf{Y}_0 \sim \pi_T$ in the backward projection. Therefore, using the backward projection removes the bias on $\pi_T$ accumulated from the forward projection. Leveraging the time-symmetry of the Markovian projection and alternating between (13) and (9) yields the DSBM methodology summarized in Algorithm 1.

It is also possible to learn *both* the forward and backward processes at each step, and enforce that the backward and forward processes match. We explore this in Appendix G.

**Initialization coupling.** We now relate Algorithm 1 to the classical IPF and practical algorithms such as DSB (De Bortoli et al., 2021). Instead of initializing DSBM with $\Pi^0_{0,T}$ given by a coupling between $\pi_0, \pi_T$, if we initialize it by $\Pi^0_{0,T} = \mathbb{Q}_{0,T}$ where $\mathbb{Q}_0 = \pi_0$ and $\mathbb{Q}_{T|0}$ is given by the reference process defined in (1), then DSBM also recovers the IPF iterates used in DSB.

**Proposition 10.** *Suppose the families of functions* $\{v_\theta \; : \; \theta \in \Theta\}$ *and* $\{v_\phi \; : \; \phi \in \Phi\}$ *are rich enough so that they can model the optimal vector fields. Let* $(\Pi^n, \mathbb{M}^{n+1})_{n \in \mathbb{N}}$ *be the optimal DSBM sequence in Algorithm 1 initialized with* $\Pi^0_{0,T} = \mathbb{Q}_{0,T}$, *and let* $(\tilde{\mathbb{P}}^n)_{n \in \mathbb{N}}$ *be the optimal DSB sequence given by the IPF iterates in (7). Then for any* $n \in \mathbb{N}, n \geq 1$, *we have* $\mathbb{M}^n = \tilde{\mathbb{P}}^n$.

We will thus call DSBM-IPF, the DSBM algorithm initialized with the joint distribution given by the forward reference process $\Pi^0_{0,T} = \mathbb{Q}_{0,T}$; and DSBM-IMF, the DSBM algorithm initialized with an independent coupling $\Pi^0_{0,T} = \pi_0 \otimes \pi_T$. However, the training procedure of DSBM-IPF is very different from the one of (De Bortoli et al., 2021; Chen et al., 2022). In existing works, $\tilde{\mathbb{P}}^{n+1}$ is

obtained as the time-reversal of $\tilde{\mathbb{P}}^n$ which requires full trajectories from $\tilde{\mathbb{P}}^n$, see e.g. (De Bortoli et al., 2021, Proposition 6). In contrast, in Algorithm 1 we only use the *coupling* $\mathbb{M}_{0,T}^n$ to create the bridge measure $\Pi^n = \mathbb{M}_{0,T}^n \mathbb{Q}_{|0,T}$. By doing so, (i) the losses (10) and (14) can be easily evaluated at any time $t \in [0, T]$; (ii) the *trajectory caching* procedure in DSBM is more computationally and memory efficient; (iii) while every IPF iteration $\tilde{\mathbb{P}}^n$ is also supposed to be in $\mathcal{R}(\mathbb{Q})$, in practice one can observe a *forgetting* of the bridge $\mathbb{Q}_{|0,T}$ (Fernandes et al., 2021). In DSBM, this effect is countered by explicit projections on the reciprocal class. See Appendix F for more details.

**Probability flow ODE.**  At equilibrium of DSBM, we have that $(\mathbf{Y}_t)_{t \in [0,T]}$ given by (13) is the time reversal of $(\mathbf{X}_t)_{t \in [0,T]}$ given by (9) and are both associated with the optimal Schrödinger Bridge path measure $\mathbb{P}^\star$. As a result, we have that $v_{\phi^\star}(t,x) = -v_{\theta^\star}(t,x) + \sigma_t^2 \nabla \log \mathbb{P}_t^\star(x)$. Hence, a probability flow $(\mathbf{Z}_t^\star)_{t \in [0,T]}$ such that $\mathrm{Law}(\mathbf{Z}_t^\star) = \mathbb{P}_t^\star$ for any $t \in [0, T]$ is given by

$$\mathrm{d}\mathbf{Z}_t^\star = \{f_t(\mathbf{Z}_t^\star) + \tfrac{1}{2}[v_{\theta^\star}(t, \mathbf{Z}_t^\star) - v_{\phi^\star}(t, \mathbf{Z}_t^\star)]\}\mathrm{d}t, \qquad \mathbf{Z}_0^\star \sim \pi_0.$$

See also De Bortoli et al. (2021); Chen et al. (2022) for derivation of this result. Note however that the path measure induced by $(\mathbf{Z}_t^\star)_{t \in [0,T]}$ does not correspond to $\mathbb{P}^\star$; in particular, $(\mathbf{Z}_0^\star, \mathbf{Z}_T^\star)$ is *not* an entropic OT plan. However, since for any $t \in [0, T]$, $\mathbf{Z}_t^\star$ has marginal distribution $\mathbb{P}_t^\star$, we can compute the log-likelihood of the model (Song et al., 2021b; Huang et al., 2021).

## 5   Related Work

**Markovian projection and Bridge Matching.**  The concept of Markovian projection has been rediscovered multiple times (Krylov, 1984; Gyöngy, 1986; Dupire, 1994). In the machine learning context, this was first proposed by Peluchetti (2021) to define Bridge Matching models. More recently, Liu et al. (2022b) derived theoretical properties of the Markovian projection in Proposition 2, first part of Lemma 6, and applied Bridge Matching for learning data on discrete and constrained domains.

**Bridge and Flow Matching.**  Flow Matching corresponds to deterministic bridges with deterministic samplers (ODEs) and has been under active study (Liu et al., 2023b; Liu, 2022; Lipman et al., 2023; Albergo and Vanden-Eijnden, 2023; Heitz et al., 2023; Pooladian et al., 2023; Tong et al., 2023). Denoising Diffusion Implicit Models (DDIM) (Song et al., 2021a) can also be formulated as a discrete-time version of Flow Matching, see Liu et al. (2023b). These models have been extended to the Riemannian setting by Chen and Lipman (2023). Recently, Albergo et al. (2023) studied the influence of stochasticity in the bridge, through the concept of stochastic interpolants. Liu et al. (2023a); Delbracio and Milanfar (2023) used Bridge Matching to perform image restoration tasks and noted benefits of stochasticity empirically. Closely related to our work is the Rectified Flow algorithm of Liu et al. (2023b), which corresponds to an iterative Flow Matching procedure in order to improve the straightness of the flow and thus eases its simulation. An iterative rectifying procedure using stochastic interpolants is also proposed in (Albergo et al., 2023, Section 3.5). Our proposed DSBM-IMF algorithm is closest to Rectified Flow, which can be seen as the deterministic limiting case of DSBM-IMF as $\sigma \to 0$. However, there are a few important theoretical and practical differences. Most notably, we adopt the SDE approach which is crucial for the validity of Proposition 5 as well as for the empirical performance of DSBM. We discuss further distinctions between DSBM and Rectified Flow in Appendix A.3.

**Diffusion Schrödinger Bridge.**  Schrödinger Bridges (Schrödinger, 1932) are ubiquitous in probability theory (Léonard, 2014b) and stochastic control (Dai Pra, 1991; Chen et al., 2021). More recently, they have been used for generative modeling: De Bortoli et al. (2021) introduced the DSB algorithm and Vargas et al. (2021); Chen et al. (2022) introduced similar algorithms. The case of Dirac delta terminal distribution was investigated by Wang et al. (2021). These methods were later extended to solve conditional simulation and more general control problems (Shi et al., 2022; Thornton et al., 2022; Liu et al., 2022a; Chen et al., 2023; Tamir et al., 2023). In Somnath et al. (2023), SBs are learned using one Bridge Matching iteration, assuming access to the true Schrödinger static coupling. Our proposed method DSBM-IPF is closest to DSB, but with improved continuous-time training and projections on the reciprocal class which mitigate two limitations of DSB. Concurrently with our work, Peluchetti (2023) independently introduced the DSBM-IMF approach (named IDBM therein).

## 6 Experiments

**2D Experiments.** We first show our proposed methods can generate correct samples and learn lower kinetic energy transport maps in some 2D examples. We compare our method DSBM with flow-based methods including Flow Matching (FM) (Lipman et al., 2023), Conditional Flow Matching (CFM), OT-CFM (Tong et al., 2023), and Rectified Flow (RF) (Liu et al., 2023b); and other SB methods including DSB (De Bortoli et al., 2021) and SB-CFM (Tong et al., 2023). OT-CFM and SB-CFM utilizes sample-based mini-batch OT or EOT solvers (Fatras et al., 2021; Flamary et al., 2021) to define an approximate OT or SB static coupling $\tilde{\Pi}_{0,T}^{\text{OT}}$ or $\tilde{\Pi}_{0,T}^{\text{SB}}$, see also Pooladian et al. (2023); Stromme (2023). We can also utilize this idea in the DSBM-IMF framework, which corresponds to using the initialization coupling $\Pi_{0,T}^0 = \tilde{\Pi}_{0,T}^{\text{SB}}$ in Algorithm 1. This approximate SB coupling $\tilde{\Pi}_{0,T}^{\text{SB}}$ also satisfies $\tilde{\Pi}_0^{\text{SB}} = \pi_0$ and $\tilde{\Pi}_T^{\text{SB}} = \pi_T$ but can provide a better initialization than the independent coupling $\Pi_{0,T}^0 = \pi_0 \otimes \pi_T$. We name this approach DSBM-IMF+. The rest of the methods do not use OT solvers. DSB and DSBM directly learn the EOT map as the solution of the diffusion process.

In Table 2, we show the 2-Wasserstein distance between the true and generated samples, as well as the integrated path energy defined as $\mathbb{E}[\int_0^T \|v(t, \mathbf{Z}_t)\|^2 dt]$ where $v$ is the learned drift along the ODE trajectory $\mathbf{Z}_t$. For direct comparability, we report for DSBM using its probability flow ODE. Lower path energies represent shorter (and potentially easier to integrate) trajectories. We find that in this low dimensional setting, OT-CFM performs the best by utilizing OT solvers, but DSBM outperforms FM and CFM when OT solvers are not used. Further, DSBM outperforms DSB on all datasets, suggesting DSBM solves the SB problem with higher accuracy. The results also show that among SB methods, DSBM-IMF+ can achieve lower sampling error than DSBM-IPF and DSBM-IMF. It also performs better than SB-CFM on 3 of the datasets and achieve lower path energy on all datasets. Finally, we find Rectified Flow achieves lower sampling error than DSBM except for the *moons-8gaussians* task, for which DSBM is significantly more accurate. Since RF can be informally seen as DSBM in the case $\sigma \to 0$, this suggests the optimal $\sigma$ varies for each task and between generative and general transfer tasks. Figure 2 visualizes how $\sigma$ affects the straightness and sample quality of learned transport maps between two mixture distributions.

**High-Dimensional Gaussian Experiment.** We next perform the Gaussian transport experiment in De Bortoli et al. (2021) with dimension $d = 50$ to verify the scalability of our proposed approach. The true SB can be computed analytically in this case (Bunne et al., 2023). In Figure 3, we plot the convergence of the learned mean $\mathbb{E}[\mathbf{X}_0]$, variance $\text{Var}(\mathbf{X}_0)$, and covariance $\text{Cov}(\mathbf{X}_0, \mathbf{X}_T)$ between times $0, T$. We also consider RF and a related baseline IMF-b, which performs IMF numerically but only in the backward direction. All methods converge approximately to the correct mean. However,

| | *2-Wasserstein (Euler 20 steps)* | | | | *Path energy* | | | |
|---|---|---|---|---|---|---|---|---|
| *Dataset* | moons | scurve | 8gaussians | moons-8gaussians | moons | scurve | 8gaussians | moons-8gaussians |
| DSBM-IPF | 0.140±0.006 | 0.140±0.024 | 0.315±0.079 | *0.812±0.092* | 1.598±0.034 | *2.110±0.059* | 14.91±0.310 | 42.16±1.026 |
| DSBM-IMF | 0.144±0.024 | 0.145±0.037 | 0.338±0.091 | 0.838±0.098 | **1.580±0.036** | **2.092±0.053** | **14.81±0.255** | **41.00±1.495** |
| DSBM-IMF+ | **0.123±0.014** | **0.130±0.025** | *0.276±0.030* | **0.802±0.172** | *1.594±0.043* | 2.116±0.018 | *14.88±0.252* | *41.09±1.206* |
| DSB | 0.190±0.049 | 0.272±0.065 | 0.411±0.084 | 0.987±0.324 | - | - | - | - |
| SB-CFM | *0.129±0.024* | *0.136±0.030* | **0.238±0.044** | 0.843±0.079 | 1.649±0.035 | 2.144±0.044 | 15.08±0.209 | 45.69±0.661 |
| FM | 0.212±0.025 | 0.161±0.033 | 0.351±0.066 | - | 2.227±0.056 | 2.950±0.074 | 18.12±0.416 | - |
| CFM | 0.215±0.028 | 0.171±0.023 | 0.370±0.049 | *1.285±0.314* | 2.391±0.043 | 3.071±0.026 | 18.00±0.090 | 116.5±2.633 |
| RF | *0.129±0.022* | *0.126±0.019* | *0.267±0.041* | 1.522±0.304 | *1.185±0.052* | *1.633±0.074* | **14.84±0.441** | *37.61±3.906* |
| OT-CFM | **0.111±0.005** | **0.102±0.013** | 0.253±0.040 | **0.716±0.187** | **1.178±0.020** | **1.577±0.036** | *15.10±0.215* | **30.50±0.626** |

Table 2: Sampling quality as measured by 2-Wasserstein distance and path energy for the 2D experiments. ±1 SD over 5 seeds. Best values are in bold and second best are italicized.

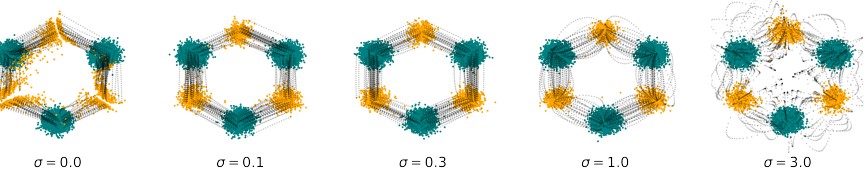

$\sigma = 0.0$     $\sigma = 0.1$     $\sigma = 0.3$     $\sigma = 1.0$     $\sigma = 3.0$

Figure 2: Learned SB probability flow between two mixtures of Gaussians (green → yellow).

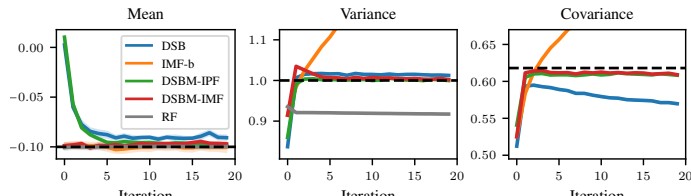

Figure 3: Convergence of Gaussian experiment in $d = 50$.

| KL $\times 10^{-3}$ | $d = 5$ | $d = 20$ | $d = 50$ |
|---|---|---|---|
| DSB | 3.26±1.60 | 13.0±3.49 | 32.8±1.28 |
| SB-CFM | 1.45±0.73 | 12.3±1.47 | 49.4±3.91 |
| DSBM-IPF | **1.23±0.23** | **4.42±0.76** | **8.75±0.87** |
| DSBM-IMF | *1.34±0.51* | *5.05±0.95* | *9.76±1.67* |

Table 3: Average $\mathrm{KL}(\mathbb{P}_t | \mathbb{P}_t^{\mathrm{SB}})$ at 21 uniformly spaced $t$.

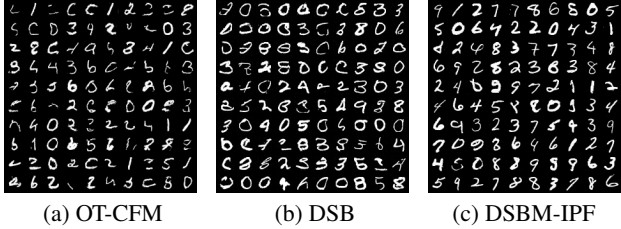

(a) OT-CFM     (b) DSB     (c) DSBM-IPF

Figure 4: Samples of MNIST digits transferred from letters.

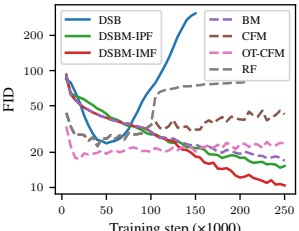

Figure 5: FID vs iteration.

the variance estimates become inaccurate for RF and IMF-b. Among SB methods, DSB and IMF-b also gave inaccurate SB covariance estimates as the number of iteration increases. On the other hand, DSBM does not suffer from this issue. In Table 3, we further quantify the accuracy and compare with SB-CFM (Tong et al., 2023) by computing the KL divergence between the marginal distributions of the learned process $\mathbb{P}_t$ and the true SB $\mathbb{P}_t^{\mathrm{SB}}$. Our proposed methods achieve similar KL divergence as SB-CFM in dimension $d = 5$, but are much more accurate in higher dimensions.

**MNIST, EMNIST transfer.** We test our method for domain transfer between MNIST digits and EMNIST letters as in De Bortoli et al. (2021). We compare DSBM as a direct substitute of DSB, and also with Bridge Matching (BM) (Peluchetti, 2021; Liu et al., 2022b), CFM, OT-CFM and RF. We plot some output samples from different algorithms in Figure 4 and the convergence of FID score in Figure 5. We find that OT-CFM becomes less applicable in higher dimensions and produces samples of worse quality (Figure 4a). On the other hand, image quality deteriorates during training of DSB and RF. DSBM achieves higher quality samples visually, and does not suffer from deterioration. It is also about 30% more efficient than DSB in terms of runtime.

**CelebA transfer.** Next, we evaluate and perform some ablations of our method on a transfer task on the CelebA $64 \times 64$ dataset. We consider the images given by the tokens male/old and female/young. In Figures 6 and 7, we show that as $\sigma$ increases, the quality of the images (as measured by the FID score) increases until $\sigma$ is too high, but the alignment (as measured by LPIPS) between the generated image and the original sample decreases. Additionally, we investigate the dependency between $\sigma$ and image dimension in Figure 8. In particular, for the same $\sigma = 1$, the outputs of DSBM for CelebA $128 \times 128$ are better aligned with the original data than for CelebA $64 \times 64$. This is in agreement with the observations of Chen (2023); Hoogeboom et al. (2023) that the *noise schedule* in diffusion models should scale with the resolution.

**AFHQ transfer.** We demonstrate the scalability of our method on an additional transfer experiment on the AFHQ $512 \times 512$ dataset between the classes cat and wild. The results are shown in Figure 9. On this higher-dimensional problem, we observe that DSBM can also generate realistic samples which are similar to the input.

**Unpaired Fluid Flows Downscaling.** Finally, we apply DSBM to perform downscaling of geophysical fluid dynamics, i.e. super-resolution of low resolution spatial data. We use the dataset in (Bischoff and Deck, 2023), which consists of unpaired low ($64 \times 64$) and high ($512 \times 512$) resolution fields. As shown in Figure 10, DSBM is able to learn high resolution reconstructions by only slightly noising the low resolution input. In contrast, Bischoff and Deck (2023) use two diffusion models in forward and backward directions (Diffusion-fb) based on Meng et al. (2022), which improves over the Random baseline. Figure 11 shows that DSBM-IPF and DSBM-IMF achieve much lower $\ell_2$

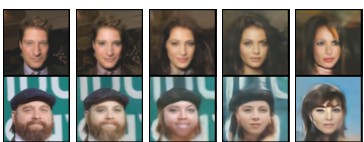
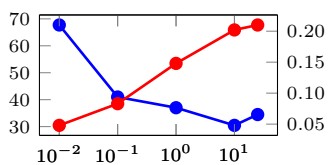
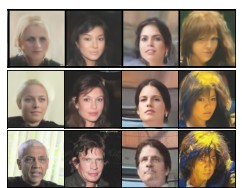

Figure 6: Left to right: initial and generated samples ($64 \times 64$) obtained after 20 DSBM-IMF iterations for $\sigma^2 \in \{0.01, 0.1, 1, 10\}$.

Figure 7: FID (blue) and LPIPS (red) scores (lower is better for both) as we vary $\sigma^2$.

Figure 8: Top to bottom: DSBM ($\sigma = 1$) $64 \times 64$; $128 \times 128$; original images.

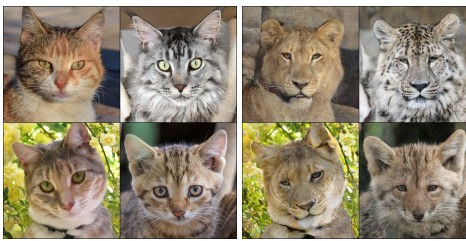
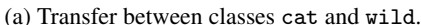
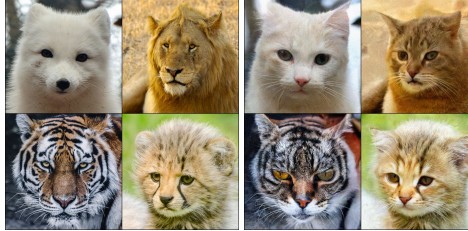

(a) Transfer between classes cat and wild.

(b) Transfer between classes wild and cat.

Figure 9: DSBM domain transfer results on the AFHQ $512 \times 512$ dataset.

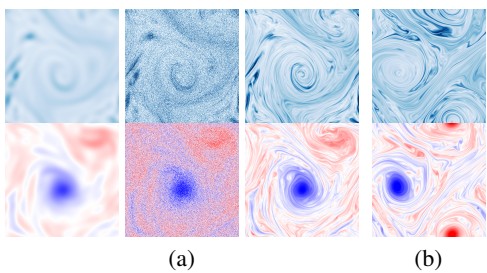
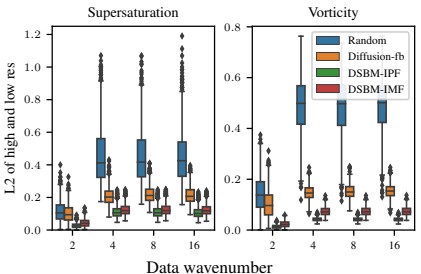

(a)      (b)

Figure 10: (a) Left to right: source low resolution sample, intermediate state and final reconstruction of DSBM-IPF; (b) an unpaired high resolution sample.

Figure 11: $\ell_2$ distance between low resolution source and high resolution reconstructed fields.

distances for all frequency classes in the dataset than Diffusion-fb (and thus Random), indicating DSBM is able to reconstruct high resolution fields consistent with the low resolution source.

# 7 Discussion

In this work, we introduce IMF, a new methodology for learning Schrödinger Bridges. IMF is an alternative to the classical IPF and can be interpreted as its dual. Building on this new framework, we present two practical algorithms, DSBM-IPF and DSBM-IMF, for learning SBs. These algorithms mitigate the time-discretization and bias accumulation issues of existing methods. However, DSBM still has some limitations. First, our results suggest DSBM is most effective for solving general transport problems. For generative modeling, we only find minor improvements compared to Bridge and Flow Matching on CIFAR-10 (see Appendix I.6). Second, while DSBM is more efficient than DSB, it still requires sampling from the learned process during the caching step. Finally, the EOT problem becomes more difficult to solve numerically for small values of $\sigma$.

In future work, we would like to further investigate the differences between DSBM-IMF and DSBM-IPF. IMF also appears useful for developing a better understanding of the Rectified Flow algorithm (Liu et al., 2023b), as IMF minimizes a clear objective (6) and Rectified Flow can be seen as a limiting case of it. Finally, Rectified Flow has also been extended to solve OT problems with general convex costs by Liu (2022), and it would be interesting to derive a SB version of this extension.

## Acknowledgements

YS acknowledges support from the Huawei UK Fellowship. AC acknowledges support from the EPSRC CDT in Modern Statistics and Statistical Machine Learning (EP/S023151/1). AD acknowledges support of the UK Dstl and EPSRC grant EP/R013616/1. This is part of the collaboration between US DOD, UK MOD and UK EPSRC under the Multidisciplinary University Research Initiative. He also acknowledges support from the EPSRC grants CoSines (EP/R034710/1) and Bayes4Health (EP/R018561/1).

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

## Outline of the Appendix

In Appendix A, we first clarify the relationship between different methods in the existing literature and our proposed DSBM framework. In Appendix B, we focus on the family of linear SDEs, and draw a link between the parameterization of bridges in this paper and the stochastic interpolant in Albergo et al. (2023). In Appendix C, we give proofs for results in the main text. In Appendix D, we present additional theoretical results for IMF in the Gaussian case. In Appendix E, we derive the discrete-time version of Markovian projection. In Appendix F, we explain the benefits of DSBM compared to DSB in more detail. In Appendix G, we describe a method for learning the forward and backward processes jointly and propose a consistency loss between the forward and backward processes. In Appendix H, we present additional methodological details for a practical scaling of the loss function to reduce variance, similar to standard Denoising Diffusion Models. In Appendix I, we give further details for all experiments and additional experimental results. Finally, we discuss broader impacts of our work in Appendix J.

## A Discussion of Existing Works

### A.1 Bridge Matching and Flow Matching Models

In this section, we clarify the relationship between variants of Flow Matching and show that they are equivalent under some conditions. We follow the nomenclature of Tong et al. (2023). We refer to the algorithm originally proposed in Lipman et al. (2023) using linear probability paths and described in (Tong et al., 2023, Section 4.1) as Flow Matching (FM), and the algorithm proposed in (Tong et al., 2023, Section 4.2) as Conditional Flow Matching (CFM). There is a small constant parameter $\sigma_{\min}$ in both algorithms, which controls the smoothing of the modeled distribution. We consider the case $\sigma_{\min} = 0$. Then CFM recovers exactly the 1st iteration of Rectified Flow (Liu et al., 2023b). Furthermore, FM, CFM and the 1st iteration of Rectified Flow are all equivalent when performing generative modeling with a standard Gaussian $\pi_0$. We refer to them collectively as Flow Matching models (FMMs) as they only differ in the smoothing method. We also present them all under the Bridge Matching framework. These models can also be interpreted in the context of *stochastic interpolants* (Albergo and Vanden-Eijnden, 2023; Albergo et al., 2023). Finally, we present recent applications of Bridge Matching and show that some of the objectives in Somnath et al. (2023); Liu et al. (2023a); Delbracio and Milanfar (2023) are identical.

**Flow Matching and Conditional Flow Matching.** In Flow Matching (FM), the objective (Lipman et al., 2023, Equation (21)) is

$$\mathbb{E}_{\Pi_{t,T}}[\|(\mathbf{X}_T - \mathbf{X}_t)/(T - t) - v_\theta(t, \mathbf{X}_t)\|^2],$$

where $\Pi_{t,T}$ is given by $\pi_T(\mathbf{X}_T)\mathrm{N}(\mathbf{X}_t; \frac{t}{T}\mathbf{X}_T, (1 - \frac{t}{T})^2)$.

In Conditional Flow Matching (CFM), $\mathbf{X}_t^{0,T} = \frac{t}{T}\mathbf{X}_T + (1 - \frac{t}{T})\mathbf{X}_0$, with $\mathbf{X}_0 \sim \mathrm{N}(0, \mathrm{Id})$ and the objective (Tong et al., 2023, Equation (16)) is given by

$$\mathbb{E}_{\Pi_{0,T}}[\|(\mathbf{X}_T - \mathbf{X}_0)/T - v_\theta(t, \mathbf{X}_t^{0,T})\|^2]. \tag{15}$$

This is the same as (Liu et al., 2023b, Equation (1)). Furthermore, $(\mathbf{X}_T - \mathbf{X}_0)/T = (1 - \frac{t}{T})(\mathbf{X}_T - \mathbf{X}_0)/(T - t) = (\mathbf{X}_T - \mathbf{X}_t^{0,T})/(T - t)$, so the CFM objective is equivalent to

$$\mathbb{E}_{\Pi_{t,T}}[\|(\mathbf{X}_T - \mathbf{X}_t^{0,T})/(T - t) - v_\theta(t, \mathbf{X}_t^{0,T})\|^2]. \tag{16}$$

The optimal $v_\theta(t, x_t) = (\mathbb{E}_{\Pi_{T|t}}[\mathbf{X}_T \mid \mathbf{X}_t = x_t] - x_t)/(T - t)$. In the case of generative modeling, $\pi_0$ is a standard Gaussian distribution and $\Pi_{0,T}$ is given by $\mathrm{N}(\mathbf{X}_0; 0, \mathrm{Id})\pi_T(\mathbf{X}_T)$. Thus, $\Pi_{t,T}$ is also given by $\pi_T(\mathbf{X}_T)\mathrm{N}(\mathbf{X}_t^{0,T}; \frac{t}{T}\mathbf{X}_T, (1 - \frac{t}{T})^2)$. Therefore, the FM (Lipman et al., 2023) and CFM (Tong et al., 2023) objectives are exactly the same. However, CFM is also applicable when $\pi_0$ is not Gaussian distributed, so CFM is a generalized version of FM[6].

---

[6]In the case $\sigma_{\min} > 0$, FM and CFM are indeed different in how smoothing is performed, and we refer to Tong et al. (2023) for a more detailed analysis.

**Stochastic Interpolant.** In (Albergo and Vanden-Eijnden, 2023; Albergo et al., 2023), the concept of stochastic interpolant is introduced. In Albergo and Vanden-Eijnden (2023), the interpolation is deterministic (not necessarily linear), of the form $I_t(x_0, x_T) = \alpha(t)x_0 + \beta(t)x_T$, while in Albergo et al. (2023), the interpolation is stochastic given by $I_t(x_0, x_T) = \alpha(t)x_0 + \beta(t)x_T + \gamma(t)\mathbf{Z}$ for $\mathbf{Z} \sim \mathrm{N}(0, \mathrm{Id})$. In Albergo and Vanden-Eijnden (2023), an ODE is learned and the associated velocity field $v_\theta$ is obtained by minimizing the following objective (Albergo and Vanden-Eijnden, 2023, Equation (9))

$$\mathbb{E}_{\Pi_{0,T}}[\|\partial_t I_t(\mathbf{X}_0, \mathbf{X}_T) - v_\theta(t, \mathbf{X}_t^{0,T})\|^2].$$

Hence, if $I_t(x_0, x_T) = \frac{t}{T}x_0 + (1 - \frac{t}{T})x_T$, we recover (15).

**Link with Bridge Matching.** When $\mathbb{Q}$ is associated with the Brownian motion $(\sigma\mathbf{B}_t)_{t \in [0,T]}$ and $\sigma \to 0$ in Bridge Matching, we recover the same objective (5) as the Flow Matching objective (16), since $\nabla \log \mathbb{Q}_{T|t}(\mathbf{X}_T|\mathbf{X}_t) = (\mathbf{X}_T - \mathbf{X}_t)/(\sigma^2(T - t))$. Bridge Matching can also be applied to general distributions $\pi_0, \pi_T$; i.e. $\pi_0$ does not have to be restricted to a Gaussian. Therefore, Bridge Matching is a generalized version of Flow Matching, see also (Liu et al., 2022b, Equation (10)).

**Inverse problems and interpolation.** Somnath et al. (2023) and Liu et al. (2023a) present Bridge Matching algorithms between aligned data $(\mathbf{X}_0, \mathbf{X}_T) \sim \Pi_{0,T}$. The objectives (Somnath et al., 2023, Equation (8)) and (Liu et al., 2023a, Equation (12)) are equivalent to the Bridge Matching objective (3). The main difference between Liu et al. (2023a) and Somnath et al. (2023) resides in the choice of $\Pi_{0,T}$. In the case of Somnath et al. (2023), this choice is motivated by the access to *aligned data* with applications in biology assuming they are distributed as the true Schrödinger static coupling, i.e. $\Pi_{0,T} = \Pi_{0,T}^{\mathrm{SB}}$. In the case of Liu et al. (2023a), $\Pi_{0,T}$ corresponds to a pairing between clean and corrupted images, e.g. with $\Pi_0 = \pi_0$ the distribution of clean images and $\Pi_T = \pi_T$ the distribution of corrupted images obtained from the clean images using the degradation kernel $\Pi_{T|0}$.

Finally, in (Delbracio and Milanfar, 2023, Equation (5)) the authors consider a reconstruction process of the form

$$\mathrm{d}\mathbf{X}_t = (\mathbb{E}_{\Pi_{0|t}}[\mathbf{X}_0 \mid \mathbf{X}_t] - \mathbf{X}_t)/t\,\mathrm{d}t, \qquad \mathbf{X}_T \sim \Pi_T, \tag{17}$$

where here we have replaced $F(x_t, t)$ by $\mathbb{E}_{\Pi_{0|t}}[\mathbf{X}_0 \mid \mathbf{X}_t = x_t]$. This is justified if the $\|\cdot\|_p$ norm in (Delbracio and Milanfar, 2023, Equation (4)) is replaced by $\|\cdot\|_2^2$ (or any Bregman Loss Function, see Banerjee et al. (2005)). In Delbracio and Milanfar (2023), $\Pi_{0,T}$ corresponds to the joint distribution of clean and corrupted images as in Liu et al. (2023a). Exchanging the role of $\Pi_0$ and $\Pi_T$, (17) can be rewritten equivalently as

$$\mathrm{d}\mathbf{X}_t = (\mathbb{E}_{\Pi_{T|t}}[\mathbf{X}_T \mid \mathbf{X}_t] - \mathbf{X}_t)/(T - t)\,\mathrm{d}t, \qquad \mathbf{X}_0 \sim \Pi_0.$$

We thus obtain the optimal Flow Matching vector field $v_\theta(t, x_t) = (\mathbb{E}_{\Pi_{T|t}}[\mathbf{X}_T \mid \mathbf{X}_t = x_t] - x_t)/(T - t)$ in (16). Note that Delbracio and Milanfar (2023) also incorporates a stochastic version of their objective (Delbracio and Milanfar, 2023, Equation (7)). It remains an open question whether this objective can be understood as a special instance of the Bridge Matching framework.

## A.2  On DSBM and Existing Works

In this section, we show that the DSBM framework recovers the above existing algorithms for different choices of bridges $\mathbb{Q}_{|0,T}$ and couplings $\Pi_{0,T}^0$ in Algorithm 1. For the independent coupling $\Pi_{0,T}^0 = \pi_0 \otimes \pi_T$ and Brownian bridge $\mathbb{Q}_{|0,T}$ (4) with diffusion parameter $\sigma_t = \sigma$, the loss function (10) recovers the Brownian Bridge Matching loss (5). Letting $\sigma \to 0$, we recover Flow Matching (Lipman et al., 2023). In this case, further iterations repeating lines 7-9 in Algorithm 1 (with only forward projections) recover Rectified Flow (Liu et al., 2023b). If the coupling $\Pi_{0,T}^0$ is given by an estimation of the OT map between $\pi_0$ and $\pi_T$, then the first iteration recovers OT-CFM (Tong et al., 2023; Pooladian et al., 2023). Finally, for general bridges $\mathbb{Q}_{|0,T}$, if we are given the optimal Schrödinger Bridge static coupling $\Pi_{0,T}^0 = \Pi_{0,T}^{\mathrm{SB}}$, then the DSBM procedure converges in one iteration and we recover Somnath et al. (2023).

## A.3  DSBM and Rectified Flow

We discuss the differences in more detail between our proposed DSBM method and Rectified Flow. Both Rectified Flow and DSBM are general frameworks for building transport maps between

two general distributions $\pi_0, \pi_T$. However, there are a few important theoretical and practical differences. Firstly, we adopt the SDE approach as opposed to the ODE approach in Rectified Flow. This distinction is crucial in theory, as Proposition 5, which guarantees the uniqueness of the characterization of SB, is valid only when $\sigma_t > 0$. Consequently, Rectified Flow is not guaranteed to converge to the dynamic optimal transport solution (see e.g. counterexample in Liu (2022)). In a following work, Liu (2022) established formal connections between Rectified Flow and OT when restricting the class of vector fields to gradient fields. In DSBM, the connection to OT is obtained by considering its entropy-regularized version. Furthermore, by adopting the SDE approach, we observe significant improvements of sample quality in our experiments when performing transport between two general distributions. This is in line with the theoretical analysis in Albergo et al. (2023). On the other hand, while Bridge Matching also achieves high sample quality using the SDE approach, the transported samples are much more dissimilar to the input data (see e.g. Figures 14, 15, 19). Lastly, Rectified Flow also performs Markovian projections iteratively, but only in the forward direction. Consequently, the bias in the learned marginals $\mathbb{P}_T^n$ is accumulated and cannot be corrected in later iterations, i.e. the first iteration of RF will achieve the most accurate marginal $\mathbb{P}_T^1$. Subsequent iterations can improve the straightness of the flow, but at the cost of sampling accuracy of $\mathbb{P}_T^n$. We observe in practice that this becomes particularly problematic if the first iteration of Rectified Flow (which is equivalent to CFM) fails to provide a good transport and learn an accurate $\mathbb{P}_T^1$, e.g. in the case of *moons-8gaussians* (Table 2), Gaussian transport (Figure 3), and MNIST, EMNIST transfer (Figure 5 and Figure 13). As Rectified Flow cannot recover from this issue, we observe the accuracy of $\mathbb{P}_T^n$ only deteriorates in further iterations as $n$ increases. In our methodology, we leverage Proposition 9 to perform forward and backward Bridge Matching, and we observe that the marginal accuracy is able to improve with iteration.

# B The Design Space of Brownian Bridges

## B.1 Relationship to Stochastic Interpolants

**From stochastic interpolants to Brownian bridges.** In this section, we draw a link between our parameterization of bridges and the one used in Albergo et al. (2023). In Albergo et al. (2023), a stochastic interpolant is defined as

$$\mathbf{X}_t = \bar{\alpha}_t x_0 + \bar{\beta}_t x_T + \bar{\gamma}_t \mathbf{Z}, \tag{18}$$

where $\mathbf{Z} \sim \mathrm{N}(0, \mathrm{Id})$. Since their methodology and analysis mainly relies on the probability flow, they work with (18), which is easier to analyse. In our setting, as we deal mostly with diffusions, it is natural to parameterize Brownian bridges as follows

$$d\mathbf{X}_t = \{-\alpha_t \mathbf{X}_t + \beta_t x_T\} dt + \gamma_t d\mathbf{B}_t. \tag{19}$$

The goal of this section is to derive explicit formulas between the parameters $\bar{\alpha}_t, \bar{\beta}_t$ and $\bar{\gamma}_t$ of (18) and the parameters $\alpha_t, \beta_t$ and $\gamma_t$ of (19). Consider $(\mathbf{X}_t)_{t \in [0,T]}$ given by (19). We have that for any $t \in [0,T]$

$$\mathbf{X}_t = \exp[-A_t] x_0 + \int_0^t \beta_s \exp[A_s - A_t] ds\, x_T + \int_0^t \gamma_s \exp[A_s - A_t] d\mathbf{B}_s,$$

where $A_t = \int_0^t \alpha_s ds$. Therefore, we have that

$$\bar{\alpha}_t = \exp[-\int_0^t \alpha_s ds], \qquad \bar{\beta}_t = \int_0^t \beta_s \exp[-\int_s^t \alpha_u du] ds, \qquad \bar{\gamma}_t^2 = \int_0^t \gamma_s^2 \exp[-2\int_s^t \alpha_u du] ds, \tag{20}$$

$$\alpha_t = -\frac{\bar{\alpha}_t'}{\bar{\alpha}_t}, \qquad \beta_t = \bar{\beta}_t' + \bar{\beta}_t \alpha_t, \qquad \gamma_t^2 = (\bar{\gamma}_t^2)' + 2\bar{\gamma}_t^2 \alpha_t = 2\bar{\gamma}_t \bar{\gamma}_t' + 2\bar{\gamma}_t^2 \alpha_t. \tag{21}$$

Using this relationship, we get that the Markovian projection, see Definition 1, is given by

$$d\mathbf{X}_t^\star = f_t^\star(\mathbf{X}_t) dt + \gamma_t d\mathbf{B}_t, \qquad f_t^\star(x_t) = \mathbb{E}_{\Pi_{T|t}}[-\alpha_t \mathbf{X}_t + \beta_t \mathbf{X}_T \mid \mathbf{X}_t = x_t].$$

We have that

$$\begin{aligned} f_t^\star(x_t) &= \mathbb{E}_{\Pi_{T|t}}[-\alpha_t \mathbf{X}_t + \beta_t \mathbf{X}_T \mid \mathbf{X}_t = x_t] \\ &= \mathbb{E}_{\Pi_{0,T|t}}[-\alpha_t(\bar{\alpha}_t \mathbf{X}_0 + \bar{\beta}_t \mathbf{X}_T + \bar{\gamma}_t \mathbf{Z}) + \beta_t \mathbf{X}_T \mid \mathbf{X}_t = x_t]. \end{aligned}$$

Using (21), we get that

$$f_t^\star(x_t) = \mathbb{E}_{\Pi_{0,T|t}}[\bar{\alpha}_t' \mathbf{X}_0 + \bar{\beta}_t' \mathbf{X}_T + \tfrac{\bar{\alpha}_t' \bar{\gamma}_t}{\bar{\alpha}_t} \mathbf{Z} \mid \mathbf{X}_t = x_t].$$

In Albergo et al. (2023), it is shown that $\nabla \log \mathbb{M}_t^\star(x_t) = -\mathbb{E}_{\Pi_{0,T|t}}[\mathbf{Z} \mid \mathbf{X}_t = x_t]/\bar{\gamma}_t$, where $\mathbb{M}^\star$ is the Markovian projection. The probability flow associated with $(\mathbf{X}_t^\star)_{t \in [0,T]}$ is given by

$$\begin{aligned}
\mathrm{d}\mathbf{Z}_t^\star &= \{f_t^\star(\mathbf{Z}_t^\star) - \tfrac{\gamma_t^2}{2}\nabla \log \mathbb{M}_t^\star(\mathbf{Z}_t^\star)\}\mathrm{d}t \\
&= \{\mathbb{E}_{\Pi_{0,T|t}}[\bar{\alpha}_t' \mathbf{X}_0 + \bar{\beta}_t' \mathbf{X}_T + (-\alpha_t \bar{\gamma}_t + \tfrac{\gamma_t^2}{2\bar{\gamma}_t})\mathbf{Z} \mid \mathbf{X}_t = \mathbf{Z}_t^\star]\}\mathrm{d}t \\
&= \{\mathbb{E}_{\Pi_{0,T|t}}[\bar{\alpha}_t' \mathbf{X}_0 + \bar{\beta}_t' \mathbf{X}_T + \bar{\gamma}_t' \mathbf{Z} \mid \mathbf{X}_t = \mathbf{Z}_t^\star]\}\mathrm{d}t.
\end{aligned}$$

Hence, we recover (Albergo et al., 2023, Theorem 2.6).

**Non-Markov path measures.** A natural question is whether (19) arises as the bridge measure of some *Markov* measure. For instance, if $\mathbb{Q}$ is associated with $(x_0 + \mathbf{B}_t)_{t \in [0,T]}$, then pinning the process at $x_T$ at time $T$, we get that the associated bridge measure $\mathbb{Q}_{|0,T}$ is given by

$$\mathrm{d}\mathbf{X}_t^{0,T} = (x_T - \mathbf{X}_t)/(T-t)\mathrm{d}t + \mathrm{d}\mathbf{B}_t.$$

Therefore, we recover (19) with $\alpha_t = \beta_t = \frac{1}{T-t}$ and $\gamma_t = 1$. Using (20), we get that $\bar{\alpha}_t = 1 - \frac{t}{T}$, $\bar{\beta}_t = \frac{t}{T}$ and $\bar{\gamma}_t^2 = (T-t)t/T$. We recover (4), upon noting that $\mathbf{B}_t - \frac{t}{T}\mathbf{B}_T$ is Gaussian with zero mean and variance $(T-t)t/T$.

More generally, we consider a Markov measure $\mathbb{Q}$ associated with $(\mathbf{X}_t)_{t \in [0,T]}$ such that

$$\mathrm{d}\mathbf{X}_t = -a_t \mathbf{X}_t \mathrm{d}t + c_t \mathrm{d}\mathbf{B}_t, \qquad \mathbf{X}_0 = x_0.$$

We now derive the associated bridge measure $\mathbb{Q}_{|0,T}$:

$$\mathbf{X}_T = \exp[-\Lambda_T + \Lambda_t]\mathbf{X}_t + \int_t^T c_s \exp[\Lambda_s - \Lambda_T]\mathrm{d}\mathbf{B}_s,$$

with $\Lambda_t = \int_0^t a_s \mathrm{d}s$. We have that

$$\begin{aligned}
c_t^2 \nabla_{x_t} \log \mathbb{Q}_{T|t}(x_T|x_t) = {} & (c_t^2 \exp[\Lambda_t - \Lambda_T]/\int_t^T c_s^2 \exp[2(\Lambda_s - \Lambda_T)]\mathrm{d}s)x_T \\
& - (c_t^2 \exp[2(\Lambda_t - \Lambda_T)]/\int_t^T c_s^2 \exp[2(\Lambda_s - \Lambda_T)]\mathrm{d}s)x_t.
\end{aligned}$$

Therefore, combining this result and (2), we get that $\mathbb{Q}_{|0,T}$ is associated with

$$\begin{aligned}
\alpha_t &= a_t + c_t^2 \exp[-2\int_t^T a_s \mathrm{d}s]/\int_t^T c_s^2 \exp[-2\int_s^T a_u \mathrm{d}u]\mathrm{d}s, \\
\beta_t &= c_t^2 \exp[-\int_t^T a_s \mathrm{d}s]/\int_t^T c_s^2 \exp[-2\int_s^T a_u \mathrm{d}u]\mathrm{d}s, \qquad \gamma_t = c_t.
\end{aligned}$$

In that case $(a_t, c_t)_{t \in [0,T]}$ entirely parameterize $(\alpha_t, \beta_t, \gamma_t)_{t \in [0,T]}$. Hence, in the Ornstein-Uhlenbeck setting, if $\mathbb{Q}_{|0,T}$ is the bridge of a Markov measure, it is fully parameterized by two functions while in the non-Markov setting it is parameterized by three functions.

In this paper, we present our framework in the Markovian setting as the Schrödinger Bridge problem is usually defined with respect to Markov reference measures. However, our methodology could be extended in a straightforward fashion to the non-Markovian setting. This would allow for a further exploration of the design space of DSBM.

## B.2  Linear SDE and Bridge Matching

In this section, we study further the diffusion bridge of linear SDEs. Arbitrary Markov measures can be chosen to build bridges; however, we want to be able to compute some representations of the bridge in an explicit way. More precisely, denoting $(\mathbf{X}_t^{0,T})_{t \in [0,T]}$ the diffusion bridge with $x_0, x_T$ the initial and final condition, we want to have access to the following:

- *integral sampler*: we want to have a formula to sample $\mathbf{X}_t^{0,T}$ for any $t \in [0,T]$ without having to run a stochastic process forward or backward.

- *forward sampler*: we want to have a forward SDE for $\mathbf{X}_t^{0,T}$ with explicit coefficients, which might depend on $x_T$, running in a forward fashion terminating at $x_T$.

- *backward sampler*: we want to have a backward SDE for $\mathbf{Y}_t^{0,T} = \mathbf{X}_{T-t}^{0,T}$ with explicit coefficients, which might depend on $x_0$, running in a backward fashion terminating at $x_0$.

We focus on *linear SDEs* of the form $\mathrm{d}\mathbf{X}_t = -\alpha\beta_t\mathbf{X}_t\mathrm{d}t + \sigma\beta_t^{1/2}\mathrm{d}\mathbf{B}_t$, which are particularly amenable, where $(\beta_t)_{t\in[0,T]}$ is a schedule with $\beta \in \mathrm{C}([0,T],(0,+\infty))$.

### B.2.1 Brownian motion

First, we consider the Brownian motion setting and let $(\mathbf{X}_t)_{t\in[0,T]}$ be associated with $\mathbb{Q}$ with $\mathrm{d}\mathbf{X}_t = \beta_t^{1/2}\mathrm{d}\mathbf{B}_t$. We consider $(\mathbf{X}_t^{0,T})_{t\in[0,T]}$ conditioned at both ends $\mathbf{X}_0^{0,T} = x_0$ and $\mathbf{X}_T^{0,T} = x_T$. First, using (Barczy and Kern, 2013, Theorem 3.3), we have that for any $t \in [0,T]$

$$\mathbf{X}_t^{0,T} = \frac{R(t,T)}{R(0,T)}x_0 + \frac{R(0,t)}{R(0,T)}(x_T - \mathbf{X}_T) + \mathbf{X}_t,$$

with $R(s,t) = \int_s^t \beta_u \mathrm{d}u = \sigma^2(B_t - B_s)$, where for any $t \in [0,T]$, $B_t = \int_0^t \beta_s \mathrm{d}s$. Therefore, we get that

$$\mathbf{X}_t^{0,T} = (1 - \frac{B(t)}{B(T)})x_0 + \frac{B(t)}{B(T)}(x_T - \mathbf{X}_T) + \mathbf{X}_t. \tag{22}$$

(22) defines the *integral sampler*. In addition, using (Barczy and Kern, 2013, Theorem 3.2), we have

$$\mathbf{X}_0^{0,T} = x_0, \qquad \mathrm{d}\mathbf{X}_t^{0,T} = \{-\sigma^2\beta_t/\gamma(t,T)\mathbf{X}_t^{0,T} + \sigma^2\beta_t/\gamma(t,T)x_T\}\mathrm{d}t + \sigma\beta_t^{1/2}\mathrm{d}\mathbf{B}_t,$$

where $\gamma(s,t) = \sigma^2(B(t) - B(s))$. Therefore, we get that

$$\mathbf{X}_0^{0,T} = x_0, \qquad \mathrm{d}\mathbf{X}_t^{0,T} = \{-\frac{\beta_t}{B(T)-B(t)}\mathbf{X}_t^{0,T} + \frac{\beta_t}{B(T)-B(t)}x_T\}\mathrm{d}t + \sigma\beta_t^{1/2}\mathrm{d}\mathbf{B}_t. \tag{23}$$

(23) defines the *forward sampler*. Finally, we derive the *backward sampler* by considering the time-reversal of the forward unconditional process (initialized at $x_0$). Following Haussmann and Pardoux (1986),

$$\mathbf{Y}_0^{0,T} = x_T, \qquad \mathrm{d}\mathbf{Y}_t^{0,T} = \sigma^2\beta_{T-t}\nabla\log\mathbb{Q}_{T-t|0}(\mathbf{Y}_t^{0,T}|x_0)\mathrm{d}t + \sigma\beta_{T-t}^{1/2}\mathrm{d}\mathbf{B}_t. \tag{24}$$

In addition, we have that

$$\mathbf{X}_t = x_0 + \sigma B(t)^{1/2}\varepsilon_t, \qquad \varepsilon_t \sim \mathrm{N}(0,\mathrm{Id}).$$

Hence, we get that for any $t \in [0,T]$ and $x \in \mathbb{R}^d$

$$\nabla\log\mathbb{Q}_{t|0}(x|x_0) = -(x - x_0)/(\sigma^2 B(t)).$$

Combining this result and (24), we get

$$\mathbf{Y}_0^{0,T} = x_T, \qquad \mathrm{d}\mathbf{Y}_t^{0,T} = \{-\frac{\beta_{T-t}}{B(T-t)}\mathbf{Y}_t^{0,T} + \frac{\beta_{T-t}}{B(T-t)}x_0\}\mathrm{d}t + \sigma\beta_{T-t}^{1/2}\mathrm{d}\mathbf{B}_t. \tag{25}$$

Combining (22), (23) and (25), we get

$$\mathbf{X}_t^{0,T} = \lambda_t x_0 + \varphi_t(x_T - \mathbf{X}_T) + \mathbf{X}_t.$$
$$\mathbf{X}_0^{0,T} = x_0, \qquad \mathrm{d}\mathbf{X}_t^{0,T} = \{\kappa_t^f \mathbf{X}_t^{0,T} + \Psi_t^f x_T\}\mathrm{d}t + \sigma\beta_t^{1/2}\mathrm{d}\mathbf{B}_t,$$
$$\mathbf{Y}_0^{0,T} = x_T, \qquad \mathrm{d}\mathbf{Y}_t^{0,T} = \kappa_{T-t}^b \mathbf{Y}_t^{0,T} + \Psi_{T-t}^b x_0\}\mathrm{d}t + \sigma\beta_{T-t}^{1/2}\mathrm{d}\mathbf{B}_t,$$

with

$$\lambda_t = 1 - \frac{B(t)}{B(T)}, \qquad \varphi_t = \frac{B(t)}{B(T)},$$
$$\kappa_t^f = -\frac{\beta_t}{B(T)-B(t)}, \qquad \Psi_t^f = \frac{\beta_t}{B(T)-B(t)},$$
$$\kappa_t^b = -\frac{\beta_t}{B(t)}, \qquad \Psi_t^b = \frac{\beta_t}{B(t)}.$$

### B.2.2 Ornstein-Uhlenbeck

Second, we consider the Ornstein-Uhlenbeck setting and let $(\mathbf{X}_t)_{t\in[0,T]}$ with $d\mathbf{X}_t = -\alpha\beta_t\mathbf{X}_t dt + \sigma\beta_t^{1/2}d\mathbf{B}_t$, with $\alpha \neq 0$. We consider $(\mathbf{X}_t^{0,T})_{t\in[0,T]}$, the stochastic process $(\mathbf{X}_t)_{t\in[0,T]}$ conditioned at both ends $\mathbf{X}_0^{0,T} = x_0$ and $\mathbf{X}_T^{0,T} = x_T$. First, using (Barczy and Kern, 2013, Theorem 3.3), we have that for any $t \in [0,T]$

$$\mathbf{X}_t^{0,T} = \frac{R(t,T)}{R(0,T)}x_0 + \frac{R(0,t)}{R(0,T)}(x_T - \mathbf{X}_T) + \mathbf{X}_t,$$

with $R(s,t) = \exp[\alpha(B(t) - B(s))]\gamma(s,t)$, with $\gamma(s,t) = \int_s^t \sigma^2\beta(u)\exp[-2\alpha(B(t) - B(u))]du$. In particular, we have

$$\gamma(s,t) = \frac{\sigma^2}{2\alpha}(1 - \exp[-2\alpha(B(t) - B(s))]), \qquad R(s,t) = \frac{\sigma^2}{\alpha}\sinh(\alpha(B(t) - B(s))). \tag{26}$$

Therefore, we get that

$$\mathbf{X}_t^{0,T} = \frac{\sinh(\alpha(B(T)-B(t)))}{\sinh(\alpha B(T))}x_0 + \frac{\sinh(\alpha B(t))}{\sinh(\alpha B(T))}(x_T - \mathbf{X}_T) + \mathbf{X}_t. \tag{27}$$

(27) defines the *integral sampler*. In addition, using (Barczy and Kern, 2013, Theorem 3.2) and (26), we have $\mathbf{X}_0^{0,T} = x_0$ and

$$\begin{aligned}
d\mathbf{X}_t^{0,T} &= \{-\alpha\beta_t\mathbf{X}_t - \frac{\sigma^2\beta_t\exp[-2\alpha(B(T)-B(t))]}{\gamma(t,T)}\mathbf{X}_t^{0,T} + \frac{\sigma^2\beta_t\exp[-\alpha(B(T)-B(t))]}{\gamma(t,T)}x_T\}dt + \sigma\beta_t^{1/2}d\mathbf{B}_t \\
&= \{-\alpha\beta_t\mathbf{X}_t - \frac{2\alpha\beta_t}{\exp[2\alpha(B(T)-B(t))]-1}\mathbf{X}_t^{0,T} + \frac{2\alpha\beta_t}{\exp[-\alpha(B(T)-B(t))]-\exp[-\alpha(B(T)-B(t))]}x_T\}dt + \sigma\beta_t^{1/2}d\mathbf{B}_t \\
&= \{-\alpha\beta_t\frac{\exp[2\alpha(B(T)-B(t))]+1}{\exp[2\alpha(B(T)-B(t))]-1}\mathbf{X}_t^{0,T} + \frac{2\alpha\beta_t}{\exp[-\alpha(B(T)-B(t))]-\exp[-\alpha(B(T)-B(t))]}x_T\}dt + \sigma\beta_t^{1/2}d\mathbf{B}_t \\
&= \{-\alpha\beta_t\coth(\alpha(B(T) - B(t)))\mathbf{X}_t^{0,T} + \alpha\beta_t\operatorname{csch}(\alpha(B(T) - B(t)))x_T\}dt + \sigma\beta_t^{1/2}d\mathbf{B}_t.
\end{aligned}$$

In the formula, $\coth$ is the hyperbolic cotangent function defined as $\coth(x) = \frac{1}{\tanh(x)} = \frac{\cosh(x)}{\sinh(x)}$ and $\operatorname{csch}$ is the hyperbolic cosecant function defined as $\operatorname{csch}(x) = \frac{1}{\sinh(x)}$. Combining this result and (26), we get that

$$\mathbf{X}_0^{0,T} = x_0, \; d\mathbf{X}_t^{0,T} = \{-\alpha\beta_t\coth(\alpha(B(T)-B(t)))\mathbf{X}_t^{0,T}+\alpha\beta_t\operatorname{csch}(\alpha(B(T)-B(t)))x_T\}dt+\sigma\beta_t^{1/2}d\mathbf{B}_t. \tag{28}$$

(28) defines the *forward sampler*. Finally, we derive the *backward sampler* by considering the time-reversal of the forward unconditional process (initialized at $x_0$). Following Haussmann and Pardoux (1986),

$$\mathbf{Y}_0^{0,T} = x_T, \qquad d\mathbf{Y}_t^{0,T} = \{\alpha\beta_{T-t}\mathbf{Y}_t^{0,T} + \sigma^2\beta_{T-t}\nabla\log\mathbb{Q}_{T-t|0}(\mathbf{Y}_t^{0,T}|x_0)\}dt + \sigma\beta_{T-t}^{1/2}d\mathbf{B}_t. \tag{29}$$

In addition, we have that

$$\mathbf{X}_t = \exp[-\alpha B(t)]x_0 + \frac{\sigma}{\sqrt{2\alpha}}(1 - \exp[-2\alpha B(t)])^{1/2}\varepsilon_t, \qquad \varepsilon_t \sim \mathrm{N}(0, \mathrm{Id}).$$

Hence, we get that for any $t \in [0,T]$ and $x \in \mathbb{R}^d$

$$\nabla\log\mathbb{Q}_{t|0}(x|x_0) = -2\alpha(x - \exp[-\alpha B(t)]x_0)/(\sigma^2(1 - \exp[-2\alpha B(t)])).$$

Combining this result and (29), we get $\mathbf{Y}_0^{0,T} = x_T$ and

$$\begin{aligned}
d\mathbf{Y}_t^{0,T} &= \{\alpha\beta_{T-t}\mathbf{Y}_t^{0,T} - \frac{2\alpha\beta_{T-t}}{1-\exp[-\alpha B(T-t)]}\mathbf{Y}_t^{0,T} + \frac{2\alpha\beta_{T-t}\exp[-\alpha B(T-t)]}{1-\exp[-2\alpha B(T-t)]}x_0\}dt + \sigma\beta_{T-t}^{1/2}d\mathbf{B}_t \\
&= \{-\alpha\beta_{T-t}\frac{1+\exp[-2\alpha B(T-t)]}{1-\exp[-2\alpha B(T-t)]}\mathbf{Y}_t^{0,T} + \frac{2\alpha\beta_{T-t}}{\exp[\alpha B(T-t)]-\exp[-\alpha B(T-t)]}x_0\}dt + \sigma\beta_{T-t}^{1/2}d\mathbf{B}_t \\
&= \{-\alpha\beta_{T-t}\coth(\alpha B(T-t))\mathbf{Y}_t^{0,T} + \alpha\beta_{T-t}\operatorname{csch}(\alpha B(T-t))x_T\}dt + \sigma\beta_{T-t}^{1/2}d\mathbf{B}_t.
\end{aligned}$$

Therefore

$$\mathbf{Y}_0^{0,T} = x_T, \; d\mathbf{Y}_t^{0,T} = \{-\alpha\beta_{T-t}\coth(\alpha B(T-t))\mathbf{Y}_t^{0,T}+\alpha\beta_{T-t}\operatorname{csch}(\alpha B(T-t))x_T\}dt+\sigma\beta_{T-t}^{1/2}d\mathbf{B}_t. \tag{30}$$

Combining (27), (28) and (30), we get

$$\mathbf{X}_t^{0,T} = \lambda_t x_0 + \varphi_t(x_T - \mathbf{X}_T) + \mathbf{X}_t.$$
$$\mathbf{X}_0^{0,T} = x_0, \qquad \mathrm{d}\mathbf{X}_t^{0,T} = \{\kappa_t^f \mathbf{X}_t^{0,T} + \Psi_t^f x_T\}\mathrm{d}t + \sigma\beta_t^{1/2}\mathrm{d}\mathbf{B}_t,$$
$$\mathbf{Y}_0^{0,T} = x_T, \qquad \mathrm{d}\mathbf{Y}_t^{0,T} = \{\kappa_{T-t}^b \mathbf{Y}_t^{0,T} + \Psi_{T-t}^b x_0\}\mathrm{d}t + \sigma\beta_{T-t}^{1/2}\mathrm{d}\mathbf{B}_t,$$

with

$$\lambda_t = \frac{\sinh(\alpha(B(T)-B(t)))}{\sinh(\alpha B(T))}, \qquad \varphi_t = \frac{\sinh(\alpha B(t))}{\sinh(\alpha B(T))},$$
$$\kappa_t^f = -\alpha\beta_t \coth(\alpha(B(T)-B(t))), \qquad \Psi_t^f = \alpha\beta_t \operatorname{csch}(\alpha(B(T)-B(t))),$$
$$\kappa_t^b = -\alpha\beta_t \coth(\alpha B(t)), \qquad \Psi_t^b = \alpha\beta_t \operatorname{csch}(\alpha B(t)).$$

Using that $\tanh(x) \sim x$ and $\sinh(x) \sim x$ for $x \to 0$, we recover the Brownian motion setting by letting $\alpha \to 0$. Note that Albergo et al. (2023) show that given an *integral sampler*, which does not necessarily comes from a Markovian process, a *forward sampler* with the same marginals can be defined, although it does not necessarily satisfies the fact that the *paths* have the same distribution.

# C Proofs

## C.1 Proof of Proposition 2

We refer the reader to Chung and Walsh (2006); Rogers and Williams (2000) for an introduction to Doob $h$-transform. Our theoretical treatment of the Doob $h$-transform closely follows Palmowski and Rolski (2002).

First, we introduce the *infinitesimal generator* $\mathcal{A}$ given for any $f \in \mathrm{C}_c^\infty([0,T] \times \mathbb{R}^d, \mathbb{R})$, $t \in [0,T]$ and $x \in \mathbb{R}^d$ by

$$\mathcal{A}f(t,x) = \langle f_t(x), \nabla f(t,x) \rangle + \tfrac{\sigma_t^2}{2}\Delta f(t,x) + \partial_t f(t,x). \tag{31}$$

The following assumption ensures that the diffusion associated with $\mathbb{Q}$ as well as its Markovian projections are well-defined.

**A1.** *$f$, $\sigma$ and $(t,x_t) \mapsto \mathbb{E}_{\Pi_{T|t}}[\nabla \log \mathbb{Q}_{T|t}(\mathbf{X}_T|\mathbf{X}_t) \mid \mathbf{X}_t = x_t]$ are locally Lipschitz and there exist $C > 0$, $\psi \in \mathrm{C}([0,T], \mathbb{R}_+)$ such that for any $t \in [0,T]$ and $x_0, x_t \in \mathbb{R}^d$, we have*

$$\|f_t(x_t)\| \leq C(1 + \|x_t\|), \qquad C \geq \sigma_t \geq 1/C,$$
$$\|\mathbb{E}_{\Pi_{T|t}}[\nabla \log \mathbb{Q}_{T|t}(\mathbf{X}_T|\mathbf{X}_t) \mid \mathbf{X}_t = x_t]\| \leq C\psi(t)(1 + \|x_t\|).$$

We consider the following assumption, which will ensure that we can apply Doob $h$-transform techniques.

**A2.** *For any $x_0 \in \mathbb{R}^d$, $\Pi_{T|0}$ is absolutely continuous w.r.t. $\mathbb{Q}_{T|0}$. For any $x_0 \in \mathbb{R}^d$, let $\varphi_{T|0}$ be given for any $x_T \in \mathbb{R}^d$ by $\varphi_{T|0}(x_T|x_0) = \mathrm{d}\Pi_{T|0}(x_T|x_0)/\mathrm{d}\mathbb{Q}_{T|0}(x_T|x_0)$ and assume that for any $x_0 \in \mathbb{R}^d$, $x_T \mapsto \varphi_{T|0}(x_T|x_0)$ is bounded. For any $x_0 \in \mathbb{R}^d$, let $\varphi_{t|0}$ given for any $x_t \in \mathbb{R}^d$ and $t \in [0,T]$ by*

$$\varphi_{t|0}(x_t|x_0) = \int_{\mathbb{R}^d} \varphi_{T|0}(x_T|x_0)\mathrm{d}\mathbb{Q}_{T|t}(x_T|x_t). \tag{32}$$

*Finally, we assume that for any $x_0 \in \mathbb{R}^d$, $(t,x_t) \mapsto 1/\varphi_{t|0}(x_t|x_0)$ and $(t,x_t) \mapsto \mathcal{A}\varphi_{t|0}(x_t|x_0)$ are bounded.*

This means that for any $x_0 \in \mathbb{R}^d$, $(t,x_t) \mapsto \varphi_t(x_t|x_0)$ is a *good function* in the sense of (Palmowski and Rolski, 2002, Proposition 3.2). Note here that these assumptions could be relaxed on a case-by-case basis. We leave this study for future work.

The following lemma is a direct consequence of **A2** and (32). It ensures that the $h$-function $\varphi_{t|0}$ satisfies the backward Kolmogorov equation.

**Lemma 11.** *Assume* **A2**. *Then,* $\varphi \in \mathrm{C}^{1,2}([0,T] \times \mathbb{R}^d, \mathbb{R})$ *and* $\mathcal{A}\varphi_{|0} = 0$.

Using (31), we have that for any $x_0 \in \mathbb{R}^d$ and $f \in \mathrm{C}_c^\infty([0,T] \times \mathbb{R}^d, \mathbb{R})$, $t \in [0,T]$ and $x_t \in \mathbb{R}^d$

$$(\mathcal{A}(f\varphi_{|0}) - f\mathcal{A}\varphi_{|0})(t,x_t)/\varphi_{|0}(t,x_t) = \mathcal{A}f(t,x_t) + \sigma_t^2\langle\nabla f(t,x_t), \nabla \log \varphi_{t|0}(x_t|x_0)\rangle.$$

Finally, we consider the following assumption, which will ensure that the Doob $h$-transform is well-defined.

**A 3.** *For any $x_0 \in \mathbb{R}^d$, there exists $C \geq 0$ such that for any $t \in [0,T]$ and $x_t \in \mathbb{R}^d$, $\|\nabla \log \varphi_{t|0}(x_t|x_0)\| \leq C(1 + \|x_0\| + \|x_t\|)$.*

We are now ready to state and prove Proposition 2. Note that the Markovian projection is defined in Definition 1. Finally, we define $\mathcal{M}$ the space of path measures such that $\mathbb{P} \in \mathcal{M}$ if $\mathbb{P}$ is associated with $\mathrm{d}\mathbf{X}_t = \{f_t(\mathbf{X}_t) + v_t(\mathbf{X}_t)\}\mathrm{d}t + \sigma_t \mathrm{d}\mathbf{B}_t$, with $\sigma, v$ locally Lipschitz. This restriction of Markov measures allows us to apply the entropic version of the Girsanov theorem (Léonard, 2012). It has no impact on our methodology.

**Proposition.** *Assume* **A**1, **A**2, **A**3. *Let $\mathbb{M}^\star = \mathrm{proj}_{\mathcal{M}}(\Pi)$. Then,*

$$\mathbb{M}^\star = \mathrm{argmin}_{\mathbb{M}}\{\mathrm{KL}(\Pi|\mathbb{M}) \; : \; \mathbb{M} \in \mathcal{M}\},$$

$$\mathrm{KL}(\Pi|\mathbb{M}^\star) = \tfrac{1}{2}\int_0^T \mathbb{E}_{\Pi_{0,t}}[\|\sigma_t^2 \mathbb{E}_{\Pi_{T|0,t}}[\nabla \log \mathbb{Q}_{T|t}(\mathbf{X}_T|\mathbf{X}_t) \mid \mathbf{X}_0, \mathbf{X}_t] - v_t^\star\|^2]/\sigma_t^2 \mathrm{d}t.$$

*In addition, we have that for any $t \in [0,T]$, $\mathbb{M}_t^\star = \Pi_t$. In particular, $\mathbb{M}_T^\star = \Pi_T$.*

*Proof.* First, we recall that $\Pi$ is given by $\Pi = \mathbb{Q}\varphi_{0,T}$ with $\varphi_{0,T} = \frac{\mathrm{d}\Pi_{0,T}}{\mathrm{d}\mathbb{Q}_{0,T}}$. In particular, we have $\Pi_{|0} = \mathbb{Q}_{|0}\varphi_{T|0}$, where $\varphi_{T|0} = \frac{\mathrm{d}\Pi_{T|0}}{\mathrm{d}\mathbb{Q}_{T|0}}$. Therefore, using Lemma 11, (Palmowski and Rolski, 2002, Lemma 3.1, Lemma 4.1), the remark following (Palmowski and Rolski, 2002, Lemma 4.1), **A**1, **A**2 and **A**3, we get that $\Pi_{|0}$ is Markov and associated with the distribution of $(\mathbf{X}_t)_{t \in [0,T]}$ given for any $t \in [0,T]$ by

$$\mathbf{X}_t = \int_0^t \{f_s(\mathbf{X}_s) + \sigma_s^2 \nabla \log \varphi_{s|0}(\mathbf{X}_s|\mathbf{X}_0)\}\mathrm{d}s + \int_0^t \sigma_s \mathrm{d}\mathbf{B}_s, \tag{33}$$

where for any $t \in [0,T]$, $x_0, x_t \in \mathbb{R}^d$ we recall that

$$\varphi_{t|0}(x_t|x_0) = \int_{\mathbb{R}^d} \varphi_{T|0}(x_T|x_0)\mathrm{d}\mathbb{Q}_{T|t}(x_T|x_t). \tag{34}$$

First, we have that for any $t \in [0,T]$, $x_t, x_0 \in \mathbb{R}^d$

$$\mathbb{Q}_{t|0}(x_t|x_0)\varphi_{t|0}(x_t|x_0) = \int_{\mathbb{R}^d} \mathbb{Q}_{t|0,T}(x_t|x_T, x_0)\mathrm{d}\Pi_{T|0}(x_T|x_0) = \Pi_{t|0}(x_t|x_0).$$

Therefore, we get that for any $t \in [0,T]$ and $x_t, x_0 \in \mathbb{R}^d$

$$\varphi_{t|0}(x_t|x_0) = \frac{\mathrm{d}\Pi_{t|0}(x_t|x_0)}{\mathrm{d}\mathbb{Q}_{t|0}(x_t|x_0)}. \tag{35}$$

In addition, we have the following identity for any $t \in [0,T]$, $x_0, x_t, x_T \in \mathbb{R}^d$

$$\mathbb{Q}_{T|0}(x_T|x_0)\mathbb{Q}_{t|0,T}(x_t|x_0, x_T) = \mathbb{Q}_{t|0}(x_t|x_0)\mathbb{Q}_{T|t}(x_T|x_t).$$

Using (34), this result and (35), we get that for any $t \in [0,T]$ and $x_0, x_t \in \mathbb{R}^d$

$$\nabla \log \varphi_{t|0}(x_t|x_0) = \int_{\mathbb{R}^d} \frac{\Pi_{T|0}(x_T|x_0)\mathbb{Q}_{T|t}(x_T|x_t)}{\mathbb{Q}_{T|0}(x_T|x_0)\varphi_{t|0}(x_t|x_0)}\nabla \log \mathbb{Q}_{T|t}(x_T|x_t)\mathrm{d}x_T$$

$$= \int_{\mathbb{R}^d} \frac{\Pi_{T|0}(x_T|x_0)\mathbb{Q}_{t|0,T}(x_t|x_0,x_T)}{\mathbb{Q}_{t|0}(x_t|x_0)\varphi_{t|0}(x_t|x_0)}\nabla \log \mathbb{Q}_{T|t}(x_T|x_t)\mathrm{d}x_T$$

$$= \int_{\mathbb{R}^d} \frac{\Pi_{t,T|0}(x_t,x_T|x_0)}{\Pi_{t|0}(x_t|x_0)}\nabla \log \mathbb{Q}_{T|t}(x_T|x_t)\mathrm{d}x_T$$

$$= \int_{\mathbb{R}^d} \nabla \log \mathbb{Q}_{T|t}(x_T|x_t)\mathrm{d}\Pi_{T|t,0}(x_T|x_t,x_0).$$

Hence, combining this result and (33), we get

$$\mathbf{X}_t = \int_0^t \{f_s(\mathbf{X}_s) + \sigma_s^2 \mathbb{E}_{\Pi_{T|t,0}}[\nabla \log \mathbb{Q}_{T|t}(\mathbf{X}_T|\mathbf{X}_t) \mid \mathbf{X}_t, \mathbf{X}_0]\}\mathrm{d}s + \int_0^t \sigma_s \mathrm{d}\mathbf{B}_s.$$

Let $\mathbb{M}$ be Markov defined by $\mathrm{d}\mathbf{X}_t = \{f_t(\mathbf{X}_t) + v_t(\mathbf{X}_t)\}\mathrm{d}t + \sigma_t \mathrm{d}\mathbf{B}_t$, such that $\mathrm{KL}(\Pi|\mathbb{M}) < +\infty$ with $\sigma, v$ locally Lipschitz. Using (Léonard, 2012, Theorem 2.3), we get that

$$\mathrm{KL}(\Pi|\mathbb{M}) = \tfrac{1}{2}\int_0^T \mathbb{E}_{\Pi_{0,t}}[\|\sigma_t^2 \mathbb{E}_{\Pi_{T|t,0}}[\nabla \log \mathbb{Q}_{T|t}(\mathbf{X}_T|\mathbf{X}_t) \mid \mathbf{X}_t, \mathbf{X}_0] - v_t(\mathbf{X}_t)\|^2]/\sigma_t^2 \mathrm{d}t.$$

In addition, we have that for any $t \in [0,T]$,

$$\mathbb{E}_{\Pi_{0,t}}[\|\sigma_t^2 \mathbb{E}_{\Pi_{T|t,0}}[\nabla \log \mathbb{Q}_{T|t}(\mathbf{X}_T|\mathbf{X}_t) \mid \mathbf{X}_t, \mathbf{X}_0] - v_t(\mathbf{X}_t)\|^2]$$

$$\geq \mathbb{E}_{\Pi_{0,t}}[\|\sigma_t^2 \mathbb{E}_{\Pi_{T|t,0}}[\nabla \log \mathbb{Q}_{T|t}(\mathbf{X}_T|\mathbf{X}_t) \mid \mathbf{X}_t, \mathbf{X}_0] - v_t^\star(\mathbf{X}_t)\|^2],$$

where $v_t^\star(x_t) = \sigma_t^2 \mathbb{E}_{\Pi_{T|t}}[\nabla \log \mathbb{Q}_{T|t}(\mathbf{X}_T|\mathbf{X}_t) \mid \mathbf{X}_t = x_t]$ which concludes the first part of the proof. For the second part of the proof, we show that for any $t \in [0,T]$, we have $\mathbb{M}_t^\star = \Pi_t$. First, we have that $\mathbb{M}_t^\star$ and $\Pi_t$ satisfy the same Fokker-Planck equation, see (Peluchetti, 2021, Theorem 2) for instance. We conclude by uniqueness of the solutions of the Fokker-Planck equation under **A**1 and **A**3, see Bogachev et al. (2021) for instance. □

## C.2 Proof of Proposition 4

*Proof.* By the additive property of KL divergence (Léonard, 2014a), $\mathrm{KL}(\mathbb{P}|\Pi) = \mathrm{KL}(\mathbb{P}_{0,T}|\Pi_{0,T}) + \mathbb{E}_{\mathbb{P}_{0,T}}[\mathrm{KL}(\mathbb{P}_{|0,T}|\Pi_{|0,T})]$. Restricting $\Pi_{|0,T} = \mathbb{Q}_{|0,T}$ directly gives that the KL minimizer is $\Pi^\star$ with $\Pi^\star_{0,T} = \mathbb{P}_{0,T}$, and thus $\Pi^\star = \mathbb{P}_{0,T}\mathbb{Q}_{|0,T}$ which we recall is short for $\Pi^\star(\cdot) = \int_{\mathbb{R}^d \times \mathbb{R}^d} \mathbb{Q}_{|0,T}(\cdot|x_0, x_T)\mathbb{P}_{0,T}(\mathrm{d}x_0, \mathrm{d}x_T)$. $\qquad\square$

## C.3 Proof of Proposition 5

This result is a direct consequence of (Léonard, 2014b, Theorem 2.12), which we recall here for completeness.

**Proposition.** *Assume that* $\mathbb{Q} \in \mathcal{M}$, *that* $\mathbb{Q}_0 = \mathbb{Q}_T = \bar{\mathbb{Q}}$, *that for any* $x_0, x_T \in \mathbb{R}^d$, $\mathrm{d}\mathbb{Q}_{0,T}/\mathrm{d}(\bar{\mathbb{Q}}\otimes\bar{\mathbb{Q}})(x_0, x_T) \geq \exp[-A(x_0)-A(x_T)]$ *with* $A \geq 0$ *measurable,* $\int_{\mathbb{R}^d \times \mathbb{R}^d} \exp[-B(x_0) - B(x_T)]\mathrm{d}\mathbb{Q}(x_0, x_T) < +\infty$ *with* $B \geq 0$ *measurable. Assume that there exists* $t_0 \in (0,T)$ *and* $\mathsf{X}$ *measurable such that* $\mathbb{Q}_{t_0}(\mathsf{X}) > 0$ *and for all* $x \in \mathsf{X}$, $\mathbb{Q}_{0,T} \ll \mathbb{Q}_{0,T|t_0}(\cdot|\mathbf{X}_{t_0} = x)$. *In addition, assume that* $\mathrm{KL}(\pi_0|\bar{\mathbb{Q}}) < +\infty$, $\mathrm{KL}(\pi_T|\bar{\mathbb{Q}}) < +\infty$, $\int_{\mathbb{R}^d}(A + B)(x_0)\mathrm{d}\pi_0(x_0) < +\infty$, $\int_{\mathbb{R}^d}(A + B)(x_T)\mathrm{d}\pi_T(x_T) < +\infty$.

*Then there exists a unique Schrödinger Bridge* $\mathbb{P}^{\mathrm{SB}}$. *In addition let* $\mathbb{P}$ *be a Markov measure in the reciprocal class of* $\mathbb{Q}$ *such that* $\mathbb{P}_0 = \pi_0$ *and* $\mathbb{P}_T = \pi_T$. *Assume that* $\mathrm{KL}(\mathbb{P}|\mathbb{Q}) < +\infty$. *Then* $\mathbb{P}$ *is the unique Schrödinger Bridge* $\mathbb{P}^{\mathrm{SB}}$.

*Proof.* The first part of the proof is a consequence of (Léonard, 2014b, Theorem 2.12(a)). The second part is a consequence of (Léonard, 2014b, Theorem 2.12(b)) and (Léonard et al., 2014, Theorem 2.14). $\qquad\square$

## C.4 Proof of Lemma 6

**Lemma.** *Let* $\mathbb{M} \in \mathcal{M}$ *and* $\Pi \in \mathcal{R}(\mathbb{Q})$ *and assume* **A**1, **A**2, **A**3. *If* $\mathrm{KL}(\Pi|\mathbb{M}) < +\infty$ *and* $\mathrm{KL}(\mathrm{proj}_{\mathcal{M}}(\Pi)|\mathbb{M}) < +\infty$ *we have*

$$\mathrm{KL}(\Pi|\mathbb{M}) = \mathrm{KL}(\Pi|\mathrm{proj}_{\mathcal{M}}(\Pi)) + \mathrm{KL}(\mathrm{proj}_{\mathcal{M}}(\Pi)|\mathbb{M}). \qquad (36)$$

*For any* $\mathbb{P} \in \mathcal{P}(\mathcal{C})$, *if* $\mathrm{KL}(\mathbb{P}|\Pi) < +\infty$, *we have*

$$\mathrm{KL}(\mathbb{P}|\Pi) = \mathrm{KL}(\mathbb{P}|\mathrm{proj}_{\mathcal{R}(\mathbb{Q})}(\mathbb{P})) + \mathrm{KL}(\mathrm{proj}_{\mathcal{R}(\mathbb{Q})}(\mathbb{P})|\Pi). \qquad (37)$$

*Proof.* We start with the proof of (36). Similarly to Proposition 2, where we have $\mathbb{M} \in \mathcal{M}$ to ensure that we can apply (Léonard, 2012, Theorem 2.3), we get

$$\mathrm{KL}(\Pi|\mathbb{M}) = \tfrac{1}{2}\int_0^T \mathbb{E}_{\Pi_{0,t}}[\|v_t(\mathbf{X}_t) - \sigma_t^2\mathbb{E}_{\Pi_{T|0,t}}[\nabla \log \mathbb{Q}_{T|t}(\mathbf{X}_T|\mathbf{X}_t) \mid \mathbf{X}_0, \mathbf{X}_t]\|^2]/\sigma_t^2\mathrm{d}t.$$

In addition, we have

$$\mathrm{KL}(\mathrm{proj}_{\mathcal{M}}(\Pi)|\mathbb{M}) = \tfrac{1}{2}\int_0^T \mathbb{E}_{\Pi_t}[\|v_t(\mathbf{X}_t) - \sigma_t^2\mathbb{E}_{\Pi_{T|t}}[\nabla \log \mathbb{Q}_{T|t}(\mathbf{X}_T|\mathbf{X}_t) \mid \mathbf{X}_t]\|^2]/\sigma_t^2\mathrm{d}t.$$

Finally, using Proposition 2, we have that

$\mathrm{KL}(\Pi|\mathrm{proj}_{\mathcal{M}}(\Pi))$
$= \tfrac{1}{2}\int_0^T \mathbb{E}_{\Pi_{0,t}}[\|\sigma_t^2\mathbb{E}_{\Pi_{T|0,t}}[\nabla \log \mathbb{Q}(\mathbf{X}_T|\mathbf{X}_t) \mid \mathbf{X}_0, \mathbf{X}_t] - \sigma_t^2\mathbb{E}_{\Pi_{T|t}}[\nabla \log \mathbb{Q}(\mathbf{X}_T|\mathbf{X}_t) \mid \mathbf{X}_t]\|^2]/\sigma_t^2\mathrm{d}t$
$= \tfrac{1}{2}\int_0^T (\mathbb{E}_{\Pi_{0,t}}[\|\mathbb{E}_{\Pi_{T|0,t}}[\nabla \log \mathbb{Q}(\mathbf{X}_T|\mathbf{X}_t) \mid \mathbf{X}_0, \mathbf{X}_t]\|^2] - \mathbb{E}_{\Pi_t}[\|\mathbb{E}_{\Pi_{T|t}}[\nabla \log \mathbb{Q}(\mathbf{X}_T|\mathbf{X}_t) \mid \mathbf{X}_t]\|^2])\sigma_t^2\mathrm{d}t.$

Using this result, we have

$$2\mathrm{KL}(\Pi|\mathrm{proj}_{\mathcal{M}}(\Pi)) + 2\mathrm{KL}(\mathrm{proj}_{\mathcal{M}}(\Pi)|\mathbb{M})$$

$$= \int_0^T (\mathbb{E}_{\Pi_{0,t}}[\|\mathbb{E}_{\Pi_{T|0,t}}[\nabla\log\mathbb{Q}(\mathbf{X}_T|\mathbf{X}_t) \mid \mathbf{X}_0, \mathbf{X}_t]\|^2] - \mathbb{E}_{\Pi_t}[\|\mathbb{E}_{\Pi_{T|t}}[\nabla\log\mathbb{Q}(\mathbf{X}_T|\mathbf{X}_t) \mid \mathbf{X}_t]\|^2])\sigma_t^2\mathrm{d}t$$

$$+ \int_0^T \mathbb{E}_{\Pi_t}[\|v_t(\mathbf{X}_t)/\sigma_t^2 - \mathbb{E}_{\Pi_{T|t}}[\nabla\log\mathbb{Q}_{T|t}(\mathbf{X}_T|\mathbf{X}_t) \mid \mathbf{X}_t]\|^2]\sigma_t^2\mathrm{d}t$$

$$= \int_0^T (\mathbb{E}_{\Pi_{0,t}}[\|\mathbb{E}_{\Pi_{T|0,t}}[\nabla\log\mathbb{Q}(\mathbf{X}_T|\mathbf{X}_t) \mid \mathbf{X}_0, \mathbf{X}_t]\|^2] - \mathbb{E}_{\Pi_t}[\|\mathbb{E}_{\Pi_{T|t}}[\nabla\log\mathbb{Q}(\mathbf{X}_T|\mathbf{X}_t) \mid \mathbf{X}_t]\|^2])\sigma_t^2\mathrm{d}t$$

$$+ \int_0^T (\mathbb{E}_{\Pi_t}[\|v_t(\mathbf{X}_t)/\sigma_t^2\|^2] + \mathbb{E}_{\Pi_t}[\|\mathbb{E}_{\Pi_{T|t}}[\nabla\log\mathbb{Q}_{T|t}(\mathbf{X}_T|\mathbf{X}_t) \mid \mathbf{X}_t]\|^2])\sigma_t^2\mathrm{d}t$$

$$- 2\int_0^T \mathbb{E}_{\Pi_t}[\langle v_t(\mathbf{X}_t)/\sigma_t^2, \mathbb{E}_{\Pi_{T|t}}[\nabla\log\mathbb{Q}_{T|t}(\mathbf{X}_T|\mathbf{X}_t) \mid \mathbf{X}_t]\rangle]\sigma_t^2\mathrm{d}t$$

$$= \int_0^T \mathbb{E}_{\Pi_{0,t}}[\|\mathbb{E}_{\Pi_{T|0,t}}[\nabla\log\mathbb{Q}(\mathbf{X}_T|\mathbf{X}_t) \mid \mathbf{X}_0, \mathbf{X}_t]\|^2]\sigma_t^2\mathrm{d}t + \int_0^T \mathbb{E}_{\Pi_t}[\|v_t(\mathbf{X}_t)/\sigma_t^2\|^2]\sigma_t^2\mathrm{d}t$$

$$- 2\int_0^T \mathbb{E}_{\Pi_t}[\langle v_t(\mathbf{X}_t)/\sigma_t^2, \mathbb{E}_{\Pi_{T|t}}[\nabla\log\mathbb{Q}_{T|t}(\mathbf{X}_T|\mathbf{X}_t) \mid \mathbf{X}_t]\rangle]\sigma_t^2\mathrm{d}t$$

$$= \int_0^T \mathbb{E}_{\Pi_{0,t}}[\|\mathbb{E}_{\Pi_{T|0,t}}[\nabla\log\mathbb{Q}(\mathbf{X}_T|\mathbf{X}_t) \mid \mathbf{X}_0, \mathbf{X}_t]\|^2]\sigma_t^2\mathrm{d}t + \int_0^T \mathbb{E}_{\Pi_t}[\|v_t(\mathbf{X}_t)/\sigma_t^2\|^2]\sigma_t^2\mathrm{d}t$$

$$- 2\int_0^T \mathbb{E}_{\Pi_{0,t}}[\langle v_t(\mathbf{X}_t)/\sigma_t^2, \mathbb{E}_{\Pi_{T|0,t}}[\nabla\log\mathbb{Q}_{T|t}(\mathbf{X}_T|\mathbf{X}_t) \mid \mathbf{X}_0, \mathbf{X}_t]\rangle]\sigma_t^2\mathrm{d}t = 2\mathrm{KL}(\Pi|\mathbb{M}),$$

which concludes the first part of the proof.

For (37), define $\Pi^\star = \mathrm{proj}_{\mathcal{R}(\mathbb{Q})}(\mathbb{P}) = \mathbb{P}_{0,T}\mathbb{Q}_{|0,T}$. Using (Csiszár, 1975, Equation 2.6), we have

$$\mathrm{KL}(\mathbb{P}|\Pi) = \mathrm{KL}(\mathbb{P}|\Pi^\star) + \int_{\mathcal{C}} \log(\tfrac{\mathrm{d}\Pi^\star}{\mathrm{d}\Pi}(\omega))\mathrm{d}\mathbb{P}(\omega)$$

$$= \mathrm{KL}(\mathbb{P}|\Pi^\star) + \int_{\mathbb{R}^d\times\mathbb{R}^d} \log(\tfrac{\mathrm{d}\Pi^\star_{0,T}}{\mathrm{d}\Pi_{0,T}}(x_0, x_1))\mathrm{d}\mathbb{P}_{0,T}(x_0, x_1)$$

$$= \mathrm{KL}(\mathbb{P}|\Pi^\star) + \int_{\mathbb{R}^d\times\mathbb{R}^d} \log(\tfrac{\mathrm{d}\Pi^\star_{0,T}}{\mathrm{d}\Pi_{0,T}}(x_0, x_1))\mathrm{d}\Pi^\star_{0,T}(x_0, x_1) = \mathrm{KL}(\mathbb{P}|\Pi^\star) + \mathrm{KL}(\Pi^\star|\Pi),$$

which concludes the proof. $\qquad\square$

## C.5 Proof of Proposition 7

**Proposition.** *Assume that the conditions of Proposition 5 and Lemma 6 apply for $\mathbb{P}^n$ for every $n \in \mathbb{N}$ and for the Schrödinger Bridge $\mathbb{P}^{SB}$, we have $\mathrm{KL}(\mathbb{P}^{n+1}|\mathbb{P}^{SB}) \leq \mathrm{KL}(\mathbb{P}^n|\mathbb{P}^{SB}) < \infty$, and $\lim_{n\to+\infty}\mathrm{KL}(\mathbb{P}^n|\mathbb{P}^{n+1}) = 0$.*

*Proof.* We follow the technique of Rüschendorf (1995) but for the *reverse* Kullback–Leibler divergence. Applying Lemma 6, we get for any $N \in \mathbb{N}$

$$\mathrm{KL}(\mathbb{P}^0|\mathbb{P}^{SB}) = \mathrm{KL}(\mathbb{P}^0|\mathbb{P}^1) + \mathrm{KL}(\mathbb{P}^1|\mathbb{P}^{SB}) = \sum_{i=0}^N \mathrm{KL}(\mathbb{P}^i|\mathbb{P}^{i+1}) + \mathrm{KL}(\mathbb{P}^{N+1}|\mathbb{P}^{SB}),$$

which concludes the proof. $\qquad\square$

## C.6 Proof of Theorem 8

**Theorem.** *Assume that the conditions of Proposition 5 and Lemma 6 apply for $\mathbb{P}^n$ for every $n \in \mathbb{N}$ and for the Schrödinger Bridge $\mathbb{P}^{SB}$, the IMF sequence $(\mathbb{P}^n)_{n\in\mathbb{N}}$ admits a unique fixed point $\mathbb{P}^\star = \mathbb{P}^{SB}$, and $\lim_{n\to+\infty}\mathrm{KL}(\mathbb{P}^n|\mathbb{P}^\star) = 0$.*

*Proof.* By Proposition 7, $\mathrm{KL}(\mathbb{P}^n|\mathbb{P}^{SB}) \leq \mathrm{KL}(\mathbb{P}^0|\mathbb{P}^{SB}) < \infty$ for all $n \in \mathbb{N}$. Now, using the coercivity of $\mathrm{KL}(\cdot|\mathbb{P}^{SB})$ (this is where our analysis differs from the one of the IPF), we have that the IMF sequence $(\mathbb{P}^n)_{n\in\mathbb{N}}$ and its subsequences $(\mathbb{M}^{n+1})_{n\in\mathbb{N}}$ and $(\Pi^n)_{n\in\mathbb{N}}$ are subsets of $\{\mathbb{P} \in \mathcal{P}(\mathcal{C}) : \mathrm{KL}(\mathbb{P}|\mathbb{P}^{SB}) \leq \mathrm{KL}(\mathbb{P}^0|\mathbb{P}^{SB})\}$ which is (relatively) compact. Thus, $(\mathbb{M}^{n+1})_{n\in\mathbb{N}}$ contains a convergent subsequence $\mathbb{M}^{n_j} \to \mathbb{M}^\star$ as $j \to \infty$, and $(\Pi^{n_j})_{j\in\mathbb{N}}$ contains a further convergent subsequence $\Pi^{n_{j_k}} \to \Pi^\star$ weakly as $k \to \infty$. As the Markov and the reciprocal classes are closed under weak convergence, $\mathbb{M}^\star \in \mathcal{M}$ and $\Pi^\star \in \mathcal{R}(\mathbb{Q})$. Now, by the lower semi-continuity of KL divergence in the weak topology (van Erven and Harremoes, 2014, Theorem 19), $0 \leq \mathrm{KL}(\mathbb{M}^\star|\Pi^\star) \leq \liminf_{k\to\infty}\mathrm{KL}(\mathbb{M}^{n_{j_k}}|\Pi^{n_{j_k}}) = 0$. Hence, $\mathbb{M}^\star = \Pi^\star$ which we denote as $\mathbb{P}^\star \in \mathcal{M} \cap \mathcal{R}(\mathbb{Q})$. Also, $\mathbb{P}^\star$ satisfies $\mathbb{P}_0^\star = \pi_0$ and $\mathbb{P}_T^\star = \pi_T$ as is satisfied by all $\mathbb{P}^n$. By Proposition 5, $\mathbb{P}^\star$ is the unique Schrödinger Bridge $\mathbb{P}^{SB}$. Finally, $\lim_{n\to+\infty}\mathrm{KL}(\mathbb{P}^n|\mathbb{P}^\star) = 0$ follows using $\lim_{k\to\infty}\mathrm{KL}(\mathbb{M}^{n_{j_k}}|\mathbb{P}^\star) = 0$ and the monotonicity of $\mathrm{KL}(\mathbb{P}^n|\mathbb{P}^{SB})$ by Proposition 7. $\qquad\square$

### C.7 Proof of Proposition 9

*Proof.* The proof is similar to the one of Proposition 2. $\qquad\square$

In particular, the time-reversal of $\mathbb{Q}_{|0,T}(\cdot|x_0, x_T)$ is associated with

$$\mathrm{d}\mathbf{Y}_t^{0,T} = \{-f_{T-t}(\mathbf{Y}_t^{0,T}) + \sigma_{T-t}^2 \nabla \log \mathbb{Q}_{T-t|0}(\mathbf{Y}_t^{0,T}|x_0)\}\mathrm{d}t + \sigma_{T-t}\mathrm{d}\mathbf{B}_t, \qquad \mathbf{Y}_0^{0,T} = x_T. \quad (38)$$

One can view both (11) (12) as SDEs with drift defined as the conditional expectation of the drift of (2) (38) under $\Pi_{0,T|t}$ in the forward and backward directions respectively.

### C.8 Proof of Proposition 10

*Proof.* We proceed by induction. Firstly, for $\Pi_{0,T}^0 = \mathbb{Q}_{0,T}$, $\Pi^0 = \tilde{\mathbb{P}}^0 = \mathbb{Q}$ at initialization. We can also define $\mathbb{M}^0 = \mathbb{Q}$, such that $\mathbb{M}^0 = \tilde{\mathbb{P}}^0$ and $\Pi^n = \mathrm{proj}_{\mathcal{R}(\mathbb{Q})}(\mathbb{M}^n)$ for all $n \in \mathbb{N}$. By (De Bortoli et al., 2021, Section 3.5), the optimal DSB sequence $\tilde{\mathbb{P}}^n$ is Markov and $\tilde{\mathbb{P}}^n = \tilde{\mathbb{P}}_{0,T}^n \mathbb{Q}_{|0,T}$, where $\tilde{\mathbb{P}}_{0,T}^n$ is the IPF sequence of the static SB problem. In other words, $\tilde{\mathbb{P}}^n \in \mathcal{M} \cap \mathcal{R}(\mathbb{Q})$.

Suppose $\mathbb{M}^{2n+1} = \tilde{\mathbb{P}}^{2n+1}$. By definition, $\mathbb{M}_0^{2n+2} = \tilde{\mathbb{P}}_0^{2n+2} = \pi_0$, i.e. both forward processes are initialized at $\pi_0$. In DSB, by De Bortoli et al. (2021), $\tilde{\mathbb{P}}^{2n+2}$ is defined as the time-reversal of $\tilde{\mathbb{P}}^{2n+1}$, such that $\tilde{\mathbb{P}}_{|0}^{2n+2} = \tilde{\mathbb{P}}_{|0}^{2n+1}$. Hence, $\tilde{\mathbb{P}}^{2n+2} = \pi_0 \tilde{\mathbb{P}}_{|0}^{2n+1}$.

In DSBM, we first perform reciprocal projection $\Pi^{2n+1} = \mathrm{proj}_{\mathcal{R}(\mathbb{Q})}(\mathbb{M}^{2n+1}) = \mathbb{M}_{0,T}^{2n+1}\mathbb{Q}_{|0,T}$. Since $\mathbb{M}^{2n+1} = \tilde{\mathbb{P}}^{2n+1} \in \mathcal{R}(\mathbb{Q})$, however, we have that $\Pi^{2n+1} = \mathbb{M}^{2n+1}$. Furthermore, since $\mathbb{M}^{2n+1} = \tilde{\mathbb{P}}^{2n+1} \in \mathcal{M}$, $\mathrm{proj}_{\mathcal{M}}(\Pi^{2n+1}) = \mathrm{proj}_{\mathcal{M}}(\mathbb{M}^{2n+1}) = \mathbb{M}^{2n+1}$. Thus, $\mathbb{M}^{2n+2}$ given by (9) is such that $\mathbb{M}_0^{2n+2} = \pi_0$ and $\mathbb{M}_{|0}^{2n+2} = \mathrm{proj}_{\mathcal{M}}(\Pi^{2n+1})_{|0} = \mathbb{M}_{|0}^{2n+1}$. We conclude that $\mathbb{M}^{2n+2} = \pi_0 \mathbb{M}_{|0}^{2n+1} = \tilde{\mathbb{P}}^{2n+2}$. Similar arguments holds for the the reverse projection (13). Therefore, $\mathbb{M}^n = \tilde{\mathbb{P}}^n$ for all $n \in \mathbb{N}$. $\qquad\square$

### C.9 The set of Markov measures is not convex

The result of Lemma 6 should be compared with the information geometry result of (Csiszár, 1975, Theorem 2.2), which states that if $\mathcal{C}$ is a convex set and $\mathbb{P} \in \mathcal{C}$, then under mild conditions, $\mathrm{KL}(\mathbb{P}|\mathbb{Q}) = \mathrm{KL}(\mathbb{P}|\mathrm{proj}_{\mathcal{C}}(\mathbb{Q})) + \mathrm{KL}(\mathrm{proj}_{\mathcal{C}}(\mathbb{Q})|\mathbb{Q})$, where $\mathrm{proj}_{\mathcal{C}}(\mathbb{Q}) = \mathrm{argmin}_{\mathbb{P}}\{\mathrm{KL}(\mathbb{P}|\mathbb{Q}) : \mathbb{P} \in \mathcal{C}\}$ is the projection of $\mathbb{Q}$ on $\mathcal{C}$. Note that, contrary to Lemma 6, (Csiszár, 1975, Theorem 2.2) is given for the *forward* Kullback–Leibler divergence whereas Lemma 6 is given for the *reverse* KL divergence. In addition, (Csiszár, 1975, Theorem 2.2) requires the projection set $\mathcal{C}$ to be convex which is not satisfied for the space of Markov measures $\mathcal{M}$. We give a simple counter-example proving that the set of Markov measures is not convex.

Let $p_1(x_0, x_1, x_2) = p_1(x_0)p_1(x_1|x_0)p_1(x_2|x_1)$ and $p_2(x_0, x_1, x_2) = p_2(x_0)p_2(x_1|x_0)p_2(x_2|x_1)$ on $\{0,1\}^3$ such that

$$p_1(x_0 = 1) = \alpha_0, \qquad p_1(x_1 = 1|x_0) = \alpha_1, \qquad p_1(x_2 = 1|x_1) = \alpha_2.$$

Additionally, we set

$$p_2(x_0 = 1) = \beta_0, \qquad p_2(x_1 = 1|x_0) = \beta_1, \qquad p_2(x_2 = 1|x_1) = \beta_2.$$

Finally, we set $q = (1/2)p_1 + (1/2)p_2$. Consider $q(x_2 = 1|x_1 = 1, x_0 = 1) = q(x_2 = 1, x_1 = 1, x_0 = 1)/q(x_1 = 1, x_0 = 1)$ and $q(x_2 = 1|x_1 = 1) = q(x_2 = 1, x_1 = 1)/q(x_1 = 1)$. Let

$$\begin{aligned}
\Delta &= 4[q(x_2 = 1, x_1 = 1, x_0 = 1)q(x_1 = 1) - q(x_2 = 1, x_1 = 1)q(x_1 = 1, x_0 = 1)] \\
&= (\alpha_0\alpha_1\alpha_2 + \beta_0\beta_1\beta_2)(\alpha_1 + \beta_1) - (\alpha_1\alpha_2 + \beta_1\beta_2)(\alpha_0\alpha_1 + \beta_0\beta_1) \\
&= \alpha_0\alpha_1\beta_1\alpha_2 + \beta_0\alpha_1\beta_1\beta_2 - \beta_0\alpha_1\beta_1\alpha_2 - \alpha_0\alpha_1\beta_1\beta_2 \\
&= \alpha_1\beta_1\beta_2(\beta_0 - \alpha_0) + \alpha_1\beta_1\alpha_2(\alpha_0 - \beta_0) \\
&= \alpha_1\beta_1(\beta_0 - \alpha_0)(\beta_2 - \alpha_2).
\end{aligned}$$

$q$ is Markov if and only if $\Delta = 0$. Therefore $q$ is not Markov as soon as $\alpha_0 \neq \beta_0$ and $\alpha_2 \neq \beta_2$.

## C.10 Impact of the resolution on the entropic regularization

In this section, we show how the resolution of the input affects the entropic regularization. First, we consider the Schrödinger bridge problem between $\pi_0$ and $\pi_1$ ($T = 1$) with $\mathbb{Q}$ associated with $(\sigma\mathbf{B}_t)_{t\in[0,1]}$. In that case the *static* Schrödinger Bridge is given by $\mathbb{P}_{0,1}^\star$ such that

$$\mathbb{P}_{0,1}^\star = \mathrm{argmin}\{\textstyle\int_{\mathbb{R}^d\times\mathbb{R}^d} \|x_0 - x_1\|^2 \mathrm{d}\mathbb{P}(x_0,x_1) - \varepsilon\mathrm{KL}(\mathbb{P}|\pi_0 \otimes \pi_1) \; : \; \mathbb{P}_0 = \pi_0, \; \mathbb{P}_1 = \pi_1\}$$

where $\varepsilon = 2\sigma^2$. We now describe the solution of a higher dimensional problem given by the *upsampling* of the marginals. Let $f \in \mathbb{N}$ with $f \geq 1$ and down $: \; \mathbb{R}^{fd} \to \mathbb{R}^d$ such that for any $k \in \{1,\ldots,d\}$, and $x \in \mathbb{R}^{fd}$, $\mathrm{down}(x) = \bar{x}$ with $\bar{x} \in \mathbb{R}^d$ and $\bar{x}_k = x_{kf}$. We also denote up $: \; \mathbb{R}^d \to \mathbb{R}^{fd}$ such that for any $k \in \{1,\ldots,d\}$, and $\bar{x} \in \mathbb{R}^d$, $\mathrm{up}(\bar{x}) = x$ with $x \in \mathbb{R}^{fd}$ and $\bar{x}_k = x_{(k-1)f+j}$, for $j \in \{1,\ldots,d\}$. Note that down $\circ$ up $= \mathrm{Id}$.

We denote $\pi_0^{\mathrm{up}} = \mathrm{up}_{\#}\pi_0$ and $\pi_1^{\mathrm{up}} = \mathrm{up}_{\#}\pi_1$. We extend up and down to $\mathbb{R}^d \times \mathbb{R}^d$ and $\mathbb{R}^{fd} \times \mathbb{R}^{fd}$ by letting for any $\bar{x}, \bar{y} \in \mathbb{R}^d$ and $x, y \in \mathbb{R}^{fd}$

$$\mathrm{up}(\bar{x}, \bar{y}) = (\mathrm{up}(\bar{x}), \mathrm{up}(\bar{y})), \qquad \mathrm{down}(x, y) = (\mathrm{down}(x), \mathrm{down}(y)).$$

First, note that since up$\circ$down $= \mathrm{Id}$ on $\mathrm{up}(\mathbb{R}^d)$ we get that for any $\mathbb{P}$ supported on $\mathrm{up}(\mathbb{R}^d) \times \mathrm{up}(\mathbb{R}^d)$, using (Kullback, 1997, Theorem 4.1)

$$\mathrm{KL}(\mathbb{P}|\pi_0^{\mathrm{up}} \otimes \pi_1^{\mathrm{up}}) = \mathrm{KL}(\mathrm{down}_{\#}\mathbb{P}|\pi_0 \otimes \pi_1).$$

Second let $\mathbb{P}$ such that $\mathbb{P}_0 = \pi_0^{\mathrm{up}}$ and $\mathbb{P}_1 = \pi_1^{\mathrm{up}}$. Then, $\mathbb{P}$ is supported on $\mathrm{up}(\mathbb{R}^d) \times \mathrm{up}(\mathbb{R}^d)$. Therefore, for any $\mathbb{P}$ such that $\mathbb{P}_0 = \pi_0^{\mathrm{up}}$ and $\mathbb{P}_1 = \pi_1^{\mathrm{up}}$

$$\mathrm{KL}(\mathbb{P}|\pi_0^{\mathrm{up}} \otimes \pi_1^{\mathrm{up}}) = \mathrm{KL}(\mathrm{down}_{\#}\mathbb{P}|\pi_0 \otimes \pi_1).$$

Finally, for any $\mathbb{P}$ such that $\mathbb{P}_0 = \pi_0^{\mathrm{up}}$ and $\mathbb{P}_1 = \pi_1^{\mathrm{up}}$ we have

$$\textstyle\int_{\mathbb{R}^{fd}\times\mathbb{R}^{fd}} \|x_0 - x_1\|^2 \mathrm{d}\mathbb{P}(x_0,x_1) = f^2 \int_{\mathbb{R}^d\times\mathbb{R}^d} \|\bar{x}_0 - \bar{x}_1\|^2 \mathrm{d}(\mathrm{down}_{\#}\mathbb{P})(\bar{x}_0,\bar{x}_1).$$

Therefore, for any $\mathbb{P}$ such that $\mathbb{P}_0 = \pi_0^{\mathrm{up}}$ and $\mathbb{P}_1 = \pi_1^{\mathrm{up}}$ we have

$$\textstyle\int_{\mathbb{R}^{fd}\times\mathbb{R}^{fd}} \|x_0 - x_1\|^2 \mathrm{d}\mathbb{P}(x_0,x_1) - \varepsilon\mathrm{KL}(\mathbb{P}|\pi_0^{\mathrm{up}} \otimes \pi_1^{\mathrm{up}})$$
$$= f^2 (\textstyle\int_{\mathbb{R}^{fd}\times\mathbb{R}^{fd}} \|\bar{x}_0 - \bar{x}_1\|^2 \mathrm{d}(\mathrm{down}_{\#}\mathbb{P})(\bar{x}_0,\bar{x}_1) - (\varepsilon/f^2)\mathrm{KL}(\mathrm{down}_{\#}\mathbb{P}|\pi_0 \otimes \pi_1)).$$

Therefore, we have the following result.

**Proposition 12.** *Let $\varepsilon > 0$. $\mathbb{P}^\star$ is the solution of the static Schrödinger bridge with marginals $\pi_0^{\mathrm{up}}$, $\pi_1^{\mathrm{up}}$ and regularization $\varepsilon$ if and only if $\mathrm{down}_{\#}\mathbb{P}^\star$ is the solution of the static Schrödinger bridge with marginals $\pi_0$, $\pi_1$ and regularization $\varepsilon/f^2$.*

This means in particular that the Schrödinger Bridge is not invariant via upsampling. In Appendix I.4, we confirm these results visually upon noting that for the same $\sigma$, running DSBM at resolution $64 \times 64$ and $128 \times 128$ gives different results, even after downsampling of the $128 \times 128$ results.

# D  Convergence of IMF in the Gaussian setting

In this section, we study the IMF in an one-dimensional Gaussian case. We consider $T = 1$, $\Pi_0 = \Pi_1 = \mathrm{N}(0, (1/2\beta^2))$ and $\mathbb{Q}$ associated with $(\sigma\mathbf{B}_t)_{t\in[0,1]}$ where $\sigma > 0$. In what follows, we let

$$\Sigma^0 = (1/2\beta^2) \begin{pmatrix} 1 & c^2 \\ c^2 & 1 \end{pmatrix}, \tag{39}$$

where $c \in [0, 1]$. We also denote $\bar{\sigma}^2 = 2\sigma^2\beta^2$. We start with the following result which gives an explicit expression of some marginals of the reciprocal projection.

**Lemma 13.** *Let $\Pi_{0,1}^0 = \mathrm{N}(0, \Sigma^0)$ with $\Sigma^0$ given by (39). Let $\Pi^0 = \Pi_{0,1}^0 \mathbb{Q}_{|0,1}$. For any $t \in [0, 1]$, we have that $\Pi_{0,1,t}^0 = \mathrm{N}(0, \Sigma)$ with*

$$\Sigma = (1/2\beta^2) \begin{pmatrix} 1 & c^2 & a_k^t \\ c^2 & 1 & a_k^{1-t} \\ a_k^t & a_k^{1-t} & b_k^t \end{pmatrix},$$

*where we have*

$$a_k^t = 1 - t + tc^2, \qquad b_k^t = 1 + t(1-t)(2(c^2 - 1) + \bar{\sigma}^2).$$

*Proof.* Let $t \in [0, 1]$ and $\mathbf{X}_t^{0,1} \sim \Pi_{t|0,1}^0 = \mathbb{Q}_{t|0,1}$. Using (Barczy and Kern, 2013, Theorem 3.3), we get that

$$\mathbf{X}_t^{0,1} = (1 - t)\mathbf{X}_0 + t\mathbf{X}_1 + \sigma(t(1 - t))^{1/2}\mathbf{Z},$$

with $\mathbf{Z} \sim \mathrm{N}(0, \mathrm{Id})$ independent from $(\mathbf{X}_0, \mathbf{X}_1)$. Hence, we get that $\mathbb{E}[\mathbf{X}_t^{0,1}] = 0$, and

$$\mathrm{Cov}(\mathbf{X}_0, \mathbf{X}_t^{0,1}) = \mathbb{E}[\mathbf{X}_t^{0,1}\mathbf{X}_0] = (1 - t)/(2\beta^2) + tc^2/(2\beta^2) = a_k^t/(2\beta^2).$$

Similarly, we get that $\mathrm{Cov}(\mathbf{X}_1, \mathbf{X}_t^{0,1}) = a_k^{1-t}/(2\beta^2)$. Finally, we get that

$$
\begin{aligned}
\mathrm{Var}(\mathbf{X}_t^{0,1}) &= \mathbb{E}[((1 - t)\mathbf{X}_0 + t\mathbf{X}_1)^2] + \sigma^2 t(1 - t) \\
&= (1 - t)^2/(2\beta^2) + 2t(1 - t)c^2/(2\beta^2) + t^2/(2\beta^2) + \bar{\sigma}^2 t(1 - t)/(2\beta^2) \\
&= (1 - 2t + 2t^2 + 2t(1 - t)c^2 + \bar{\sigma}^2 t(1 - t))/(2\beta^2) \\
&= (1 + t(1 - t)(2(c^2 - 1) + \bar{\sigma}^2))/(2\beta^2),
\end{aligned}
$$

which concludes the proof. $\qquad\square$

Leveraging Lemma 13, we can give an explicit expression of the drift term in the Markovian projection.

**Lemma 14.** *Let $\Pi_{0,1}^0 = \mathrm{N}(0, \Sigma^0)$ with $\Sigma^0$ given by (39). Let $\Pi^0 = \Pi_{0,1}^0 \mathbb{Q}_{|0,1}$. For any $t \in [0, 1]$ and $x_t \in \mathbb{R}^d$, we have that*

$$\sigma^2 \mathbb{E}_{\Pi_{1|t}^0}[\nabla \log \mathbb{Q}_{1|t}|\mathbf{X}_t = x_t] = \tfrac{(1-2t)(c^2-1)-\bar{\sigma}^2 t}{1+t(1-t)(2(c^2-1)+\bar{\sigma}^2)} x_t.$$

*Hence, the Markovian projection of $\Pi^0$, denoted $\mathbb{M}^1$ is associated with $(\mathbf{X}_t)_{t\in[0,1]}$ with*

$$\mathbf{X}_0 \sim \Pi_0, \qquad \mathrm{d}\mathbf{X}_t = \tfrac{(1-2t)(c^2-1)-\bar{\sigma}^2 t}{1+t(1-t)(2(c^2-1)+\bar{\sigma}^2)}\mathbf{X}_t + \sigma\mathrm{d}\mathbf{B}_t. \qquad (40)$$

*Proof.* Using (Barczy and Kern, 2013, Theorem 3.2), we get that $\mathbb{Q}_{|0,1}$ is associated with

$$\mathrm{d}\mathbf{X}_t^{0,1} = \sigma^2 \nabla \log \mathbb{Q}_{1|t}(x_1|\mathbf{X}_t)\mathrm{d}t + \sigma\mathrm{d}\mathbf{B}_t, \qquad \mathbf{X}_0^{0,1} = x_0,$$

where for any $t \in [0, 1)$, we have and $x_t \in \mathbb{R}^d$, $\sigma^2 \nabla \log \mathbb{Q}_{1|t}(x_1|x_t) = (x_1 - x_t)/(1 - t)$. Therefore, we get that $\mathbb{M}^1$ is associated with $(\mathbf{X}_t)_{t\in[0,1]}$ such that

$$\mathrm{d}\mathbf{X}_t = \mathbb{E}_{\Pi_{1|t}^0}[\sigma^2 \nabla \log \mathbb{Q}_{1|t}(\mathbf{X}_1|\mathbf{X}_t)|\mathbf{X}_t]\mathrm{d}t + \sigma\mathrm{d}\mathbf{B}_t, \qquad \mathbf{X}_0 = x_0.$$

Therefore, we get that

$$\mathrm{d}\mathbf{X}_t = (\mathbb{E}_{\Pi_{1|t}^0}[\mathbf{X}_1|\mathbf{X}_t] - x_t)/(1 - t)\mathrm{d}t + \sigma\mathrm{d}\mathbf{B}_t, \qquad \mathbf{X}_0 = x_0.$$

Using Lemma 13, we have that for any $t \in [0, 1]$ and $x_t \in \mathbb{R}^d$, $\mathbb{E}_{\Pi_{1|t}^0}[\mathbf{X}_1|\mathbf{X}_t = x_t] = a_k^{1-t}/b_k^t x_t$. In addition, we have for any $t \in [0, 1]$

$$
\begin{aligned}
a_k^{1-t}/b_k^t - 1 &= (t + (1 - t)c^2 - 1 - t(1 - t)(2(c^2 - 1) + \bar{\sigma}^2))/(1 + t(1 - t)(2(c^2 - 1) + \bar{\sigma}^2)) \\
&= ((1 - t)(c^2 - 1) - t(1 - t)(2(c^2 - 1) + \bar{\sigma}^2))/(1 + t(1 - t)(2(c^2 - 1) + \bar{\sigma}^2)) \\
&= (1 - t)((c^2 - 1) - t(2(c^2 - 1) + \bar{\sigma}^2))/(1 + t(1 - t)(2(c^2 - 1) + \bar{\sigma}^2)) \\
&= (1 - t)((1 - 2t)(c^2 - 1) - t\bar{\sigma}^2))/(1 + t(1 - t)(2(c^2 - 1) + \bar{\sigma}^2)),
\end{aligned}
$$

which concludes the proof. $\qquad\square$

Note that since $\sigma > 0$ and $c^2 \in [0, 1]$, we get that for any $t \in [0, 1]$, $1 + t(1 - t)(2(c^2 - 1) + \bar{\sigma}^2) > 0$ and therefore the drift is well-defined, smooth and sublinear. In particular, (40) admits a unique strong solution. In what follows, we denote $G : [0, 1] \times [0, 1] \to \mathbb{R}$ given for any $t \in [0, 1]$ by

$$G(t, c^2) = \int_0^t \tfrac{(1-2s)(c^2-1)-\bar{\sigma}^2 s}{1+s(1-s)(2(c^2-1)+\bar{\sigma}^2)}\mathrm{d}s.$$

We have the following useful lemma.

**Lemma 15.** *Let $c \in [0,1]$, $\bar{\sigma} > 0$ and $p = 2c^2 + \bar{\sigma}^2$. We distinguish three cases:*

*(a) If $p < 2$, then*

$$G(1, c^2) = -\bar{\sigma}^2 (4 - p^2)^{-1/2} \tan^{-1}((4 - p^2)^{1/2}/p).$$

*(b) If $p = 2$, then*

$$G(1, c^2) = -\bar{\sigma}^2/2.$$

*(c) If $p > 2$, then*

$$G(1, c^2) = -\bar{\sigma}^2 (p^2 - 4)^{-1/2} \tanh^{-1}((p^2 - 4)^{1/2}/p).$$

This lemma is useful combined with the following proposition, which gives the analytical update formula of Gaussian IMF covariances as a function of $G(1, c^2)$.

**Proposition 16.** *Let $\Pi^0_{0,1} = \mathrm{N}(0, \Sigma^0)$ with $\Sigma^0$ given by (39). Then $\Pi^1_{0,1} = \mathrm{N}(0, \Sigma^1)$ with*

$$\Sigma^1 = (1/2\beta^2) \begin{pmatrix} 1 & c_1^2 \\ c_1^2 & 1 \end{pmatrix}, \qquad c_1^2 = f(c_0^2),$$

*with $f : [0,1] \to [0,1]$ given for any $c \in [0,1]$ by*

$$f(c^2) = \exp[G(1, c^2)].$$

*Proof.* We have that $\mathbf{X}_1 = \exp[G(1, c^2)]\mathbf{X}_0 + \mathbf{M}_1$, where $\mathbf{M}_1$ is a Gaussian random variable with zero mean independent from $\mathbf{X}_0$. Therefore, we get that $\mathrm{Cov}(\mathbf{X}_0, \mathbf{X}_1) = \exp[G(1, c^2)]/(2\beta^2)$. In addition, we have that $\mathbb{E}[\mathbf{X}_1^2] = \mathbb{E}[\mathbf{X}_0^2] = 1/2\beta^2$, which concludes the proof. $\square$

Iterating the procedure in Proposition 16, we obtain a sequence of IMF covariances $(c_n^2)_{n \in \mathbb{N}}$ satisfying $c_{n+1}^2 = f(c_n^2)$. Finally, we show that this iterative procedure recovers the true SB coupling $\Pi^{\mathrm{SB}}_{0,1} = \mathrm{N}(0, \Sigma^{\mathrm{SB}})$ as a fixed point. The formula of $\Sigma^{\mathrm{SB}}$ is given e.g. in (Bunne et al., 2023, Equation (2)) which we use below.

**Proposition 17.** *Let $\Pi^{\mathrm{SB}}_{0,1} = \mathrm{N}(0, \Sigma^{\mathrm{SB}})$ be the true static SB solution, with*

$$\Sigma^{\mathrm{SB}} = (1/2\beta^2) \begin{pmatrix} 1 & c_{\mathrm{SB}}^2 \\ c_{\mathrm{SB}}^2 & 1 \end{pmatrix}, \qquad c_{\mathrm{SB}}^2 = \frac{1}{2}(\sqrt{4 + \bar{\sigma}^4} - \bar{\sigma}^2).$$

*Then $\Pi^{\mathrm{SB}}_{0,1}$ is a fixed point of the iterative procedure in Proposition 16, i.e. $f(c_{\mathrm{SB}}^2) = c_{\mathrm{SB}}^2$.*

*Proof.* By straightforward calculations, $p_{\mathrm{SB}} = 2c_{\mathrm{SB}}^2 + \bar{\sigma}^2 = \sqrt{4 + \bar{\sigma}^4}$. If $\bar{\sigma} = 0$, $p_{\mathrm{SB}} = 2$ and thus $G(1, c_{\mathrm{SB}}^2) = -\bar{\sigma}^2/2 = 0$. Hence $f(c_{\mathrm{SB}}^2) = \exp(G(1, c_{\mathrm{SB}}^2)) = 1 = c_{\mathrm{SB}}^2$. If $\bar{\sigma} > 0$, we have $p_{\mathrm{SB}} > 2$ and $G(1, c_{\mathrm{SB}}^2) = -\tanh^{-1}(\bar{\sigma}^2/\sqrt{4 + \bar{\sigma}^4})$. Hence, $f(c_{\mathrm{SB}}^2) = \exp(G(1, c_{\mathrm{SB}}^2)) = \frac{1}{2}(\sqrt{4 + \bar{\sigma}^4} - \bar{\sigma}^2) = c_{\mathrm{SB}}^2$. We thus correctly recover the true static SB solution $\Pi^{\mathrm{SB}}_{0,1}$ as fixed point of IMF. $\square$

We visualize the convergence of this fixed point procedure for a variety of parameter settings in Figure 12. The convergence appears to be very fast in only two or three iterations.

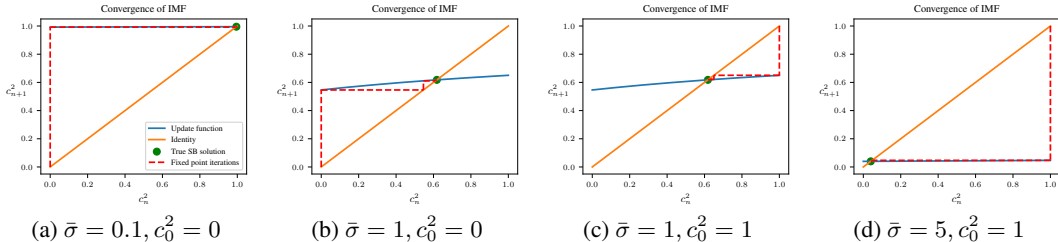

(a) $\bar{\sigma} = 0.1, c_0^2 = 0$    (b) $\bar{\sigma} = 1, c_0^2 = 0$    (c) $\bar{\sigma} = 1, c_0^2 = 1$    (d) $\bar{\sigma} = 5, c_0^2 = 1$

Figure 12: Convergence of IMF in the analytic case given by Proposition 16.

# E    Discrete-Time Markovian Projection

We derive in this section a discrete-time version of the Markovian projection and show that, in some limiting case, we recover the continuous-time projection. In the discrete case, we let

$$\pi(x_{0:N}) = \pi(x_0, x_N) \textstyle\prod_{k=0}^{N-1} q_{k+1|0,k,N}(x_{k+1}|x_0, x_k, x_N)$$
$$= \pi(x_0, x_N) \textstyle\prod_{k=0}^{N-2} q_{k+1|k,N}(x_{k+1}|x_k, x_N).$$

We consider a Markovian measure $p$ given by $p(x_{0:N}) = p(x_0) \prod_{k=0}^{N-1} p_{k+1|k}(x_{k+1}|x_k)$. Now let us compute $\mathrm{KL}(\pi|p)$. We have

$$\mathrm{KL}(\pi(x_{0:N})|p(x_{0:N})) = \textstyle\sum_{k=0}^{N-2} \int_{\mathbb{R}^d \times \mathbb{R}^d} \mathrm{KL}(q_{k+1|k,N}|p_{k+1|k})\pi_{k,N}(x_k, x_N)\mathrm{d}x_k\mathrm{d}x_N$$
$$+ \mathrm{KL}(\pi_0|p_0) + \int_{\mathbb{R}^d} \mathrm{KL}(\pi(x_N|x_0)|p(x_N|x_{N-1}))\pi(x_0, x_{N-1})\mathrm{d}x_0\mathrm{d}x_{N-1}.$$

In what follows, we denote

$$\mathcal{L}_0 = \mathrm{KL}(\pi_0|p_0), \mathcal{L}_N = \textstyle\int_{\mathbb{R}^d} \mathrm{KL}(\pi(x_N|x_0)|p(x_N|x_{N-1}))\pi(x_0, x_{N-1})\mathrm{d}x_0\mathrm{d}x_{N-1},$$
$$\mathcal{L}_{k+1} = \textstyle\int_{\mathbb{R}^d \times \mathbb{R}^d} \mathrm{KL}(q_{k+1|k,N}|p_{k+1|k})\pi_{k,N}(x_k, x_N)\mathrm{d}x_k\mathrm{d}x_N,$$

We have the following proposition.

**Proposition 18.** *The minimizer $p_{k+1|k}$ of $\mathcal{L}_{k+1}$ is given by*

$$p_{k+1|k}(x_{k+1}|x_k) = \textstyle\int_{\mathbb{R}^d} q_{k+1|k,N}(x_{k+1}|x_k, x_N)\pi_{N|k}(x_N|x_k)\mathrm{d}x_N. \tag{41}$$

*If $p_0 = q_0$, then for any $k \in \{0, \dots, N-1\}$, $p_k = \pi_k$. In addition, assume that $p_{k+1|k}(x_{k+1}|x_k) = \exp[-\|x_{k+1} - x_k - \gamma f(x_k)\|^2/(2\gamma)]/(2\pi\gamma)^{-d/2}$ and $q_{k+1|k,N}(x_{k+1}|x_k, x_N) = \exp[-\|x_{k+1} - x_k - \gamma f(x_k, x_N)\|^2/(2\gamma)]/(2\pi\gamma)^{d/2}$. Finally, assume that $\|x_{k+1} - x_k\| \le \gamma^{1/2}$. Then, we have that*

$$f(x_k) = \textstyle\int_{\mathbb{R}^d} f(x_k, x_N)\pi(x_N|x_k)\mathrm{d}x_N + o(\gamma^{1/2}). \tag{42}$$

*Proof.* The proofs of (41) and (42) are straightforward and left to the reader. We now prove that if $p_0 = \pi_0$, then for any $k \in \{1, \dots, N\}$, $p_k = \pi_k$. First, we have that for any $k \in \{0, \dots, N-1\}$,

$$\pi(x_k, x_{k+1}, x_N) = \pi(x_k, x_N)q(x_{k+1}|x_k, x_N).$$

Assume now that $p_k = \pi_k$, then we have

$$p_{k+1}(x_{k+1}) = \textstyle\int_{\mathbb{R}^d} p_k(x_k)p_{k+1|k}(x_{k+1}|x_k)\mathrm{d}x_k$$
$$= \textstyle\int_{\mathbb{R}^d \times \mathbb{R}^d} p_k(x_k)q_{k+1|k,N}(x_{k+1}|x_k, x_N)\pi_{N|k}(x_N|x_k)\mathrm{d}x_k\mathrm{d}x_N$$
$$= \textstyle\int_{\mathbb{R}^d \times \mathbb{R}^d} \pi_k(x_k)q_{k+1|k,N}(x_{k+1}|x_k, x_N)\pi_{N|k}(x_N|x_k)\mathrm{d}x_k\mathrm{d}x_N$$
$$= \textstyle\int_{\mathbb{R}^d \times \mathbb{R}^d} \pi_{k,k+1,N}(x_k, x_{k+1}, x_N)\mathrm{d}x_k\mathrm{d}x_N = \pi_{k+1}(x_{k+1}),$$

which concludes the proof. $\qquad\square$

In particular, in the previous proposition, if $f(x_k, x_N) = \nabla \log q(x_N|x_k)$, i.e. we have a discretization of the bridge then $f(x_k) = \int_{\mathbb{R}^d} \nabla \log q(x_N|x_K)\pi(x_N|x_k)\mathrm{d}x_N$, which recovers the Markovian projection in continuous-time.

# F    Comparing DSBM-IPF and DSB

We analyze further the differences between DSBM-IPF proposed here and DSB proposed in De Bortoli et al. (2021) and related algorithms in Vargas et al. (2021); Chen et al. (2022). All algorithms solve the SB problem using the IPF iterates. However, DSB-type algorithms solve for the IPF iterates using *time-reversals*, whereas DSBM solves for the iterates using *Markovian* and *reciprocal projections*. A comparison between these two methodologies is made in Section 4.

We investigate here further benefit (iii) of DSBM in Section 4, i.e. the benefit of explicitly projecting onto the reciprocal class of $\mathbb{Q}$. Intuitively speaking, we directly incorporate the reference measure $\mathbb{Q}$ in the training procedure as our inductive bias. More formally, suppose we have at the current

IPF iteration $\mathbb{M}^{2n}$ (Markov diffusion in the forward direction) and want to learn $\mathbb{M}^{2n+1}$ (Markov diffusion in the backward direction). Due to training error and the *forgetting* issue (Fernandes et al., 2021), however, $\mathbb{M}^{2n}$ no longer has the correct bridge $\mathbb{Q}_{|0,T}$. Now suppose we first perform IMF for $\mathbb{M}^{2n}$ and learn $\mathbb{M}^{2n,\star}$ in the forward direction. That is to say, we repeat alternative reciprocal and Markovian projections and obtain a sequence $(\mathbb{M}^{2n,m})_{m\in\mathbb{N}}$ in the forward direction converging to $\mathbb{M}^{2n,\star}$. Then $\mathbb{M}^{2n,\star}$ now has the correct bridge $\mathbb{Q}_{|0,T}$ by Proposition 7, since $\mathbb{M}^{2n,\star}$ is the SB between $\mathbb{M}_0^{2n}$ and $\mathbb{M}_T^{2n}$. Theoretically, $\mathbb{M}_t^{2n} = \mathbb{M}_t^{2n,\star}$ for $t = 0, T$, but due to training error accumulating it may be that $\mathbb{M}_T^{2n} \neq \mathbb{M}_T^{2n,\star}$. However, $\mathbb{M}_0^{2n} = \mathbb{M}_0^{2n,\star}$, since $\mathbb{M}^{2n,\star}$ is in the forward direction and starting from samples from $\pi_0$. As a result, we can obtain a Markov forward diffusion $\mathbb{M}^{2n,\star}$, which has $\mathbb{M}_0^{2n,\star} = \pi_0$ and the correct bridge $\mathbb{Q}_{|0,T}$. These are the same set of properties that the reference measure $\mathbb{Q}$ has. As a result, replacing $\mathbb{Q}$ with $\mathbb{M}^{2n,\star}$ in (6) results in the same SB solution. Consequently, now continuing the IPF iterations from $\mathbb{M}_0^{2n,\star}$, it is as if we restart IPF afresh using a modified SB problem

$$\mathbb{P}^{\mathrm{SB}} = \mathrm{argmin}\{\mathrm{KL}(\mathbb{P}|\mathbb{M}^{2n,\star}) \ : \ \mathbb{P}_0 = \pi_0, \ \mathbb{P}_T = \pi_T\}.$$

If $\mathbb{M}^{2n,\star}$ is closer than $\mathbb{Q}$ to $\mathbb{M}^\star$ in the sense of KL divergence, then we obtain a better initialization of the IPF procedure. As proposed in Algorithm 1, DSBM performs the Markovian and reciprocal projection only once before switching between the forward and backward directions. However, it is still beneficial compared to DSB with less bias accumulation in the bridge.

Algorithmically, one main difference between DSB-type algorithms and DSBM due to the above distinction occurs in the trajectory caching step. In DSB, a fixed discretization of SDE needs to be chosen, and all intermediate samples from the discretized Euler-Maruyama simulation of the SDE need to be saved. Furthermore, a second set of drift evaluation needs to be performed for all datapoints in the trajectory (De Bortoli et al., 2021, Equations (12), (13)). The IPML algorithm in Vargas et al. (2021) is also similar to DSB, but Gaussian processes are used to fit the drifts of forward and backward SDEs instead of neural networks. In Chen et al. (2022), the implicit score matching loss is used instead, but all intermediate points in the SDE trajectory also need to be saved. On the contrary, DSBM does not require intermediate samples during trajectory caching and only retains the joint samples at times $0, T$. Then, the intermediate trajectories are reconstructed using the reference bridge $\mathbb{Q}_{|0,T}$.

## G   Joint Training of Forward and Backward Processes

We recall below the DSBM algorithm given in Algorithm 1.

---
**Algorithm 2** Diffusion Schrödinger Bridge Matching

---
1: **Input:** Joint distribution $\Pi_{0,T}^0$, tractable bridge $\mathbb{Q}_{|0,T}$, number of outer iterations $N \in \mathbb{N}$.
2: Let $\Pi^0 = \Pi_{0,T}^0 \mathbb{Q}_{|0,T}$.
3: **for** $n \in \{0, \ldots, N-1\}$ **do**
4:    Learn $v_{\phi^\star}$ using (14) with $\Pi = \Pi^{2n}$.
5:    Let $\mathbb{M}^{2n+1}$ be given by (13).
6:    Let $\Pi^{2n+1} = \mathbb{M}_{0,T}^{2n+1}\mathbb{Q}_{|0,T}$.
7:    Learn $v_{\theta^\star}$ using (10) with $\Pi = \Pi^{2n+1}$.
8:    Let $\mathbb{M}^{2n+2}$ be given by (9).
9:    Let $\Pi^{2n+2} = \mathbb{M}_{0,T}^{2n+2}\mathbb{Q}_{|0,T}$.
10: **end for**
11: **Output:** $v_{\theta^\star}, v_{\phi^\star}$

---

Our main observation comes from Proposition 9. In particular, under mild assumptions, we have that the Markovian projection $\mathbb{M} = \mathrm{proj}_{\mathcal{M}}(\Pi)$ is associated with

$$\mathrm{d}\mathbf{X}_t = \{f_t(\mathbf{X}_t) + \sigma_t^2 \mathbb{E}_{\Pi_{T|t}}[\nabla \log \mathbb{Q}_{T|t}(\mathbf{X}_T|\mathbf{X}_t) \mid \mathbf{X}_t]\}\mathrm{d}t + \sigma_t \mathrm{d}\mathbf{B}_t, \quad \mathbf{X}_0 \sim \pi_0,$$
$$\mathrm{d}\mathbf{Y}_t = \{-f_{T-t}(\mathbf{Y}_t) + \sigma_{T-t}^2 \mathbb{E}_{\Pi_{0|T-t}}[\nabla \log \mathbb{Q}_{T-t|0}(\mathbf{Y}_t|\mathbf{Y}_T) \mid \mathbf{Y}_t]\}\mathrm{d}t + \sigma_{T-t}\mathrm{d}\mathbf{B}_t, \quad \mathbf{Y}_0 \sim \pi_T.$$

Considering the following losses,

$$\theta^\star = \mathrm{argmin}_\theta\{\int_0^T \mathbb{E}_{\Pi_{t,T}}[\|\sigma_t^2\nabla\log\mathbb{Q}_{T|t}(\mathbf{X}_T|\mathbf{X}_t) - v_\theta(t,\mathbf{X}_t)\|^2]/\sigma_t^2\mathrm{d}t \ : \ \theta\in\Theta\}, \quad (43)$$

$$\phi^\star = \mathrm{argmin}_\phi\{\int_0^T \mathbb{E}_{\Pi_{t,0}}[\|\sigma_t^2\nabla\log\mathbb{Q}_{t|0}(\mathbf{X}_t|\mathbf{X}_0) - v_\phi(t,\mathbf{X}_t)\|^2]/\sigma_t^2\mathrm{d}t \ : \ \phi\in\Phi\}. \quad (44)$$

If the families of functions $\{v_\theta \ : \ \theta\in\Theta\}$ and $\{v_\phi \ : \ \theta\in\Phi\}$ are rich enough, we have for any $t\in[0,T]$ and $x_t\in\mathbb{R}^d$, $v_{\theta^\star}(t,x_t) = \sigma_t^2\mathbb{E}_{\Pi_{T|t}}[\nabla\log\mathbb{Q}_{T|t}(\mathbf{X}_T|\mathbf{X}_t) \mid \mathbf{X}_t = x_t]$ and $v_{\phi^\star}(t,x_t) = \sigma_t^2\mathbb{E}_{\Pi_{0|t}}[\nabla\log\mathbb{Q}_{t|0}(\mathbf{X}_t|\mathbf{X}_0) \mid \mathbf{X}_t = x_t]$. In practice, this means that the Markovian projection can be computed in a forward *or* backward fashion equivalently.

Therefore, given a coupling $\Pi = \Pi^{2n}$, we can update *both* $v_\theta$ and $v_\phi$. This means that we train the forward and backward model *jointly*. We then consider $\mathbb{M}_b^{2n+1}$ associated with (13) and $\mathbb{M}_f^{2n+1}$ associated with (9). Note that if the families of functions $\{v_\theta \ : \ \theta\in\Theta\}$ and $\{v_\phi \ : \ \theta\in\Phi\}$ are rich enough then $\mathbb{M}_f^{2n+1} = \mathbb{M}_b^{2n+1}$.

**Mixture from forward and backward.**  Once we have obtained both the forward update and the backward update, our next task is to define the new mixture of bridge $\Pi^{2n+1}$. In Algorithm 1, since we train only the *backward* model $\mathbb{M}^{2n+1} = \mathbb{M}_b^{2n+1}$, we define $\Pi^{2n+1} = \mathbb{M}_{0,T}^{2n+1}\mathbb{Q}_{|0,T}$. In the case of *joint training*, we have access to $\mathbb{M}_b^{2n+1}$ and $\mathbb{M}_f^{2n+1}$. One way to define a new mixture of bridge is to compute $\Pi^{2n+1} = \frac{1}{2}(\mathbb{M}_{b,0,T}^{2n+1}\mathbb{Q}_{|0,T} + \mathbb{M}_{f,0,T}^{2n+1})\mathbb{Q}_{|0,T}$. This choice ensures that in the case where $\mathbb{M}_f^{2n+1} = \mathbb{M}_b^{2n+1}$ we have

$$\Pi^{2n+1} = \mathbb{M}_{f,0,T}^{2n+1}\mathbb{Q}_{|0,T} = \mathbb{M}_{b,0,T}^{2n+1}\mathbb{Q}_{|0,T}.$$

It also ensures that all the steps in the *joint* DSBM training algorithms are symmetric. We leave the study of an optimal combination of $\mathbb{M}_f^{2n+1}$ and $\mathbb{M}_b^{2n+1}$ for future work.

**Consistency loss.**  In addition to the losses (43) and (44), we also consider an additional *consistency* loss. A similar idea was explored in Song (2022). In DSB (De Bortoli et al., 2021; Chen et al., 2022) and DSBM, see Algorithm 1, the processes parameterized by $v_\theta$ (forward) and $v_\phi$ backward are identical *only at equilibrium*. Thus imposing the forward and the backward processes match at each step of DSB or DSBM would lead to some bias. However, this is not the case in the *joint training* setting. Indeed, in that case, we have $\mathbb{M}_f^{2n+1} = \mathbb{M}_b^{2n+1}$ if the families are rich enough. Therefore, we get that

$$\mathrm{d}\mathbf{Y}_t = \{-f_{T-t}(\mathbf{Y}_t) + v_\phi(T - t, \mathbf{Y}_t)\}\mathrm{d}t + \sigma_{T-t}\mathrm{d}\mathbf{B}_t, \qquad \mathbf{Y}_0 \sim \pi_T, \qquad (45)$$

is the time reversal of

$$\mathrm{d}\mathbf{X}_t = \{f_t(\mathbf{X}_t) + v_\theta(t, \mathbf{X}_t)\}\mathrm{d}t + \sigma_t\mathrm{d}\mathbf{B}_t, \qquad \mathbf{X}_0 \sim \pi_0. \qquad (46)$$

Computing the time-reversal of (45), we have

$$\mathrm{d}\mathbf{X}_t = \{f_t(\mathbf{X}_t) - v_\phi(t, \mathbf{X}_t) + \sigma_t^2\nabla\log\Pi_t^{2n}(\mathbf{X}_t)\}\mathrm{d}t + \sigma_t\mathrm{d}\mathbf{B}_t, \qquad \mathbf{X}_0 \sim \pi_0. \qquad (47)$$

Identifying (47) and (46), we get that for any $t\in[0,T]$ and $x_t\in\mathbb{R}^d$

$$v_\theta(t, x_t) = -v_\phi(t, x_t) + \sigma_t^2\nabla\log\Pi_t^{2n}(x_t). \qquad (48)$$

We highlight that letting $\sigma_t \to 0$ for any $t\in[0,T]$, we get that $v_\theta = -v_\phi$, which confirms that the time-reversal of an ODE is simply given by flipping the sign of the velocity. Therefore, we propose the following loss which links the parameters $\theta$ and $\phi$

$$\mathcal{L}_{\mathrm{cons}}(\theta,\phi) = \int_0^T \mathbb{E}_{\Pi_t^{2n}}[\|v_\theta(t,\mathbf{X}_t) + v_\phi(t,\mathbf{X}_t) - \sigma_t^2\nabla\log\Pi_t^{2n}(\mathbf{X}_t)\|^2]/\sigma_t^2\mathrm{d}t.$$

Leveraging tools from implicit score matching (Hyvärinen, 2005) and the divergence theorem, we get that

$$\mathcal{L}_{\mathrm{cons}}(\theta,\phi) = \int_0^T \mathbb{E}_{\Pi_t^{2n}}[\|v_\theta(t,\mathbf{X}_t) + v_\phi(t,\mathbf{X}_t)\|^2/\sigma_t^2 + 2\mathrm{div}(v_\theta(t,\mathbf{X}_t) + v_\phi(t,\mathbf{X}_t))]\mathrm{d}t + C,$$

where $C \geq 0$ is a constant which does not depend on $\theta$ and $\phi$. Alternatively, (48) shows that

$$\nabla\log\Pi_t^{2n}(x_t) = \mathbb{E}_{\Pi_{T|t}^{2n}}[\nabla\log\mathbb{Q}_{T|t}|\mathbf{X}_t = x_t] + \mathbb{E}_{\Pi_{0|t}^{2n}}[\nabla\log\mathbb{Q}_{t|0}|\mathbf{X}_t = x_t].$$

We thus also have the following denoising score matching consistency loss

$$\mathcal{L}_{\mathrm{cons}}(\theta,\phi) = \int_0^T \mathbb{E}_{\Pi_{0,t,T}^{2n}}[\|v_\theta(t,\mathbf{X}_t)+v_\phi(t,\mathbf{X}_t)-\sigma_t^2(\nabla\log\mathbb{Q}_{T|t}(\mathbf{X}_T|\mathbf{X}_t)+\nabla\log\mathbb{Q}_{t|0}(\mathbf{X}_t|\mathbf{X}_0))\|^2]/\sigma_t^2\mathrm{d}t,$$

The advantage of this DSM loss is that it does not rely on any divergence computation.

Below, we recall the two losses used to estimate the Markovian projection (43) and (44)

$$\mathcal{L}(\theta) = \int_0^T \mathbb{E}_{\Pi_{t,T}}[\|\sigma_t^2\nabla\log\mathbb{Q}_{T|t}(\mathbf{X}_T|\mathbf{X}_t) - v_\theta(t,\mathbf{X}_t)\|^2]/\sigma_t^2\mathrm{d}t,$$
$$\mathcal{L}(\phi) = \int_0^T \mathbb{E}_{\Pi_{t,0}}[\|\sigma_t^2\nabla\log\mathbb{Q}_{t|0}(\mathbf{X}_t|\mathbf{X}_0) - v_\phi(t,\mathbf{X}_t)\|^2]/\sigma_t^2\mathrm{d}t.$$

The complete loss we consider in the joint training of the algorithm is of the form

$$\mathcal{L}_\lambda(\theta,\phi) = \mathcal{L}(\theta) + \mathcal{L}(\phi) + \lambda\mathcal{L}_{\mathrm{cons}}(\theta,\phi), \tag{49}$$

where $\lambda > 0$ is an additional regularization parameter. We now state a version of DSBM which performs *joint training* in Algorithm 3.

---

**Algorithm 3** Diffusion Schrödinger Bridge Matching (Joint Training)

---

1: **Input:** Coupling $\Pi_{0,T}^0$, tractable bridge $\mathbb{Q}_{|0,T}$, $N \in \mathbb{N}$
2: Let $\Pi^0 = \Pi_{0,T}^0\mathbb{Q}_{|0,T}$.
3: **for** $n \in \{0,\dots,N-1\}$ **do**
4:     Learn $v_{\phi^\star}, v_{\theta^\star}$ using (49) with $\Pi = \Pi^n$.
5:     Let $\mathbb{M}_f^{n+1}$ be given by (9).
6:     Let $\mathbb{M}_b^{n+1}$ be given by (13).
7:     Let $\mathbb{M}^{n+1} = \frac{1}{2}(\mathbb{M}_f^{n+1} + \mathbb{M}_b^{n+1})$.
8:     Let $\Pi^{n+1} = \mathbb{M}_{0,T}^{n+1}\mathbb{Q}_{|0,T}$.
9: **end for**
10: **Output:** $v_{\theta^\star}, v_{\phi^\star}$

---

## H   Loss Scaling

Similar to the loss weighting in standard diffusion models (Song et al., 2021b; Ho et al., 2020), we derive a similar weighting to reduce the variance of our objective. We focus on the forward direction of Markovian projection in this case, and the backward case can be derived similarly. Our forward loss in the DSBM framework is given by (10), where the inner expectation is given by

$$\mathbb{E}_{\Pi_{t,T}}[\|\sigma_t^2\nabla\log\mathbb{Q}_{T|t}(\mathbf{X}_T|\mathbf{X}_t) - v_\theta(t,\mathbf{X}_t)\|^2].$$

Letting $\mathbb{Q}_{|0,T}$ be a Brownian bridge with diffusion parameter $\sigma$ and assuming $T = 1$, this becomes

$$\mathbb{E}_{(\mathbf{X}_0,\mathbf{X}_1)\sim\Pi_{0,1},\mathbf{Z}\sim\mathcal{N}(0,\mathrm{Id})}[\|\mathbf{X}_1 - \mathbf{X}_0 - \sigma\sqrt{t/(1-t)}\mathbf{Z} - v_\theta(t,\mathbf{X}_t^{0,T})\|^2]$$

with $\mathbf{X}_t^{0,T} = t\mathbf{X}_1 + (1-t)\mathbf{X}_0 + \sigma\sqrt{t(1-t)}\mathbf{Z}$. When $t \approx 1$, we see that the regression target is dominated by the noise term $\sigma\sqrt{t/(1-t)}\mathbf{Z}$ which needs to be predicted based on information contained within $\mathbf{X}_t^{0,T}$. The loss will have an approximate scale of $\sigma^2 t/(1-t)$ when $t \approx 1$ which will be very large. To avoid these large values affecting gradient descent, we can downweight the loss by $1 + \sigma^2 t/(1-t)$ (we can add 1 to effectively cause no loss scaling when $t$ is close to 0)

$$(1 + \sigma^2 t/(1-t))^{-1}\mathbb{E}_{(\mathbf{X}_0,\mathbf{X}_1)\sim\Pi_{0,1},\mathbf{Z}\sim\mathcal{N}(0,\mathrm{Id})}[\|\mathbf{X}_1 - \mathbf{X}_0 - \sigma\sqrt{t/(1-t)}\mathbf{Z} - v_\theta(t,\mathbf{X}_t^{0,T})\|^2].$$

Similar arguments can be applied to the backward loss (14)

$$\mathbb{E}_{\Pi_{t,0}}[\|\sigma_t^2\nabla\log\mathbb{Q}_{t|0}(\mathbf{X}_t|\mathbf{X}_0) - v_\phi(t,\mathbf{X}_t)\|^2]$$
$$= \mathbb{E}_{(\mathbf{X}_0,\mathbf{X}_1)\sim\Pi_{0,1},\mathbf{Z}\sim\mathcal{N}(0,\mathrm{Id})}[\|\mathbf{X}_0 - \mathbf{X}_1 - \sigma\sqrt{(1-t)/t}\mathbf{Z} - v_\phi(t,\mathbf{X}_t^{0,T})\|^2],$$

which we then downweight by $1 + \sigma^2(1-t)/t$

$$(1 + \sigma^2(1-t)/t)^{-1}\mathbb{E}_{(\mathbf{X}_0,\mathbf{X}_1)\sim\Pi_{0,1},\mathbf{Z}\sim\mathcal{N}(0,\mathrm{Id})}[\|\mathbf{X}_0 - \mathbf{X}_1 - \sigma\sqrt{(1-t)/t}\mathbf{Z} - v_\phi(t,\mathbf{X}_t^{0,T})\|^2].$$

# I Experiments

In this section, we present further details of the experiment setups as well as further experiment results. In all experiments, we use Brownian motion for the reference measure $\mathbb{Q}$ with corresponding Brownian bridge (4) and $T = 1$. We use the Adam optimizer with learning rate $10^{-4}$ and SiLU activations unless specified otherwise. The experiments are run on computing clusters with a mixture of both CPU and GPU resources.

## I.1 2D Experiments

For the 2D experiments, we closely follow Tong et al. (2023) and the released code[7], and use the same synthetic datasets and the 2-Wasserstein distance between the test set and samples simulated using probability flow ODE as the evaluation metric. However, we use 10000 samples in the test set since we find the 2-Wasserstein distance can vary greatly with only 1000 samples (which can be as high as 0.3 even between two set of samples both drawn from the ground truth distribution). We use a simple MLP with 3 hidden layers and 256 hidden units to parameterize the forward and backward drift networks. We use batch size 128 and 20 diffusion steps with uniform schedule at sampling time. Each outer iteration is trained for 10000 steps and we train for 20 outer iterations. As the initial coupling is more optimal in DSBM-IMF+, we reduce the number of outer iterations to 4 and train for 50000 steps in each outer iteration. For flow methods, we train for 200000 steps in total. For Table 2 we use $\sigma_t = 1$ in all cases, except the *moons-8gaussians* dataset where we use $\sigma_t = 5$. Note that FM cannot be used for the *moons-8gaussians* task since it requires a Gaussian source, but CFM is applicable. The experiments are run using 1 CPU and take approximately 200 minutes (for both training and testing).

### I.1.1 Variance of the reference measure $\mathbb{Q}$

We comment further on the effect of $\sigma_t$ in the reference path measure $\mathbb{Q}$. We assume a time-homogeneous $\sigma_t = \sigma$ for simplicity. In Figure 2, we vary $\sigma$ and visualize the learned transport for a *3gaussians* problem of transporting between two Gaussian mixtures. In Table 4 we show the 2-Wasserstein distance between the test set and generated samples for this *3gaussians* problem as well as the *moons-8gaussians* problem. We find that large values of $\sigma$ result in increasingly curved transport paths, and correspondingly reduced performance when $\sigma$ is excessively large. Conversely, we also find reduced performance when $\sigma$ is excessively small. We conjecture this is due to increased optimization difficulty and bias accumulation. Firstly, the EOT problem becomes more difficult to solve as $\sigma$ is taken to 0, which would require a higher number of outer iterations. Further, the introduction of noise also decreases optimization difficulty by smoothing the intermediate marginals between the two terminal distributions. The benefit of setting $\sigma > 0$ and using a stochastic sampler was also observed in Albergo et al. (2023); Delbracio and Milanfar (2023). Finally, we conjecture setting $\sigma > 0$ could also increase the diversity of sampled couplings and may alleviate some bias accumulation issues in the outer iterations. When $\sigma = 0$, these issues result in the artifacts observed in the transferred samples (marked using yellow points) in Figure 2. The appropriate value for $\sigma$ depends on the spatial scaling of the problem as shown in Table 4, where the optimum $\sigma$ is larger for the larger scale *moons-8gaussians* problem.

| 3gaussians | | moons-8gaussians | |
|---|---|---|---|
| $\sigma$ | *2-Wasserstein* | $\sigma$ | *2-Wasserstein* |
| $\sigma = 0$ | 0.646±0.028 | $\sigma = 0$ | 1.459±0.008 |
| $\sigma = 0.1$ | 0.724±0.039 | $\sigma = 1.0$ | 1.285±0.346 |
| $\sigma = 0.3$ | 0.546±0.169 | $\sigma = 2.0$ | *0.916±0.292* |
| $\sigma = 1.0$ | **0.439±0.072** | $\sigma = 4.0$ | **0.818±0.249** |
| $\sigma = 3.0$ | *0.543±0.078* | $\sigma = 8.0$ | 0.989±0.179 |

Table 4: 2-Wasserstein distance for varying value of $\sigma$ used in the DSBM-IMF method.

---

[7] https://github.com/atong01/conditional-flow-matching (code released under MIT license)

## I.2 Gaussian Experiment

Similar to the 2D experiments, we use a simple MLP with 2 hidden layers and 256 hidden units to parameterize the forward and backward drift networks. This is a smaller network compared to the "large" network in De Bortoli et al. (2021). We use batch size 128 and 20 diffusion sampling steps with uniform time schedule at inference time. Each outer iteration is trained for 10000 steps and we train for 20 outer iterations. The experiments are also run using 1 CPU, and for the case $d = 50$ finish in approximately 200 minutes. We note that DSB, IMF-b, and DSBM methods all solve for the Schrödinger Bridge transport, whereas Rectified Flow does not and so is not plotted on the covariance convergence plot in Figure 3 since it is not comparable.

For Table 3, we assume the marginals of the learned process $\mathbb{P}_t$ are also independently Gaussian distributed in each dimension. We thus estimate the KL divergence from the sample mean and variance of each dimension of $\mathbb{P}_t$ using the analytic KL formula between Gaussian distributions.

## I.3 MNIST Transfer Experiment

We follow De Bortoli et al. (2021) closely for the setup of this experiment. We use the set of first 5 letters of EMNIST (A-E, a-e) such that both domains have 10 classes in total. We use the same U-Net architecture, batch size 128 and 30 diffusion sampling steps. The network size is approximately 6.6 million parameters. Each outer iteration is trained for 5000 steps. We refresh the cache dataloader every 1000 steps with 10000 new samples. Contrary to De Bortoli et al. (2021), in our experiments we find that we obtain better sampling quality for both DSB and DSBM using a uniform noising schedule. We simply choose $\sigma_t = 1$ for all $t$ and $T = 1$ in our experiments.

For DSBM-IPF, we train for at most 50 outer iterations (i.e. 250000 total number of steps). For DSBM-IMF and Rectified Flow, since their first iteration corresponds to Bridge Matching and Flow Matching respectively, we first pretrain the forward and backward networks for 100000 steps using the Bridge Matching or Flow Matching losses. DSBM-IMF then switches to iterative training with 5000 steps per outer iteration, the same as DSBM-IPF. For Rectified Flow, we train for 50000 steps per outer iteration. The experiments are performed using 2 GPUs and take approximately one day.

We provide further experiment results in Figures 13, 14 and 15. In Figure 13, we show samples generated using different algorithms and at different points of convergence. Samples generated using CFM and Rectified Flow with 2 rectified steps in (a) and (b) appear to be less clear and identifiable. OT-CFM improves upon CFM slightly in (c), but many samples still appear to be unclear. For DSB, the algorithm has not converged after 10 iterations, and many samples in (d) still appear to be letters. After 20 iterations, there are still letter-like samples in Figure 4b, and the digit classes also appear to be unbalanced with many instances of digit '0'. After 30 iterations, however, the sample quality of DSB becomes very poor in (e). On the other hand, as shown in (f)-(j), we observe that DSBM-IPF converges faster than DSB, with more accurate samples even in iterations 10 and 20, and the sample quality continues to improve until the end of training after 50 iterations.

We present some additional trajectory samples at the end of training in Figures 14 and 15 in both the forward and backward directions. We observe DSBM is able to transfer samples across the two domains faithfully, and the output samples preserve interesting similarities compared to the input, whereas Bridge Matching preserves much less similarity. The Mean Squared Distance (MSD) between the initial and final samples also confirm that DSBM methods transfer more closely to the original inputs, and DSBM-IMF further achieves the best FID score out of the three methods.

## I.4 CelebA Transfer Experiment

For CelebA (Liu et al., 2015), we test DSBM on resized $64 \times 64$ and $128 \times 128$ resolution images. We evaluate the dependency of the results with respect to $\sigma > 0$ on CelebA $64 \times 64$. We also showcase the scalibility of our method by training DSBM on CelebA $128 \times 128$. In both cases, the dataset is split between `male/old` and `female/young`. We gather 20000 samples of each class on which we perform classical data augmentation such as horizontal flipping.

For our ablation study w.r.t. the value of $\sigma > 0$, we run DSBM for 20 iterations (note that the loss and the quality was still improving after 20 DSBM iterations but by stopping the runs early we were allowed to draw comparisons with more values of $\sigma$). We use a U-Net architecture with 4 resolution levels and 2 residual blocks per resolution level. The batch size is fixed to 64 and the EMA rate is

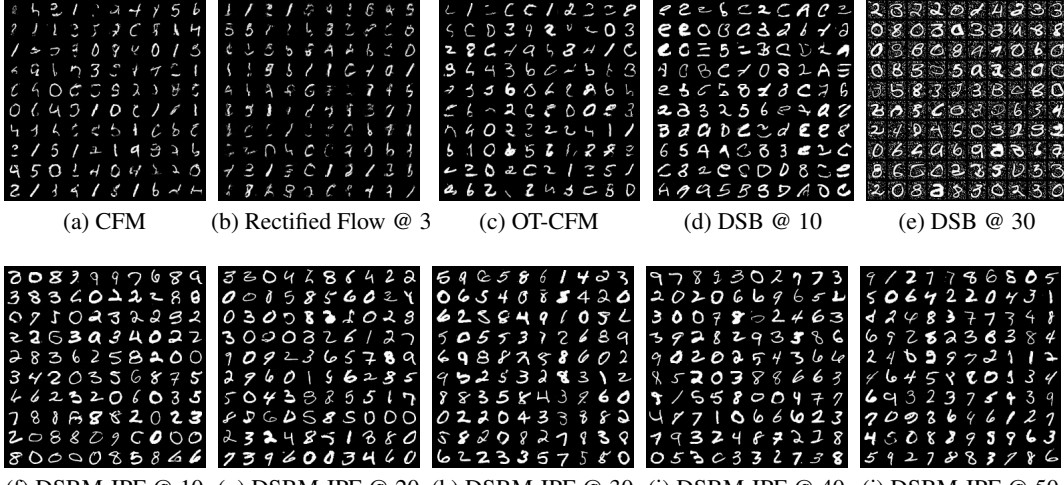

| (a) CFM | (b) Rectified Flow @ 3 | (c) OT-CFM | (d) DSB @ 10 | (e) DSB @ 30 |

(f) DSBM-IPF @ 10  (g) DSBM-IPF @ 20  (h) DSBM-IPF @ 30  (i) DSBM-IPF @ 40  (j) DSBM-IPF @ 50

Figure 13: Samples of MNIST digits transferred from the EMNIST letters using different methods. @ indicates the progressed number of outer iterations $n + 1$.

fixed to 0.999. We refresh the cache dataloader every 500 steps with 10000 new samples. The SDE sampler is chosen to be the modified Euler-Maruyama sampler, see Heng et al. (2021) for instance, with a constant schedule for the stepsizes. We use 100 sampling steps at inference time and to refresh the cache. For each outer DSBM iteration we train the model for 20000 iterations.

We provide additional transfer results in resolution $128 \times 128$ in Figure 16. We do not change the training setting for this experiment.

### I.5 AFHQ Transfer Experiment

For AFHQ (Choi et al., 2020), we test DSBM between classes cat and wild with $512 \times 512$ resolution images. Each class contains approximately 5000 samples. We first pretrain the networks using Bridge Matching for 100000 steps, then run DSBM for 20 iterations with 25000 steps per outer iteration. We follow Liu et al. (2023b) and use the same U-Net architecture[8]. The batch size is 4 and the EMA rate is 0.999. We choose $\sigma^2 = 5$ and again we use 100 sampling steps with constant stepsizes.

### I.6 CIFAR-10 Generative Modeling Experiment

We also test our method in the standard generative modeling framework on the CIFAR-10 dataset. We again use a U-Net architecture with 4 resolution levels and 2 residual blocks per resolution level. The network size is approximately 39.6 million parameters. The batch size is fixed to 128 and the EMA rate is fixed to 0.9999. The AdamW optimizer is used for this task. For DSBM-IMF and Rectified Flow, again we first pretrain the networks using the Bridge Matching or Flow Matching losses, then switch to DSBM-IMF or RF training with 100000 steps per outer iteration. We find that 1 or 2 additional outer iterations appear sufficiently effective on this task, and additional outer iterations can cause sample quality to drop. For our main experiment, the pretraining stage ran for approximately 6 days, and DSBM-IMF ran for approximately 4 additional days using 4 V100 GPUs.

The best results during training for each method are reported in Table 5 and Figure 17, where we compute the FID score between 50000 samples in the CIFAR-10 training set and 50000 generated samples following standard practice for this task. We observe that DSBM-IMF can clearly improve upon Bridge Matching at the same value of $\sigma$, which suggests that further outer iterations in DSBM is beneficial for improving sample quality. This is contrary to Rectified Flow, which causes the FID score to worsen compared to Flow Matching after only 1 rectified iteration. However, as $\sigma$ increases,

---

[8]https://github.com/gnobitab/RectifiedFlow/blob/main/ImageGeneration/configs/rectified_flow/afhq_cat_pytorch_rf_gaussian.py (code released under Apache-2.0 license)

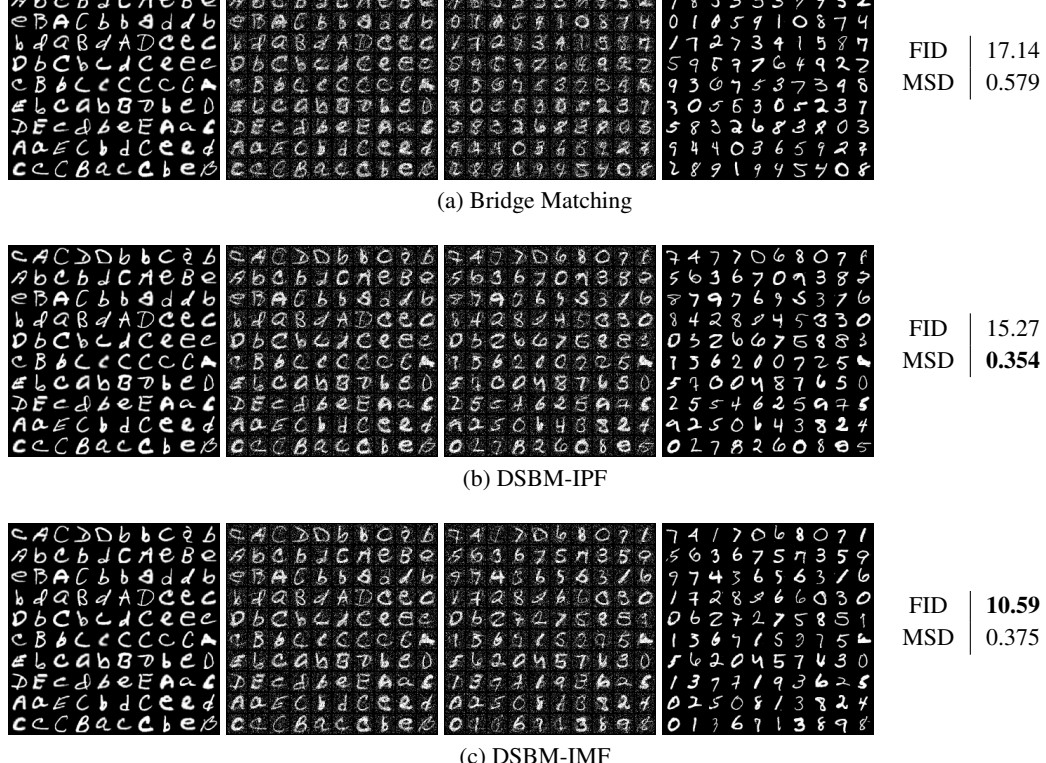

Figure 14: Left: EMNIST to MNIST sample trajectory with 30 diffusion steps at $t = 0, 1/3, 2/3, 1$. Right: FID score of final samples, and Mean Squared Distance between initial and final samples.

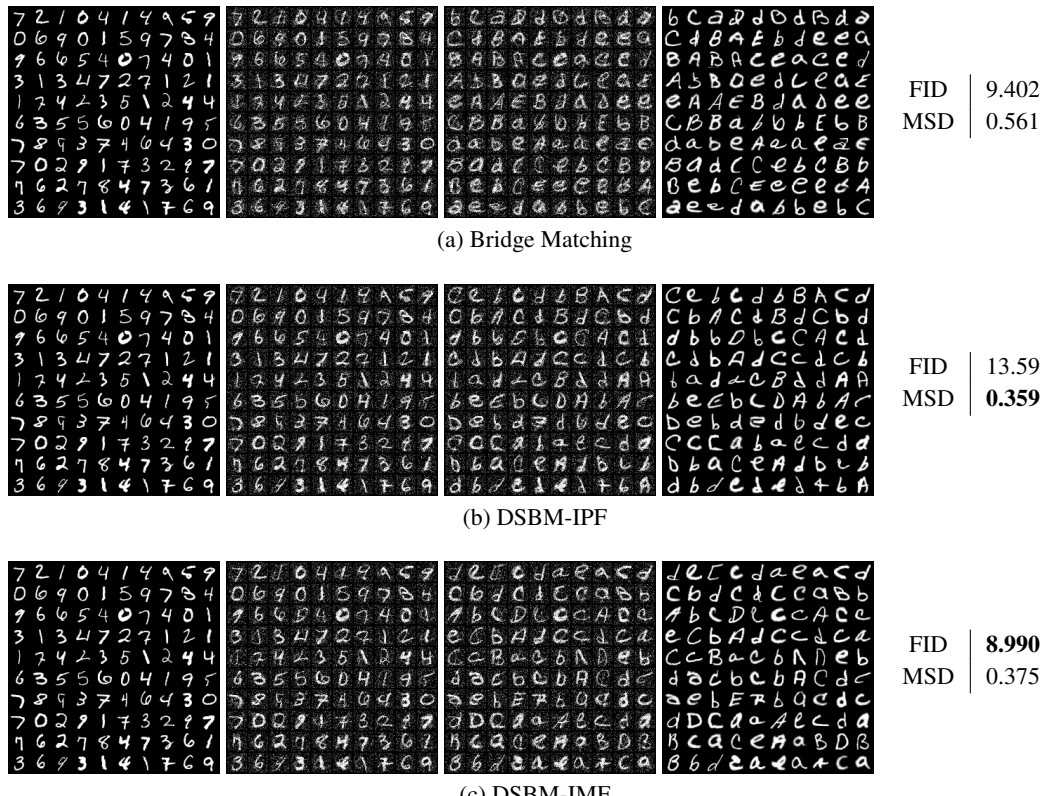

Figure 15: Left: MNIST to EMNIST sample trajectory with 30 diffusion steps at $t = 0, 1/3, 2/3, 1$. Right: FID score of final samples, and Mean Squared Distance between initial and final samples.

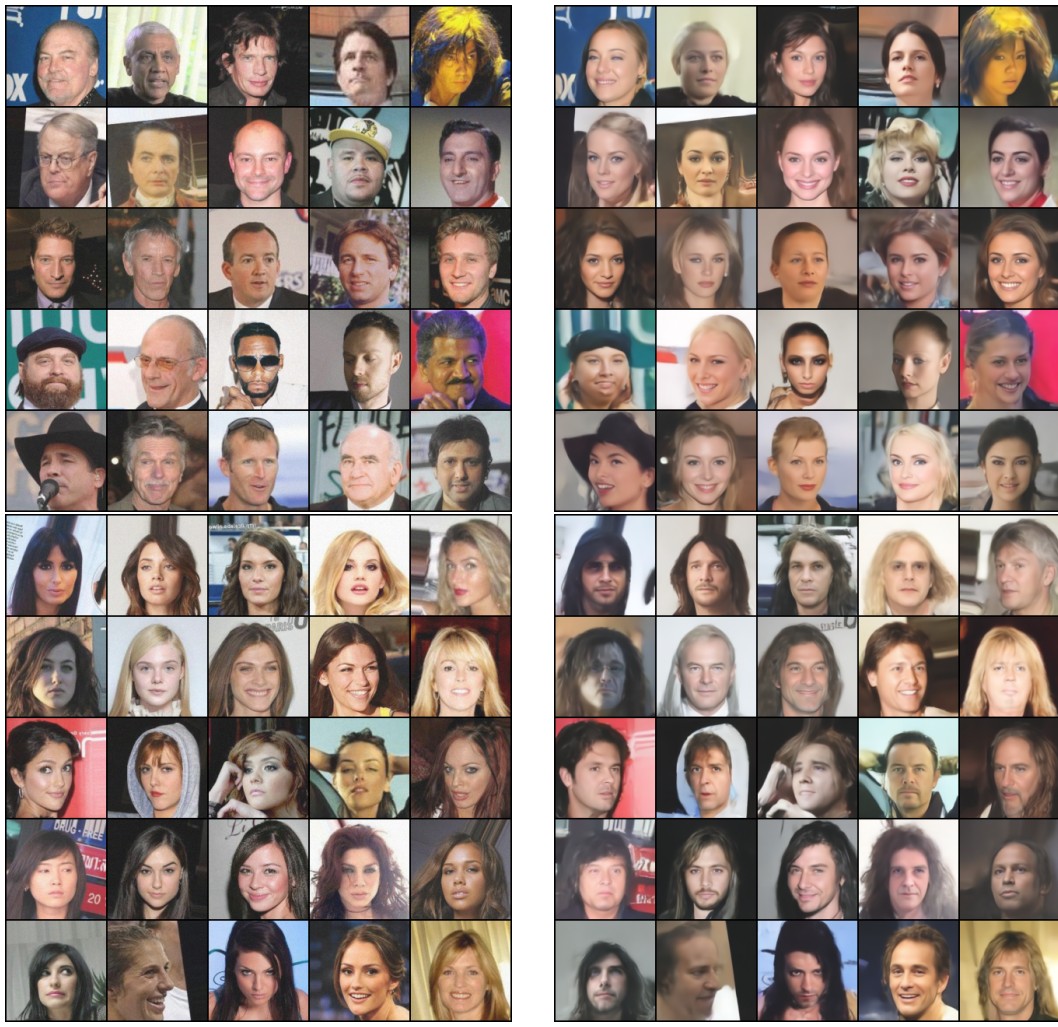

Figure 16: Transfer results between images given by the tokens `female/young` and `male/old`. Top row: original images (left) and generated images (right). Bottom row: original images (left) and generated images (right).

we observe the FID score worsens for both Bridge Matching and DSBM as more stochasticity is introduced in the sampler. The best result of DSBM-IMF is obtained using $\sigma^2 = 0.2$ and is slightly better than FM (i.e. with $\sigma^2 = 0$) using 100 Euler steps. On the other hand, using the dopri5 ODE solver, FM achieves a FID of 4.055 with on average 148 integration steps. In Figure 17, we observe that both RF and DSBM-IMF are very effective in improving sampling quality at low number of diffusion steps, i.e. low number of function evaluations (NFEs), compared to Bridge and Flow Matching as well as OT-CFM which improves upon CFM slightly. DSBM-IMF also achieves lower FID score than RF as the NFEs are taken higher. Additional strategies such as distillation and fast SDE solvers can also be useful for improving few-step sampling quality further.

## I.7 Fluid Flows Experiment

We use the fluid flows dataset[9] from Bischoff and Deck (2023). The dataset consists of unpaired low ($64 \times 64$) and high ($512 \times 512$) resolution fields, as well as a context field with local information for the high resolution field. The data fields consist of two channels representing supersaturation and vorticity. The context field is dependent on the wavenumber $k_x = k_y \in \{1, 2, 4, 8, 16\}$, which

---

[9] https://github.com/CliMA/CliMADatasets.jl (code released under MIT license)

| $\sigma^2$ | FM | RF |
|---|---|---|
| 0 | 4.931 | 6.010 |

| $\sigma^2$ | BM | DSBM-IMF |
|---|---|---|
| 0.2 | 5.427 | **4.511** |
| 0.5 | 8.274 | 6.896 |
| 1 | 12.749 | 9.881 |

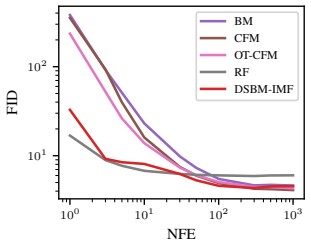
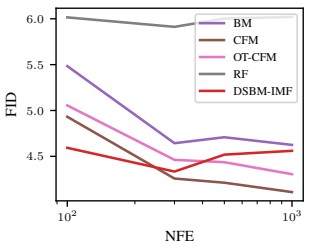

Table 5: FID results on the CIFAR-10 train set using 100 Euler(-Maruyama) steps.

Figure 17: FID vs number of diffusion steps (NFE) with NFE between 1 and 1000.

Figure 18: Zoomed-in version of Figure 17 with NFE between 100 and 1000.

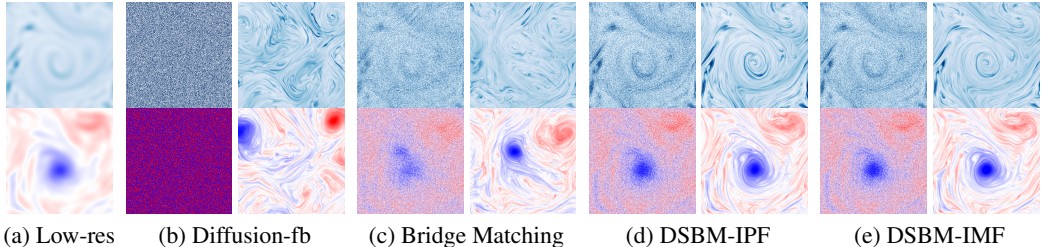

| (a) Low-res | (b) Diffusion-fb | (c) Bridge Matching | (d) DSBM-IPF | (e) DSBM-IMF |

Figure 19: (a) Source low resolution sample; (b)(c)(d)(e) intermediate state and final reconstruction of each algorithm, for wavenumber $k_x = k_y = 2$.

specifies the frequency of the saturation specific humidity modulation. We follow Bischoff and Deck (2023) for data processing and network architecture, which is given by a U-Net with an additional spatial mean bypass network given by an MLP. The network size is approximately 11.3 million parameters in total. We train using batch size 4, learning rate $2 \times 10^{-4}$, for 5000 steps per outer iteration and $N = 12$ outer iterations. We refresh the cache dataloader every 2500 steps with 1250 new samples. The training takes approximately 20 hours on a single RTX GPU. We used $\sigma^2 = 0.3$ and sample using 30 diffusion steps without finetuning these parameters. For the Diffusion-fb method in Bischoff and Deck (2023), we use the released code[10] without modifying any parameters.

We visualize intermediate and final reconstruction samples for different algorithms in Figure 19. We see that DSBM-IPF and DSBM-IMF provide consistent samples with the low resolution source, whereas Diffusion-fb and Bridge Matching produce dissimilar samples. We also follow Bischoff and Deck (2023) for a more refined statistical analysis in Figures 20, 21, 22. DSBM-IMF achieves comparable performance as Diffusion-fb in terms of these statistical profiles, and can be comparatively more accurate e.g. in the tails of the distributions in Figure 20, and for the case $k_x = k_y = 4$ in Figure 21 for which the power spectrum of supersaturation is correctly captured by DSBM-IMF but not by other methods. Comparing this analysis with Figure 11, DSBM-IMF is also significantly more accurate in terms of conditional consistency than Diffusion-fb. On the other hand, DSBM-IPF appears less accurate in terms of these unconditional statistics than Diffusion-fb and DSBM-IMF, but achieves lower $\ell_2$ distances from the input sources in Figure 11. This suggests that DSBM-IPF and DSBM-IMF exhibit different empirical biases before convergence, and DSBM-IMF is more preferable when the accuracy of the samples are important. This is in line with IMF theory as the marginals $\pi_0, \pi_T$ are preserved in IMF but not in IPF.

# J   Broader Impact

Our work focuses on theoretical and methodological research and is intended to bring closer the fields of generative modeling and optimal transport. It can be useful for learning transport maps between general distributions with high accuracy and high scalability, which can have useful applications

---

[10]https://github.com/CliMA/diffusion-bridge-downscaling (code released under Apache-2.0 license)

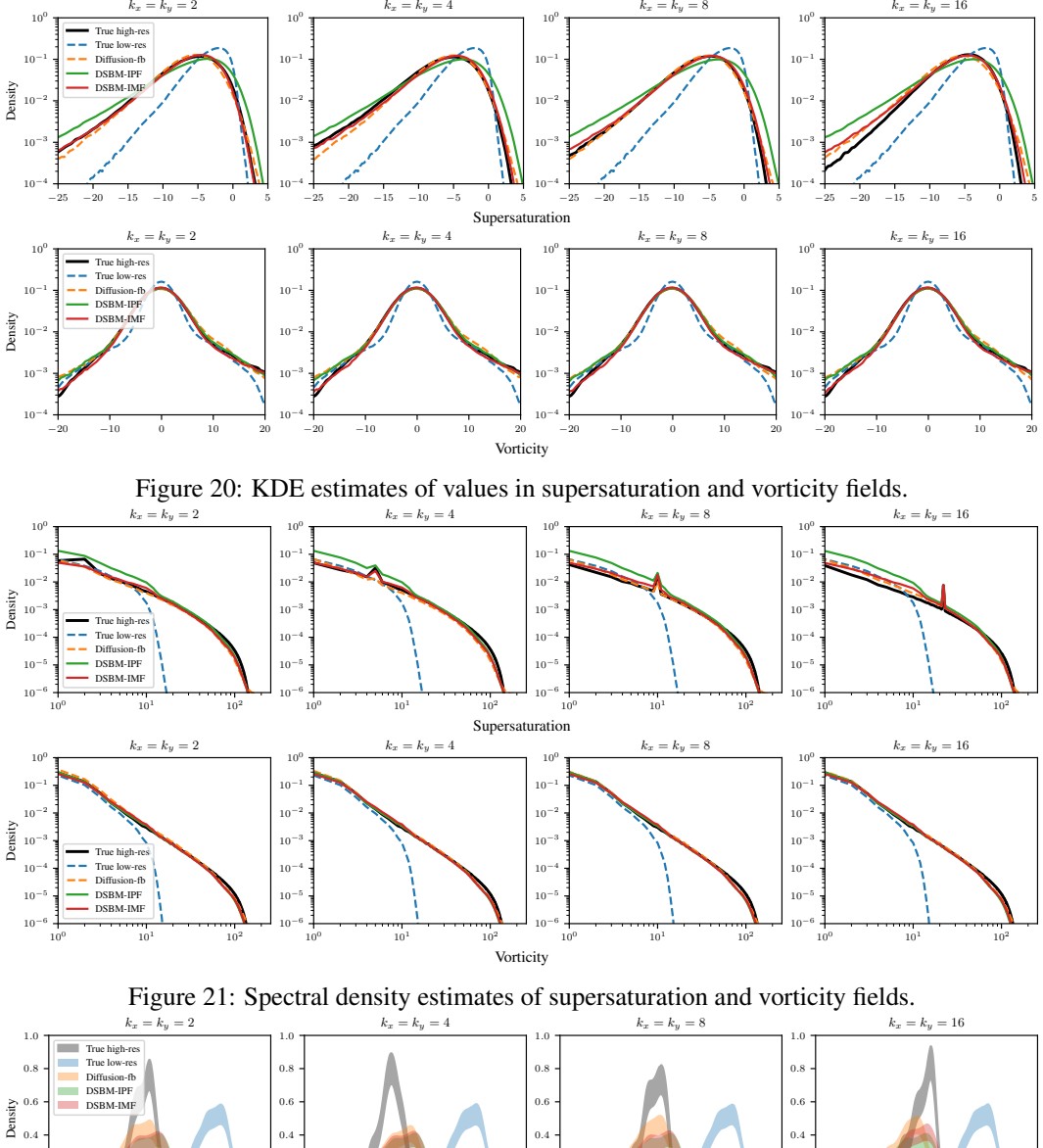

Figure 20: KDE estimates of values in supersaturation and vorticity fields.

Figure 21: Spectral density estimates of supersaturation and vorticity fields.

Figure 22: KDE estimates of spatial means of the supersaturation field. The shaded areas denote 99% confidence interval obtained using 10000 bootstrap samples.

in machine learning, but also natural science areas such as physics, biology and geosciences in which optimal transport maps with theoretical guarantees are appealing. Our fluid flows experiment demonstrates such potentials. However, as is the case for generative models as a whole, intentional malicious use could cause detrimental societal impacts.

