# OpenReview forum: "Diffusion Schrödinger Bridge Matching"
_NeurIPS.cc/2023/Conference — NeurIPS 2023 poster_

### Official Review · Reviewer_Lk9k · 2023-07-01

**Soundness:** 3 good
**Presentation:** 3 good
**Contribution:** 3 good
**Rating:** 6
**Confidence:** 4

**Summary:**

This paper formulates a new framework of a probabilistic model called Diffusion Schrödinger Bridge Matching (DSBM) and an iterative algorithm which is called iterative Markovian fitting (IMF). Firstly, the paper has contribution of pointing out a relationship between famous generative models (score matching and flow matching) in the form of DSBM. IMF is also novel and might be more suitable to be trained in neural networks than well-established algorithms such as IPF. To achieve these, the authors utilize a tractable bridge structure (Brownian bridge), which enables preserving key properties, i.e., densities of marginal distribution at the start and end of the process. The proposed algorithm often shows better generation results in certain tasks, such as image transfer.


**Strengths:**

1. The paper is well-written despite dealing with a complex subject. As far as I understand, most of the arguments seem to be theoretically sound.
2. SB algorithms have suffered from numerical errors and computational complexity. Since the IMF approach is akin to standard matching algorithms for neural networks, the provided empirical results show that it scales well.
3. The paper provides a novel perspective on the diffusion bridge problem with a novel algorithm. As a result, DSBM-IMF shows promising results in certain tasks.

**Weaknesses:**

1. Significance. In Tables 2, 3 and Figure 10, It is difficult for me to observe that DSBM-IMF excels DSMB-IPF. Therefore, I would say significance of the proposed algorithm is not clearly shown in the experiments. The paper needs more empirical evidence of the actual benefits of choosing this algorithm, such as performance boost and scalability, in more challenging, or large-scale settings.
2. Limited applications. I believe the actual implementation of the algorithm heavily relies on the well-known Brownian bridge. Although I think this is actually a valid contraint for certain applications such as image generation, at the same time it means that the algorithm is not always applicable to general optimal transport problems. Therefore, I recommend putting more effort on analyzing more classes of $\mathbb{Q}$ in the experiment (if there are any), or putting clear justification on certain classes of $\mathbb{Q}$ used in the experiment.


**Questions:**

1. Does $f_t$ is predefined in all tasks?
2. Is $\sigma_t$ constant and **not** scheduled? The authors of [1] mentioned time-symmetric scheduling the diffusion term, as suggested by prior SB models.
3. Could you explain how the IMF scheme is fundamentally better than IPF algorithms?



[1] Guan-Horng Liu et al. (2023), I2SB: Image-to-Image Schrödinger Bridge.

**Limitations:**

Although I think overall paper is good, but I also think the proposed experimental results are not quite sufficient; the paper need more empirical evidence to show the excellence of the model. I hope to see response from the authors for this point.

---

> ### Author Rebuttal · Authors · 2023-08-09
>
> We thank the reviewer for their feedback and their positive evaluation. We appreciate their interest in our work.
> We address here the reviewer's concerns raised in the review.
>
> **“Difficult to observe DSBM-IMF excels DSMB-IPF ...”**: We would like to first make a clarification that DSBM-IPF is also a proposed novel algorithm in our work. While DSBM-IPF is based on the classical IPF theory, it is different from previous diffusion-based algorithms based on IPF such as DSB which we compare to in the paper. Comparing DSBM-IPF with DSB, our proposed methodology achieves significantly better results, see Figure 5. On the other hand, comparing DSBM-IPF with DSBM-IMF, they exhibit a comparatively minor difference like you mentioned. Algorithmically, the two DSBM methods only differ in the initialization of $\Pi^0$ in the first iteration. To summarize, DSBM-IMF and DSBM-IPF are both original contributions of the paper (differing mostly on their theoretical foundations while being very close in practice).
>
> **“Actual implementation of the algorithm heavily relies on the well-known Brownian bridge ...”**: We thank the reviewer for this comment. We refer to the "Choice of diffusion bridge" section in our response to all reviewers for a detailed response.
>
> **“More empirical evidence”**: We have demonstrated in our paper the usefulness of our proposed method in a number of high-dimensional tasks, such as CelebA ($128 \times 128$) and fluid flows downscaling ($512 \times 512)$. We have additionally performed an experiment on the AFHQ dataset at $512 \times 512$ resolution using the unmodified DSBM-IMF algorithm, and have included it in the one-page response PDF. Additionally, we note that the scalability of our method is comparable to the one of bridge matching, as our method can be viewed as a refinement of it.
>
> **“Is $f_t$ predefined ...?”**: Yes, the reference process $\mathbb{Q}$ has a similar role as the “forward noising process” in standard diffusion models. Two standard choices of $\mathbb{Q}$ include the Brownian motion for which $f_t(x_t)=0$, and the Ornstein-Uhlenbeck (OU) process for which $f_t(x_t)=-\frac{1}{2}x_t$.
>
> **“Is $\sigma_t$ constant and not scheduled? ...”**: We have used a constant schedule for the diffusion coefficient $\sigma_t$ in our work. We have experimented with a symmetric schedule of integration step sizes similar to the related works you mention, but did not observe any improvement. We agree that the schedules for both the noising term and the integration steps may be important directions to study for future work.
>
> **“Could you explain how the IMF scheme is fundamentally better than IPF algorithms?”**: In short, the IMF scheme always preserves $\pi_0, \pi_T$ as the marginals at the initial and final times, whereas the IPF scheme only reaches the SB solution $\mathbb{P}^\star$ which satisfies both marginal constraints in the limit. For extra details, we refer to the “Relationship between IPF and IMF” section in our response to all reviewers.

---

### Official Review · Reviewer_7Dnn · 2023-07-03

**Soundness:** 3 good
**Presentation:** 3 good
**Contribution:** 4 excellent
**Rating:** 7
**Confidence:** 3

**Summary:**

Flow matching and alpha blending have achieved tremendous attention in the matching problems. Although the straight trajectories based on $X_t = (1-\alpha)X_0+\alpha X_1$ yield **fast inference**, it doesn't necessarily mean they are efficient in score estimations in training. To make the training theoretically more efficient, rectified flow leverages the optimal transport ideas to minimize the transport cost. However, the stochastic alternative is still lacking (stochastic interpolant may not be optimal in terms of optimal transport). To tackle this issue, the authors proposed the stochastic version of flow matching/ alpha blending with entropic optimized transport. A novel scheme IMF that resembles the IPF iteration is proposed.


**Strengths:**

1. Despite the advances of stochastic interpolant, the transport from marginals may not be optimal in general. The **optimal-transport**-based stochastic version of flow matching and alpha blending is **still missing** in the community.

2. In my understanding, the proposal of the IMF algorithm is the key/ most important contribution of this paper.

3. solid experiments and comparisons are studied, which verify the promising potential of DSBM.

4. Connections to flow matching, rectified flow, and stochastic interpolant are studied, which makes the authors easier to understand the benefit of this paper.

**Weaknesses:**

1. Clearer clarification between IPF and IMF would be required to facilitate the understanding. IPF is based on forward K; IMF is based on backward KL, this is good in math, but more detailed information is lacking. for example, if we solely check proposition 2, we can derive almost the same loss based on forward KL (which leads to the IPF algorithm) except that the underlying measure is not based on $\Pi_{0, t}$. So **what is the real benefit/ advantages of simulating $\Pi_{0, t}$ to train the loss in proposition 2 would clarify our concerns**

2. Algorithm 1 is not clear enough. The authors may consider providing us more details in the learning of $\upsilon_{\phi^{\star}}$ in Alg 1. For example, in the training of alpha-blending (line 60-64 in https://github.com/tchambon/IADB/blob/main/iadb.py), given some samples of $x_1$ and $x_0$, very simple code would be used to train the models. However, when it comes to DSBM, how is it implemented? pseudocode would work if it clarifies our concerns.

3. My concern is that this algorithm may be quite expensive.




Minor

1. missing one related work:

    Discussions on the accumulations of IPF errors based on general optimal-transport cost functions studied in [1] would be appreciated.

    [1] Provably Convergent Schrödinger Bridge with Applications to Probabilistic Time Series Imputation. ICML'23.


2. writing of Equation between line 637 and line 638 is weird, because given X_t=x_t and x_T, what is the randomness there and why do we need expectation w.r.t. $\Pi_{T|t}$?

3. derivation of eq after line 113 is not clear to me. why $E_{\Pi_{0, T}}[||X_0-X_T||]^2$? is there any reference for that?

4. first part of section 4 in the main paper, the need the alternating schemes is still not well motivated

5. what is IMF-b?

**Questions:**


1. DDMs can be seen as the first iteration of DSM, which implies that we can use DDMs model weights to initialize the model weights of DSM. Does it mean that empirically we can also use weights from stochastic interpolant / the stochastic version of flow matching to train the first step of DSBM?

2. In line 993 in the appendix, the authors mentioned that "The advantage of this DSM loss is that it does **not rely on any divergence computation**". I believe this is quite an important part of scalability, but why it is only emphasized in the joint training instead of stating it in the main paper?

---

> ### Author Rebuttal · Authors · 2023-08-09
>
> We thank the reviewer for your thorough review. We will incorporate your suggestions regarding the clarity of Algorithm 1 and the related work. We now address your comments in more detail.
>
> **“Clearer clarification between IPF and IMF”**: Like you mentioned, we have related IPF with the forward KL, and IMF with the backward KL; however, they are not simply interchangable by switching the direction of KL in Proposition 2. We refer to the “Relationship between IPF and IMF” section in our response to all reviewers for details.
>
> **“Algorithm 1 is not clear enough...”**: We were constrained by space in the main paper to describe the algorithm fully. However, we plan to revise the paper to include a more detailed and down to earth description of the algorithm. In what follows, we include a more complete description of the algorithm. We start by describing a function DatasetUpdate which updates the dataset.
>
> >Input: direction $d$ (forward or backward), drift $v$
>
> >If $d$ is forward do
>
> >>Sample $\mathbf{X}_0\sim\pi_0$
>
> >>Simulate $\mathrm{d}\mathbf{X}_t=v(t,\mathbf{X}_t)\mathrm{d}t+\sigma\mathrm{d}\mathbf{B}_t$
>
> >>Output Dataloader($(\mathbf{X}_0,\mathbf{X}_T)$)
>
> >If $d$ is backward do
>
> >>Sample $\mathbf{Y}_0\sim\pi_T$
>
> >>Simulate $\mathrm{d}\mathbf{Y}_t=v(t,\mathbf{Y}_t)\mathrm{d}t+\sigma\mathrm{d}\mathbf{B}_t$
>
> >>Output Dataloader($(\mathbf{Y}_T,\mathbf{Y}_0)$)
>
> Equipped with this function we are now ready to describe the full pseudocode of DSBM-IMF.
>
> >Input: forward drift network $v_\theta$, backward drift network $v_\phi$
>
> >Initialize $d$ = backward and PairedDataset with independent samples $\mathbf{X}_0\sim\pi_0$ and $\mathbf{X}_T\sim\pi_T$
>
> >For $n\in\{0,\dots,N-1\}$ do
>
> >>While not converged do
>
> >>>Sample $t\sim\mathrm{Unif}([0,T])$
>
> >>>Sample $(\mathbf{X}_0,\mathbf{X}_T)\sim$ PairedDataset and $\mathbf{Z}\sim\mathrm{N}(0,\mathrm{Id})$
>
> >>>Compute $\mathbf{X}_t=(1-t/T)\mathbf{X}_0+(t/T)\mathbf{X}_T+\sigma\sqrt{t(T-t)/T}\mathbf{Z}$
>
> >>>If $d$ is forward $v=v_\theta$; update $\theta$ using ADAM on the loss function $\|v_\theta(t,\mathbf{X}_t)-(\mathbf{X}_T-\mathbf{X}_t)/(T-t)\|^2$
>
> >>>If $d$ is backward $v=v_\phi$; update $\phi$ using ADAM  on the loss function $\|v_\phi(T-t,\mathbf{X}_t)-(\mathbf{X}_0-\mathbf{X}_t)/t\|^2$
>
> >>Update PairedDataset using DatasetUpdate$(d,v)$
>
> >>If $d$ is forward change it to backward; if $d$ is backward change it to forward
>
> In practice, we can use caching to reuse samples from PairedDataset. We hope that this pseudocode resolves the
> concerns of the reviewer and we are happy to provide more details if more
> clarifications are needed.
>
> **“My concern is that this algorithm may be quite expensive.”**: For the first stage of DSBM, there is no computational difference with bridge matching. Later stages are trained similarly to Rectified Flow. The main computational complexity is that the coupling $(\mathbf{X}_0,\mathbf{X}_T)$ is not independent but given by the previous iteration. This requires running a sampling algorithm and is shared between DSBM, DSB and Rectified Flow. In practice, the computational load of this step can be greatly reduced using caching strategies.
>
> **“Missing one related work...”**: We thank the reviewer for pointing us to [1]. In [1], the authors study the bias accumulation for IPF when the potentials are approximated. The results can be applied to provide error bounds on the marginals. However, in the IMF procedure, there does not exist potentials associated with the path measure in the sense of [1]. It would be extremely interesting to derive similar results for the IMF sequence. We leave this study for future work.
>
> **“Writing of Equation between line 637 and line 638 ...”**: Thank you for this question. The variables $x_0$ and $x_T$ in lines 637-642 should all be changed to $\mathbf{X}_0$ or $\mathbf{X}_T$. We will make this modification in the paper.
>
> **“Why $\mathbb{E}\_{\Pi_{0,T}}[||X_0-X_T||]^2$?”**: One reference for this result is [2] around equation (1.2). This is a standard result when $\mathbb{Q}$ is a Brownian process $\sigma \mathbf{B}_t$.
>
> **“The need of alternating schemes...”“what is IMF-b?”**: As detailed in lines 186-192, a naive version of IMF can be implemented by iteratively performing Markovian projections in the forward-time direction using equation (9). Due to time-symmetry (Proposition 8), IMF can also be derived in the reverse-time direction using equation (13). They are essentially the same algorithm, and only differs in the direction of transport we learn. We name the latter "IMF-b" which stands for backward-only IMF. However, as shown in Figure 3, IMF-b suffers from increasing error as the number of iterations $n$ increases. This is in line with the analysis in lines 190-192. On the other hand, Figure 3 shows DSBM-IMF does not suffer from the same issue. This indicates that the forward-backward scheme is highly useful for avoiding bias accumulation and improving accuracy in the learned marginals.
>
> **“Can use pretrained weights in first step of DSBM?”**: That is correct. The first iteration of DSBM coincides with bridge matching, and so DSBM can also be interpreted as further refinements of bridge matching.
>
> **“The advantage of this DSM loss is that it does not rely on any divergence computation...”**: We did not mention this advantage in the main paper because some previous SB methods (such as DSB) also don't rely on divergence computation. However, we agree with the reviewer that any loss based on divergence (like the ISM loss) may suffer from scalability issues.
>
> We thank the reviewer for your suggestions to clarify our contribution and the pseudocode. We hope that our rebuttal resolves the main concerns of the reviewer.
>
> [1] Chen, Deng, Fang, Li, Yang, Zhang, Rasul, Schneider, Nevmyvaka (2023) -- Provably Convergent Schrödinger Bridge with Applications to Probabilistic Time Series Imputation
>
> [2] Léonard (2013) -- A survey of the Schrödinger problem and some of its connections with optimal transport

---

> > ### Comment · Reviewer_7Dnn · 2023-08-12
> > **reply**
> >
> > I appreciate the authors' detailed reply and thanks for the suggested reference. I am satisfied with the response.

---

> > > ### Author Response · Authors · 2023-08-18
> > >
> > > Thank the reviewer for your response and your acknowledgement of the rebuttal.

---

### Official Review · Reviewer_24jx · 2023-07-05

**Soundness:** 3 good
**Presentation:** 3 good
**Contribution:** 3 good
**Rating:** 7
**Confidence:** 2

**Summary:**

The submission suggests a numerically effective approach for solving the Schroedinger Bridge (SB) problem and illustrates its potential for generative modeling. The approach generalizes in some sense some recent flow matching methods. Furthermore, it provides a more efficient (no full trajectory caching) algorithm with less error accumulations that better remembers the prior reference measure for solving SB problems compared to previous works. Empirical results indicate for instance that it can yield better generative performance for MNIST-EMNIST transfer compared to previous work, or better reconstructions for downscaling geophysical fluid dynamics.

**Strengths:**

Originality: The proposed methodology and algorithm is novel as far as I am aware. The work provides additional contributions compared to cited concurrent work [Peluchetti (2023)] that shares some similar ideas. Related literature and approaches are discussed in detail.

Quality: Claims are adequately supported by proofs, and numerical experiments seem to support this.

Clarity: The paper is well written. The presentation is quite rigorous.

Significance: In my opinion, this work is of interest to the community as the method (i) generalizes recent flow matching approaches with non-degenerate noise and (ii) improves experimentally previous Diffusion Schrödinger Bridge approaches and (iii) provides some theoretical guarantees for solving the SP problem (under idealized assumptions).

**Weaknesses:**

It was not clear to me how the computational cost of the suggested method compares to related works per iteration or diffusion step (e.g. in Figures 5 and 16).


**Questions:**

In Figure 16, it appears that FID can be improved by using more than 100 steps. Does DBSM obtain better FID values compared to Bridge matching for more than 100 steps?

It was not clear to me what to take away from some parts of the appendix, such as as E Discrete-Time Markovian Projection or the Brownian/OU bridge representations in B.2. Are the results in B.2 helpful for saying anything about how to specify the SDE in the forward process $X$ for the suggested approach?



**Limitations:**

Yes

---

> ### Author Rebuttal · Authors · 2023-08-09
>
> We would like to thank the reviewer for their thorough evaluation of our work. We appreciate your acknowledgment of the paper’s merits. Here we would like to expand on the other comments raised in the review.
>
> **“Computational cost of the suggested method”**: At training time, our proposed method has comparable computational cost with previous SB methods such as DSB [1], but is more costly than flow matching [2] and bridge matching [3,4,5,6] (however, the first iteration of DSBM corresponds to bridge matching and so has comparable cost).
> DSBM can be seen as a refinement of bridge matching, and so pre-trained bridge matching models can be used to initialize DSBM.
> The additional training cost comes from the trajectory caching procedure in subsequent iterations (see footnote 4), which requires sampling from the SDE model trained in the previous iteration.
> For instance, at iteration $n+1$ we need to sample from $\mathbb{M}^n$ and save a batch of joint samples $(\mathbf{X}_0, \mathbf{X}_T)$.
> This batch of samples can then be cached and looped over for a number of epochs to sample different $\mathbf{X}_t$ at different values of $t$.
> Under reasonable assumptions on the SDE simulation cost and the number of epochs, the training cost averaged per iteration is about 1.5-2 times the training cost of flow matching and bridge matching (which is the case in e.g. Figure 5).
> Also, as both the forward and backward models are trained, the total training cost is doubled.
> On the other hand, DSBM is about 30\% more efficient than DSB, as the trajectory caching procedure in DSBM requires less NFEs than DSB (see lines 944-954).
>
> At sampling time, the computational costs are all equal among all methods per diffusion step. In Figure 16, all methods have the same sampling cost for each vertical slice.
> Therefore, DSBM can attain similar sampling performance as flow and bridge matching with significantly less sampling cost.
> We also note that FID score is not the only metric for well-learned transport. Indeed, while the FID score quantifies information about the accuracy of the marginal distribution, we also need to check for the similarity between the samples $(\mathbf{X}_0, \mathbf{X}_T)$, see the Unpaired Fluid Flows Downscaling experiment for instance.
>
> **“Not clear what to take away from some parts of the appendix ...”**: In
> Appendix B, we showcase that the DSBM methodology can cover a large class of
> linear drifts. Our goal is to derive the OU bridge and push the methodology to
> have an equivalent of VPSDE used in standard diffusion models. We think such a
> formulation could be beneficial for generative modeling (when
> $\pi_T=\mathcal{N}(0,I)$) but we haven't thoroughly tested it. In our
> preliminary experiments, we did not observe significant benefits for using the
> OU bridge.
>
> Regarding Appendix E, we have
> derived a discrete-time version of all of our results for readers who would
> prefer to introduce the methods without having the knowledge of SDE. Another
> goal of Appendix E is to show that there is nothing in our method that is
> intrinsic to the continuous-time  framework and that our methodology could be recovered as
> the limiting case of a fully discrete numerical scheme.
>
> [1] De Bortoli, Thornton, Heng, Doucet (2021) -- Diffusion Schrödinger Bridge with Applications to Score-Based Generative Modeling
>
> [2] Lipman, Chen, Ben-Hamu, Nickel, Le (2022) -- Flow Matching for Generative Modeling
>
> [3] Delbracio, Milanfar (2023) -- Inversion by Direct Iteration: An Alternative to Denoising Diffusion for Image Restoration
>
> [4] Liu, Vahdat, Huang, Theodorou, Nie, Anandkumar (2023) -- I2SB: Image-to-Image Schrödinger Bridge
>
> [5] Heitz, Belcour, Chambon (2023) -- Iterative $\alpha$-(de)Blending: Learning a Deterministic Mapping Between Arbitrary Densities
>
> [6] Albergo, Boffi, Vanden-Eijnden (2023) -- Stochastic Interpolants: A Unifying Framework for Flows and Diffusions

---

> > ### Comment · Reviewer_24jx · 2023-08-15
> > **Response to authors**
> >
> > I thank the authors for their detailed response that have fully addressed my concerns or questions. I have also read the other reviews and intend to keep my accept score.

---

> > > ### Author Response · Authors · 2023-08-18
> > >
> > > Thank the reviewer for your response and your acknowledgement of the rebuttal.

---

> > > ### Author Response · Authors · 2023-08-18
> > >
> > > Thank the reviewer for your response and your acknowledgement of the rebuttal.

---

### Official Review · Reviewer_J5AJ · 2023-07-06

**Soundness:** 3 good
**Presentation:** 4 excellent
**Contribution:** 3 good
**Rating:** 5
**Confidence:** 4

**Summary:**

The paper introduces Iterative Markovian Fitting (IMF) as a new method to compute Schrödinger Bridges (SBs), which are dynamic versions of entropy-regularized optimal transport. IMF alternates between projecting on the space of Markov processes and the reciprocal class, and it preserves the initial and terminal distributions. The paper also proposes Diffusion Schrödinger Bridge Matching (DSBM) as an algorithm to approximate SB solutions derived from IMF. DSBM overcomes issues of previous techniques and solves a simple regression problem at each iteration. The performance of DSBM is demonstrated in various transport tasks. The paper provides theoretical results, notations, and definitions related to the topics discussed.

**Strengths:**

1, The paper introduces Iterative Markovian Fitting (IMF) as a new procedure for computing Schrödinger Bridges (SBs). This approach provides a fresh perspective and contributes to the field of optimal transport and generative modeling.
2, The paper establishes various theoretical results for IMF, demonstrating the validity and effectiveness of the proposed method. These results enhance the understanding of SBs and their applications in machine learning.

**Weaknesses:**

1， While the paper mentions that IMF and DSBM address scalability issues, it lacks a detailed analysis of their scalability in terms of computational resources and dataset sizes. Understanding the scalability limitations of the proposed methods is crucial for their applicability to real-world, large-scale problems.
2, The paper assumes locally Lipschitz continuity of drift and limited settings for the proposed methods. These assumptions may limit the applicability of IMF and DSBM to certain scenarios. A discussion of the limitations imposed by these assumptions would provide a more comprehensive view of the proposed methods.

**Questions:**

1, How does the Iterative Markovian Fitting (IMF) method differ from the traditional Iterative Proportional Fitting (IPF) approach in computing Schrödinger Bridges (SBs)?
2, Can you provide more insights into the theoretical results established for IMF in the paper?
3, How does Diffusion Schrödinger Bridge Matching (DSBM) overcome the time-discretization and "forgetting" issues of previous diffusion-based techniques?

**Limitations:**

As stated in "Weakness"

---

> ### Author Rebuttal · Authors · 2023-08-09
>
> We thank the reviewer for taking the time to review our submission and for their constructive feedback. We address here the main points from the review.
>
> **“Analysis of scalability”**: Our proposed DSBM method leverages tools from the recent flow/bridge matching literature, which are highly scalable generative models.
> In particular, the first stage of DSBM is the same as bridge matching, and so DSBM can be viewed as refinements of bridge matching in later stages.
> We have demonstrated our proposed method in a number of high-dimensional tasks, such as CelebA ($128 \times 128$) and fluid flows downscaling ($512\times512)$. We have additionally performed an experiment on the AFHQ dataset at $512 \times 512$ resolution using DSBM-IMF, and have included it in the one-page response PDF.
> Our experimental setup is fully described in the appendix of our paper (see Appendix I).
> Regarding the computational resources, our MNIST experiments were run on two 2080Ti GPUs,
> while the CelebA and fluid flows downscaling experiments were run on a single A100 or RTX GPU.
> In the case of image experiments, we use the standard dataloaders provided by the torchvision package.
> For the fluid flow experiment, further details on the climate dataset and preprocessing can be found in [1].
> We hope this clarifies the reviewer's concerns and are happy to provide further clarifications if needed.
>
> **“Limitations of locally Lipschitz continuity assumptions”**: We thank the reviewer for pointing out this
> limitation. First, we would like to highlight that in most applications the
> quantities of interest are locally Lipschitz (which is one of the weakest
> requirement regarding the regularity of the drift and diffusion
> matrix). Dropping this requirement might lead to technical difficulties (for
> instance it is known that the solutions of SDEs with locally Lipschitz
> coefficients have a unique strong solution, which is not the case if the drift
> is only assumed to be continuous [2]). However, it is likely that the local Lipschitzness assumption could be dropped and
> replaced with finite entropy conditions. Currently, we
> make the Lipschitzness assumption to use known results regarding the
> well-posedness of the Doob $h$-transform [3]. However, there is another line of work that studies Doob $h$-transform
> under finite entropy conditions [4]. These conditions might be more amenable
> to the study of Schrödinger Bridges. We plan to study further the theoretical properties of the IMF under this finite entropy setting in a future work.
>
> **“How does IMF differ from IPF”**: IPF and IMF schemes project on different classes of path measures (see Table 1). We refer to the “Relationship between IPF and IMF” section in our response to all reviewers for a more detailed comparison.
>
> **“More insights into the theoretical results established
> for IMF in the paper”**: In what follows, we give a high level explanation of
> the main results of the paper. Our main contribution is to show that the two
> alternate projections considered in the IMF are 1) indeed projections
> under the Kullback--Leibler divergence (Proposition 2 and 4);
> 2) that they satisfy a Pythagorean theorem (Lemma
> 6). The fact that these two projections satisfy a Pythagorean theorem is
> surprising and one of the important results of our work. Our results
> differ from classical information theoretic results [5] in that our
> Pythagorean theorems are stated for the backward KL divergence
> and not the forward one. This is a key difference which greatly simplifies the
> analysis. As a result, the concurrent work [6] was able to prove the
> convergence of IMF to the Schrödinger Bridge. We would like to emphasize
> that since the submission of this paper we have found a proof of the
> convergence of IMF to the SB which is shorter while similar in spirit to the one of [6].
> We summarize this proof below and
> and will include it in the revised version of the paper.
>
> Using the Pythagorean theorems, we can show that the KL
> divergence $\mathrm{KL}(\mathbb{P}^n|\mathbb{P}^\star)$ between the IMF iterates and the SB is
> finite. Using the coercivity of $\mathrm{KL}(\mathbb{P}^n|\mathbb{P}^\star)$
> (this is where our analysis differs from the one of the IPF) we have that
> the IMF sequence is relatively compact. Then, we use the space of Markov
> path measure and the reciprocal class are closed under weak convergence to
> show that the only limiting point of IMF must be Markov and in
> the reciprocal class. Since they met the marginal constraint at endpoints, the unique limit point is the Schrödinger Bridge (Proposition 5).
>
> **“How does DSBM overcome the time-discretization and 'forgetting' issues”**: DSBM straightforwardly overcomes the time-discretization issue, as both the diffusion bridge $\mathbb{Q}_ {|0,T}$ (e.g. equation (4)) and the training objectives (e.g. equations (9)(13)) are derived in continuous time. As for the "forgetting" issue, previous diffusion SB methods only use the reference process $\mathbb{Q}$ in the first iteration of the algorithm which leads to the "forgetting" issue. On the contrary, DSBM directly uses the diffusion bridge $\mathbb{Q}_{|0,T}$ in each iteration. This can be intuitively understood as incorporating $\mathbb{Q}$ as inductive bias in the DSBM algorithm. See also Appendix F for a more detailed discussion on how DSBM overcomes the "forgetting" issue.
>
> We hope that our answers have resolved the reviewer's concerns.
>
> [1] Bischoff, Deck (2023) -- Unpaired Downscaling of Fluid Flows with Diffusion Bridges
>
> [2] Ikeda, Watanabe (1996) -- Itô’s Stochastic Calculus and Probability Theory
>
> [3] Palmowski, Rolski (2002) -- A technique for exponential change of measure for Markov processes
>
> [4] Leonard (2011) -- Stochastic derivatives and generalized h-transforms of Markov processes
>
> [5] Csiszar (1975) -- I-Divergence Geometry of Probability Distributions and Minimization Problems
>
> [6] Peluchetti (2023) -- Diffusion Bridge Mixture Transports, Schrödinger Bridge Problems and Generative Modeling

---

### Official Review · Reviewer_Axbu · 2023-07-08

**Soundness:** 3 good
**Presentation:** 2 fair
**Contribution:** 2 fair
**Rating:** 6
**Confidence:** 3

**Summary:**

This paper proposes IMF, an iterative method for solving Schrodinger Bridge problems where the solution at each iteration preserves the correct marginal distributions at times 0 and T. This is in contrast to the existing method IPF which only satisfies this in the limit (and is very difficult to reach in practice).

**Strengths:**

  - An interesting marginal-preserving algorithm for solving Schrodinger bridge problems. The popular method IPF can have convergence problems in practice, while this seems to fix some of that.

**Weaknesses:**

  - Presentation. While I eventually understood the resulting IMF algorithm, I felt the presentation can be much more simplified. The algorithm is actually quite simple and closely related to priors works on SB and OT. While I understand the need to prove convergence and showcase novelty, the heavily front-loaded mathematical notation made reading on the first pass very difficult. I imagine the current paper may be off-putting for many readers who are not extremely familiar with related works.


  - Novelty. The proposed algorithm seems very similar to Rectified Flow but just with an additional noise parameter.


  - Limitation. The proposed algorithm only works for the standard formulation of Schrodinger bridge with quadratic costs. However, in this setting, it isn't clear to me why the extra stochasticity is useful, as opposed to solving deterministic maps (optimal transport problems).

**Questions:**

  - "Most notably, we adopt the SDE approach which is crucial for the validity of Proposition 5" - Can the authors expand on what breaks when there is no stochasticity? From what I understand, RF should also preserve marginal distributions and converge to the optimal transport solution, which is discussed more in [1]. The solutions are also unique similar to any Flow Matching-based approach.


  - All of the experiments in the main paper are currently only for transferring between two data distributions, but being able to find optimal transport maps is useful for generative modeling as well. I think it could really make the paper stronger by having experiments in generative modeling (Gaussian -> data) and showing some empirical metrics from e.g. [2] such as sampling cost and consistency, since these are important problems in generative modeling at the moment. [Edit: I found the experiments in I.5. I see that sampling efficiency is on par with RF but Table 6 seems pretty good! Although it does look like if we extrapolate to NFE > 100, the other methods are descending quite a bit faster in FID...]


  - Can the authors expand on the differences between the two proposed methods, DSBM-IPF and DSBM-IMF? If I understood correctly, the main difference between DSBM-IPF and DSBM-IMF is only in the first iteration, where the pairs (x0, xT) are sampled from (either an OU process Q or the independent joint distribution). And the real difference is that Q has a slight bias at time T because it may not have reached its stationary distribution. Practically, this difference seems very negligible, yet there seems to be differences in empirical results (e.g. Table 3 and Figure 5). Theoretically, since Q does not match the correct marginal at T, this is also problematic for the proof of convergence to SB right?


[1] "Rectified Flow: A Marginal Preserving Approach to Optimal Transport" https://arxiv.org/abs/2209.14577


[2] "Multisample Flow Matching: Straightening Flows with Minibatch Couplings" https://arxiv.org/abs/2304.14772

**Limitations:**

The authors do mention limitations such as marginal improvement, computational issues, and difficulties when sigma --> 0. I think these are sufficient.

---

> ### Author Rebuttal · Authors · 2023-08-09
>
> We thank the reviewer for their thorough evaluation of our work. We appreciate their interest and their thoughtful questions. We would like to address the raised questions here which hopefully would clarify the role of stochasticity further.
>
> **“Algorithm presentation can be simplified.”**:
> We have tried to make the terminology clearer with the
> introduction of a Notation section. The introduction of some mathematical
> concepts like the Markovian and reciprocal projections seems
> important to us, as it allows us to describe within a same framework many
> different existing algorithms ([1,2,3,4,5] can be seen as special case of the
> Markovian projection procedure for instance). However, we understand the
> reviewer's concerns and will try to improve the clarity of the work in the revised version of the paper.
> If there are specific presentation issues the reviewer would like to point out,
> we would be happy to change them as well.
>
> **“Role of stochasticity”**: We thank the reviewer for this question. Indeed, it can be unclear at first why stochasticity is important. We would like to provide three motivations regarding the use of stochasticity:
>
> * Firstly, from a theoretical point of view, stochasticity is important for
> establishing the convergence of the IMF algorithm to the SB solution.
> The IMF methodology provably converges to the unique EOT/SB solution, see [8, Theorem 2] (concurrent work) for a proof.
> We have also found a short proof of this which we will include in the revised version of the paper.
> In contrast, such results appear to be more difficult to establish for
> Rectified Flow (RF).
> The RF algorithm can only provably solve the OT problem when the learned vector fields $v_t(x_t)$ are restricted to gradient
> fields, and counterexamples exist if this condition does not hold [7].
> Even with the above restriction on the vector fields, the
> strongest convergence results for RF as of now appear to be Theorem 5.6 and
> Corollary 5.7 in [7], which rely on a surrogate loss and does not
> directly prove the convergence of the learned coupling to the OT map.
>
> * Empirically, we also encountered difficulties with the original RF formulation
> for transfer tasks, and we observe adding stochasticity can improve
> performance significantly. Qualitatively speaking, in the case of image
> transfer, when $\sigma$ is close to zero (i.e. our approach is closer to RF)
> the learned process only changes minimally the input samples.
> As a result, the output samples might look unrealistic, especially if the two domains are very different (for example interpolating
> between the classes "horse'' and "plane'' in CIFAR-10, see the additional
> results in the one-page PDF).
>
> * Finally, we would like to make a comparison between non-regularized OT and
> the SB problem. In itself, the Schrödinger Bridge problem is an important
> theoretical problem, with many recent works tackling it. Unlike
> non-regularized OT, the SB formulation can be useful when the solution does
> not necessarily have a one-to-one mapping structure. For example, in inverse
> problems (such as super-resolution), there are multiple solutions
> corresponding to a given condition, and so the solution mapping is one-to-many
> and the SB formulation is a more natural framework.
>
> We also would like to emphasize that another crucial difference with RF is
> that DSBM-IMF applies forward-backward training, whereas RF only trains the flow
> matching process iteratively in one direction. The latter will cause errors in
> the marginal distributions to accumulate, and marginal accuracy to deteriorate
> in each iteration, whereas DSBM-IMF can still improve marginal accuracy (as
> shown in e.g. Figure 5).  Therefore, we believe our proposed method is a novel
> and relevant contribution, which also helps to
> unify the flow matching and SB research areas.
>
> We do hope that our clarification of the main differences between our algorithm
> and RF [6], as well as our justification of the use of stochasticity, have resolved the concerns raised in the
> review.
>
> **“Differences between DSBM-IPF and DSBM-IMF”**: Your understanding is indeed
> correct. The DSBM-IPF and DSBM-IMF algorithms only differ in the first
> iteration, and when the OU process is used their differences should be
> comparatively small. We use Brownian process in our work motivated by optimal
> transport, and their difference is more significant. However, both
> algorithms converge theoretically to the true SB solution. In the case of DSBM-IPF,
> although the process $\mathbb{Q}$ does not match with $\pi_T$ at time
> $T$, by Proposition 9 DSBM-IPF can recover the true IPF sequence
> $(\tilde{\mathbb{P}}^n)$ (see lines 118-125), for which a different, classical
> proof of convergence of IPF exists. To summarize, both algorithms converge to
> the true SB (assuming that the score is perfectly learned),
> however the theoretical frameworks are very
> different. We will highlight this in the revised version of the paper.
>
> Overall, we would like to thank the reviewer for their insightful questions
> which have helped us to clarify several points of the paper, in particular the
> importance of stochasticity.
>
> [1] Delbracio, Milanfar (2023) -- Inversion by Direct Iteration: An Alternative to Denoising Diffusion for Image Restoration
>
> [2] Liu, Vahdat, Huang, Theodorou, Nie, Anandkumar (2023) -- I2SB: Image-to-Image Schrödinger Bridge
>
> [3] Lipman, Chen, Ben-Hamu, Nickel, Le (2022) -- Flow Matching for Generative Modeling
>
> [4] Heitz, Belcour, Chambon (2023) -- Iterative $\alpha$-(de)Blending: Learning a Deterministic Mapping Between Arbitrary Densities
>
> [5] Albergo, Boffi, Vanden-Eijnden (2023) -- Stochastic Interpolants: A Unifying Framework for Flows and Diffusions
>
> [6] Liu, Gong, Liu (2022) -- Flow Straight and Fast: Learning to Generate and Transfer Data with Rectified Flow
>
> [7] Liu (2022) -- Rectified Flow: A Marginal Preserving Approach to Optimal Transport
>
> [8] Peluchetti (2023) -- Diffusion Bridge Mixture Transports, Schrödinger Bridge Problems and Generative Modeling

---

### Author Rebuttal · Authors · 2023-08-09

We would like to thank all reviewers for their time and their very helpful feedback.
We will take care to address all suggestions on improving clarity and minor typos in the next version of the paper.
We would like to take the opportunity here to address a few common questions raised by several reviewers.

**“Relationship between IPF and IMF”**: Several reviewers are interested in a more detailed discussion on the relationship between the classical IPF and the novel IMF schemes. In short, IPF and IMF schemes use *different projections* on *different classes* of path measures (see Table 1 for a summary). In more detail, IPF uses alternative forward KL projections onto the path measure classes $\\{\mathbb{P}: \mathbb{P}_0=\pi_0\\}$ and $\\{\mathbb{P}:\mathbb{P}_T=\pi_T\\}$. IMF uses alternative backward KL projections onto the classes $\mathcal{M}$ (Markovian class) and $\mathcal{R}(\mathbb{Q})$ (reciprocal class).
The true Schrödinger Bridge solution $\mathbb{P}^\star$ is the unique path measure in all 4 classes.
A fundamental difference is that the IMF iterates always admit $\pi_0, \pi_T$ as the marginals at the initial and final times. In contrast, each IPF iterate only satisfies one of the marginal constraints (odd iterate satisfies $\tilde{\mathbb{P}}^{2n+1}_T=\pi_T$, even iterate satisfies $\tilde{\mathbb{P}}^{2n+2}_0=\pi_0$, see line 119), and they only reach the SB solution $\mathbb{P}^\star$ which satisfies both marginal constraints in the limit as $n\to\infty$. Therefore, the IMF scheme is more preferable when the marginal accuracy of the samples at the initial and final times are important, and we would like to obtain more accurate samples before the algorithms have converged.

From a practical point of view, IMF also gives rise to more stable training procedures. The reciprocal projection can be carried out empirically by simulating the modeled SDEs, while the Markovian projection leverages tools from the flow/bridge matching literature [1,2,3,4,5]. In contrast, in DSB there is an additional approximation that the transition density is Gaussian for small stepsizes. This assumption is only true for small stepsizes and can make the loss of DSB quite unstable. In addition, DSB suffers from bias accumulation throughout the IPF iterations. The IMF scheme circumvents this issue since we interpolate between the endpoints using the Brownian bridge.

**“Choice of diffusion bridge”**: We use a Brownian motion as our reference process $\mathbb{Q}$ in our experiments, as it corresponds to solving the standard Wasserstein-2 EOT problem (see line 113). It is also the choice used in previous bridge matching papers [1,2,3,4]. We have analyzed a larger class of linear diffusion bridges in Appendix B. Extending bridge matching procedure to non-linear processes would be non-trivial and is still an open question,
as this requires solving and sampling from an intractable diffusion bridge.
However, even if the diffusion bridge is not available in close form, one might be able to first learn the diffusion bridge using methodologies such as [6], and then apply DSBM using the learned diffusion bridge.
This extension should also be motivated by specific applications, which we leave for future work. In our case, the choice of Brownian motion is motivated by the optimal transport point of view.

**“FID with more than 100 steps”**: We have performed further evaluations for the CIFAR-10 generative modeling task in Appendix I.5 with NFEs higher than 100 as suggested by several reviewers. We have included the result in our submitted one-page response PDF. We observe that at higher NFEs, DSBM-IMF still achieves better FID scores than bridge matching (BM) and rectified flow (RF), but achieves slightly worse FID than conditional flow matching (CFM) and OT-CFM for NFE ≥ 300. Overall, DSBM-IMF is effective in improving sample quality when the NFE is very low, but unlike RF, DSBM-IMF still results in comparably low FID when the NFE is very high.

**“Scalability of DSBM”**: The scalability of our method should be comparable to bridge matching, as our method can be viewed as a refinement of it. For the first stage of DSBM, there is no computational difference between our proposed scheme and [5].
Later stages correspond to refinements of the process using improved couplings $(\mathbf{X}_0, \mathbf{X}_T)$, which involves a sampling process similar to DSB and Rectified Flow. The computational load of this step can be
greatly reduced using caching strategies.
We have demonstrated in our paper the usefulness of our proposed method in a number of high-dimensional tasks, such as CelebA ($128 \times 128$) and fluid flows downscaling ($512 \times 512)$. We have additionally performed an experiment on the AFHQ dataset at $512 \times 512$ resolution using DSBM-IMF, and have included it in the one-page response PDF.

[1] Delbracio, Milanfar (2023) -- Inversion by Direct Iteration: An Alternative to Denoising Diffusion for Image Restoration

[2] Liu, Vahdat, Huang, Theodorou, Nie, Anandkumar (2023) -- I2SB: Image-to-Image Schrödinger Bridge

[3] Heitz, Belcour, Chambon (2023) -- Iterative $\alpha$-(de)Blending: Learning a Deterministic Mapping Between Arbitrary Densities

[4] Albergo, Boffi, Vanden-Eijnden (2023) -- Stochastic Interpolants: A Unifying Framework for Flows and Diffusions

[5] Lipman, Chen, Ben-Hamu, Nickel, Le (2022) -- Flow Matching for Generative Modeling

[6] Heng, De Bortoli, Doucet, Thornton (2021) -- Simulating Diffusion Bridges with Score Matching

---

### Comment · Area_Chair_DjHQ · 2023-08-11

 Dear reviewers and authors,

Thank you very much for your work on this submission and its evaluation. Now that the authors have responded to the reviews, I **strongly encourage** the reviewers to acknowledge the review, to look at other reviews and rebuttals for this submission, and to adjust their scores if needed. Thanks to those that have already done so.

Authors have the possibility to reply if further questions are needed, until the 16th.

Thank you very much to all,

Area Chair

---

### Decision · Program_Chairs · 2023-09-21

**Decision:**

Accept (poster)

**Comment:**

The paper proposes a novel numerical approach for solving the Schroedinger Bridge (SB) problem and
show its potential impacts for generative modeling. Experimental results illustrate the the benefit
it bring for generative modelling.

Overall, all reviewers are happy about the paper  and the rebuttal and  propose to accept the paper.